# On the Ergodicity, Bias and Asymptotic Normality of Randomized Midpoint Sampling Method

**Ye He**
University of California, Davis
leohe@ucdavis.edu

**Krishnakumar Balasubramanian**
University of California, Davis
kbala@ucdavis.edu

**Murat A. Erdogdu**
University of Toronto and Vector Institute
erdogdu@cs.toronto.edu

## Abstract

The randomized midpoint method, proposed by [40], has emerged as an optimal discretization procedure for simulating the continuous time underdamped Langevin diffusion. In this paper, we analyze several probabilistic properties of the randomized midpoint discretization method, considering both overdamped and underdamped Langevin dynamics. We first characterize the stationary distribution of the discrete chain obtained with constant step-size discretization and show that it is biased away from the target distribution. Notably, the step-size needs to go to zero to obtain asymptotic unbiasedness. Next, we establish the asymptotic normality of numerical integration using the randomized midpoint method and highlight the relative advantages and disadvantages over other discretizations. Our results collectively provide several insights into the behavior of the randomized midpoint discretization method, including obtaining confidence intervals for numerical integrations.

## 1 Introduction

We consider the problem of computing the following expectation

$$\mathbb{E}_\pi[\varphi(x)] \quad \text{where} \quad \pi(x) = \tfrac{1}{Z_f} e^{-f(x)}, \tag{1}$$

for a potential function $f : \mathbb{R}^d \to \mathbb{R}$ and a test function $\varphi : \mathbb{R}^d \to \mathbb{R}$, when the normalization constant $Z_f = \int e^{-f(x)} dx$ is unknown. This problem frequently arises in statistics and machine learning with numerous applications to high-dimensional Bayesian inference [45, 24, 30, 10], numerical integration [21, 19], volume computation [43], optimization and learning [37, 17, 32], graphical models [20], and molecular dynamics [34, 22]. Markov chain Monte Carlo (MCMC) methods provide a powerful framework for computing the integral in (1), and have been successfully deployed in various scientific fields [26].

In particular, MCMC algorithms that are based on diffusion processes have received a lot of attention recently. The fundamental idea behind such algorithms is that a continuous-time diffusion with its invariant measure as the target $\pi$ is approximately simulated via a numerical sampler. The intuition behind the success of these methods is that by appropriately selecting the step-size parameter, the discrete approximation resulting from the numerical sampler tracks the continuous-time diffusion. Thus, rapid convergence properties of the diffusion process (see, for example, [38, 23, 14, 15, 27, 12]) is inherited by the discrete algorithm with an invariant measure that

is close to that of the diffusion, which is the target $\pi$. While a variety of diffusion processes can lead to a rich class of MCMC samplers, algorithms that are based on discretizing Langevin dynamics have been the primary focus of research due to their simplicity, accuracy, and well-understood theoretical guarantees in high-dimensional settings [6, 3, 4, 9, 42, 29, 3, 11, 13, 16].

Although motivated by the problem of computing the integral in (1), much of the theoretical focus on analyzing sampling methods in the recent literature has been on providing guarantees for the sampling problem itself (see [41] for an exception), i.e., the number of iterations needed to reach $\epsilon$-neighborhood of a $d$-dimensional target distribution in some probability metric. The choice of step-size of the sampler is crucial to obtain such theoretical guarantees. While the problem of estimating expectations such as in (1) is based on sampling from the target $\pi$ itself, the theoretical guarantees established for the sampling problem can provide very little to no information on computing the expectation in (1) based on the sampler. The main reason for this is, the step-size choice of the sampler required to obtain optimal theoretical guarantees for numerical integration of (1) turns out to be different from that of sampling. Furthermore, if the ultimate task is to perform inference on the quantity $\mathbb{E}_\pi[\varphi(x)]$, confidence intervals are required. Thus, one needs central limit theorems (CLT) to quantify the fluctuations of the estimator of the expectation in (1), depending on a specific numerical integrator being used.

The randomized midpoint method, a numerical sampler proposed by [40], has emerged as an optimal algorithm for sampling from strongly log-concave densities, achieving the information theoretical lower bound for this problem in terms of both dimension and tolerance dependency [1]. In lieu of this optimality result, one anticipates a superior performance from the randomized midpoint method in other fundamental problems that relies on a MCMC sampler as the main computation tool, e.g. estimating expectations of the form (1). However, properties of this sampler for the purpose of numerical integration, in particular its inferential properties, are not well-studied. In this paper, we explore various probabilistic properties of the randomized midpoint discretization method, when used as a numerical integrator. Towards that, we examine several results for the randomized midpoint method considering both the overdamped and underdamped Langevin diffusions. Our first contribution is the explicit characterization of the bias of the randomized midpoint numerical scheme, namely the difference between its stationary distribution and the target distribution $\pi$. We show that asymptotic unbiasedness, a desired property in general, can be achieved under a decreasing step-size sequence. As our principal contribution, we establish the ergodicity of the randomized midpoint method and prove a central limit theorem which can be leveraged for inference on the expectation (1). We compute the bias and the variance of the asymptotic normal distribution for various step size choices, and show that different step-size sequences are suitable for making inference in different settings.

**Our Contributions.** We summarize our contributions as follows:

1. We show the ergodicity of constant step-size (denoted as $h$) randomized midpoint discretization of the overdamped and underdamped Langevin diffusions in Theorems 1 and 3, respectively. For both cases, the stationary distribution $\pi_h$ of the resulting discretized Markov chain is unique and is biased away from the target distribution $\pi$.

2. The choice of a constant step-size for the randomized midpoint discretization causes bias in sampling. We characterize this bias explicitly in Propositions 2.2 and 3.1 for the overdamped and underdamped Langevin diffusions, respectively. We show that Wasserstein-2 distance between $\pi_h$ and $\pi$ is of order $\mathcal{O}(h^{0.5})$ and $\mathcal{O}(h^{1.5})$ respectively for the overdamped and underdamped Langevin diffusions.

3. The established order of bias points toward using particular choices of decreasing step-size sequence for the sake of inference. Specifically, we prove a CLT for numerical integration using the randomized midpoint discretization of the overdamped and underdamped Langevin diffusions in Theorems 2 and 4 respectively, for various choices of decreasing step-size. Depending on the specific choice of step-size sequence, the CLT is either unbiased or biased. When discretizing the overdamped Langevin diffusion with polynomially decreasing step-size choices, the rate of unbiased CLT turns out to be $\mathcal{O}(n^{(1/3)-\epsilon})$ for any $\epsilon > 0$. But the optimal rate turns out to be $\mathcal{O}(n^{1/3})$ for which one can only obtain a biased CLT. When discretizing underdamped Langevin diffusions with polynomially decreasing step-size choices, we show that the optimal rate can be improved to $\mathcal{O}(n^{5/8})$ under a certain condition, which is satisfied only by the class of constant test functions.

## 1.1 Notations and Preliminaries

We denote an $\ell$-th order symmetric tensor of dimension $d$ by $A \in \mathbb{R}^{d \otimes \ell}$. For a given vector $u \in \mathbb{R}^d$, we use $\|u\|$ to denote the Euclidean-norm of the vector. We define the $\ell$-th order rank-1 tensor formed from $u \in \mathbb{R}^d$ as $u^{\otimes \ell}$. In addition, let $A$ and $B$ be two $\ell$-th order tensors, we define the inner product between $A$ and $B$ as $\langle A, B \rangle = \sum_{j_1=1}^d \cdots \sum_{j_\ell=1}^d A_{j_1 j_2 \ldots j_k} \cdot B_{j_1 j_2 \ldots j_k}$. For a function $f : \mathbb{R}^d \to \mathbb{R}$, $\nabla f \in \mathbb{R}^d$ and $D^\ell \in \mathbb{R}^{d \otimes \ell}$ represents the gradient, and $\ell$-th order derivative tensor (for $\ell > 1$). We let $(\Omega, \mathcal{F}, P)$ represent a probability space, and denote by $\mathcal{B}(\mathbb{R}^d)$, the Borel $\sigma$-field of $\mathbb{R}^d$. We use $\xrightarrow{d}$ and $\xrightarrow{p}$ to denote convergence in distribution and probability respectively. The set of all twice continuously differentiable functions $f : \mathbb{R}^d \to \mathbb{R}$ is denoted as $\mathcal{C}^2(\mathbb{R}^d)$. We use $I_d$ to represent the $d \times d$ identity matrix. Let $x_0, x_1, \ldots$ be a $d$-dimensional Markov chain. The transition probability of the chain, at the $k$-th step is defined as $P^k(x, A) := P(x_k \in A | x_0 = x)$, for some $x \in \mathbb{R}^d$ and represents the probability that the chain is in set $A$ at time $n$ given the starting point was $x \in \mathbb{R}^d$. We use $\tilde{\mathcal{O}}$ to hide $\log$ factors. Finally, for a sequence $\gamma_k$ and positive integer $\ell$, we define $\Gamma_n^{(\ell)} := \sum_{k=1}^n \gamma_k^\ell$. We also make the following widely-used assumption on the potential function.

**Assumption 1.1.** *The potential function $f \in \mathcal{C}^2(\mathbb{R}^d)$ satisfies the following properties. For some $0 < m \le M < \infty$: (a) $f$ has a $M$-Lipschitz gradient; that is, $D^2 f \preceq M I_d$, and (b) $f$ is $m$-strongly convex; that is, $mI_d \preceq D^2 f$. We also define the condition number as $\kappa := M/m$.*

## 2 Results for the Overdamped Langevin Diffusion

The overdamped Langevin diffusion is described by the following stochastic differential equation:

$$dx(t) = -\nabla f(x(t))dt + \sqrt{2}dW(t), \tag{2}$$

where $W(t)$ is a $d$-dimensional Brownian motion. It is well-known that this diffusion has $\pi(x) \propto e^{-f(x)}$ as its stationary distribution under mild regularity conditions. In general, simulating a continuous-time diffusion such as (2) is impractical; thus, a numerical integration scheme is needed.

We now describe the *randomized midpoint discretization* of the above diffusion in (2), which we denote as RLMC. Denoting the $n$-th iteration of the algorithm with $x_n$, the integral formulation of the diffusion with $x_n$ as the initial value would then be $x_n^*(t) = x_n - \int_0^t \nabla f(x_n^*(s))ds + \sqrt{2}W(t)$. Let $h > 0$ be the choice of step size for the discretization and, let $(\alpha_n)$ be an i.i.d. sequence of random variables following uniform distribution on $[0, 1]$, i.e. $\alpha_n \sim U[0, 1]$. The fundamental idea behind the randomized midpoint technique is to use $h\nabla f(x_n^*(\alpha_{n+1}h))$ to approximate the integral $\int_0^h \nabla f(x_n^*(s))ds$. Indeed, notice that $\mathbb{E}[h\nabla f(x_n^*(\alpha_{n+1}h))] = h\int_0^1 \nabla f(x_n^*(\alpha h))d\alpha = \int_0^h \nabla f(x_n^*(s))ds$. RLMC proceeds by approximating $x_n^*(\alpha_{n+1}h)$ with the Euler discretization, which ultimately yields an explicit numerical integration step. Although [40] considered this discretization only for the constant step-size choice and the *underdamped* Langevin diffusion (which we discuss in Section 3), below we present a single iteration of the RLMC algorithm with the choice of variable step-size $\gamma_{n+1}$ for the overdamped diffusion in (2):

$$
\begin{aligned}
x_{n+\frac{1}{2}} &= x_n - \alpha_{n+1}\gamma_{n+1}\nabla f(x_n) + \sqrt{2\alpha_{n+1}\gamma_{n+1}}U'_{n+1}, \\
x_{n+1} &= x_n - \gamma_{n+1}\nabla f(x_{n+\frac{1}{2}}) + \sqrt{2\gamma_{n+1}}U_{n+1},
\end{aligned}
\tag{RLMC}
$$

where $(U_n)$ and $(U'_n)$ are sequences of i.i.d $d$-dimensional standard Gaussian vectors independent of $(\alpha_n)$ and the initial point $x_0$. We briefly digress now to make the following remark. If instead of $\alpha_n \sim U[0, 1]$, one uses $\alpha_n = 1$ for all $n$ deterministically, then the iterates of (RLMC) algorithm is reminiscent of the extra-gradient descent algorithm from the optimization literature [28], perturbed by Gaussian noise in each step. Furthermore, its noteworthy that with the deterministic choice of $\alpha_n = 1$, one cannot obtain the improved rates that the uniformly random $\alpha_n$ provides. Lastly, the filtration $(\mathcal{F}_n)$ is defined by $\mathcal{F}_n := \sigma(\alpha_k, U_k, U'_k; 1 \le k \le n)$, the smallest $\sigma$-algebra generated by the noise sequence and uniform random variables that are used in the first $n$ iterations.

### 2.1 Wasserstein-2 Rates for Constant Step-size RLMC

Before, we state our main result, we investigate a few important characteristics of the (RLMC) algorithm that are not explored yet. We start with its rate of convergence in Wasserstein-2 distance

(see [44] for definition) for the (RLMC) algorithm. The proof of the proposition below essentially follows from a similar idea of the more general result for the underdamped Langevin dynamics in [40]. We include the result with its proof for the sake of completeness.

**Proposition 2.1.** *Suppose $f$ satisfies Assumption 1.1. Set $x_0 = \arg\min_x f(x)$, $\gamma_n := h = \mathcal{O}(\epsilon^{2/3}/\kappa^{1/3}M)$ when $\kappa h M > 1$, and $\gamma_n := h = \mathcal{O}(\epsilon/M)$ when $\kappa h M \leq 1$ for some constant $C > 0$. After running the (RLMC) algorithm for*

$$K = \tilde{\mathcal{O}}\left(\frac{\kappa^{4/3}}{\epsilon^{2/3}} + \frac{\kappa}{\epsilon}\right) \quad steps,$$

*we have $W_2(\nu_K, \pi) \leq \epsilon\sqrt{d/m}$, where $\nu_K$ is the probability distribution of $x_K$.*

When $\kappa$ is of constant order, we see that $W_2$ rate is of order $\tilde{\mathcal{O}}(1/\epsilon)$. Notably, with the randomized midpoint technique, we obtain this particular $\epsilon$-dependency by discretizing just the overdamped Langevin diffusion with only the Lipschitz gradient condition on the potential function $f$. Prior works require Euler-discretization of higher-order Langevin diffusions to obtain a $W_2$ rate of order $\tilde{\mathcal{O}}(1/\epsilon)$ [8, 35] or require higher-order smoothness assumption along with other specialized discretization methods [39, 25, 10, 8].

## 2.2 Analysis of the Markov Chain Generated by Constant Step-size RLMC

Using the randomized midpoint technique, we obtain an improved dependency on $\epsilon$ for the $W_2$ rate under weaker assumptions while discretizing the Langevin diffusion in (2). Although not explicit from the proof of Proposition 2.1, the rate improvement is obtained by a careful balancing of bias and variance through the choice of step-size parameter $h$. In this section, in Theorem 1, we first show that the (RLMC) Markov chain is ergodic and has a unique stationary distribution, denoted by $\pi_h$. Due to the choice of constant step-size $h$, it's not hard to see that the stationary distribution of the (RLMC) chain is different from the stationary distribution $\pi$ of the Lanvegin diffusion in (2), i.e $\pi_h \neq \pi$. Hence, in Proposition 2.2, we characterize the Wasserstein-2 distance between $\pi$ and $\pi_h$.

Firstly, if $f \in \mathcal{C}^2(\mathbb{R}^d)$ and $f$ has a Lipschitz gradient with parameter $M$, then we can immediately see that the transition kernel of chain $(x_n)$, $P(x,y) \in \mathcal{C}(\mathbb{R}^d \times \mathbb{R}^d)$ is positive everywhere. Therefore, it's easy to obtain that the chain $(x_n)$ is $\mu^{\text{Leb}}$-irreducible and aperiodic. Given all this information, we can give a sufficient condition to make sure that the chain has a unique invariant probability measure, and it is ergodic.

**Theorem 1.** *Let the potential function $f$ satisfy part (a) of Assumption 1.1, and let $\gamma_n := h$ be small enough. Then the (RLMC) Markov chain $(x_n)$ has a unique stationary probability measure $\pi_h$, and for every $x \in \mathbb{R}^d$, we have*

$$\sup_{A \in \mathcal{B}(\mathbb{R}^d)} |P^n(x, A) - \pi_h(A)| \to 0 \quad as \quad n \to \infty.$$

We next address the question: how far is $\pi_h$ from $\pi$? This question can be typically answered by a careful inspection on the proof of Proposition 2.1. However, for (RLMC), this is not the case, and requires using a different technique. Towards that, we derive an upper bound of $W_2(\pi, \pi_h)$ under the same assumptions in the previous theorem and the additional assumption that $f$ is also strongly convex with parameter $m$.

**Proposition 2.2.** *Let the potential function satisfy Assumption 1.1, and let $\gamma_n := h \in (0, \frac{2}{m+M})$ in the (RLMC) algorithm. Then, we have*

$$W_2(\pi, \pi_h) \leq 3\sqrt{dh}\frac{(1 + 2Mh)^2}{\kappa^{-1} - Mh/\sqrt{3}}. \tag{3}$$

**Remark 1.** *The above proposition shows that the order of the bias between the stationary distribution of the Langevin diffusion and that of the (RLMC) chain is of the order $\mathcal{O}(\sqrt{h})$.*

## 2.3 Wasserstein-2 rates and CLT with Decreasing Step-size

In this part, we consider the (RLMC) algorithm with a fast decreasing time step sequence $(\gamma_n)$ and establish a convergence rate in $W_2$ distance as well as a CLT for the numerical integration (1).

**Proposition 2.3.** *Suppose $f$ satisfies Assumption 1.1. Let $x_0 := \arg\min_x f(x)$ and $\gamma_{n+1} \leq \frac{m}{m^2 + M^2(33+n)}$. After running (RLMC) algorithm for $K = \mathcal{O}\left(\kappa^{1.5}/\epsilon\right)$ steps, we obtain $W_2(\nu_K, \pi) \leq \epsilon\sqrt{d/m}$, where $\nu_K$ is the probability distribution of $x_K$.*

**Remark 2.** *There are two aspects of this result. The first aspect is rather standard; there is no logarithmic factor in $1/\epsilon$ compared to the result in Proposition 2.1. Similar phenomenon has been previously observed for the LMC algorithm [8]. The second aspect is that we never obtain the $\mathcal{O}(1/\epsilon^{2/3})$ term as in Proposition 2.1, with the constant step-size choice. This is not an artifact of our analysis. This is due to the fact that with this choice of decreasing step-size, we reduce the bias much more at the expense of slightly increased variance. However, as we demonstrate next, this choice of decreasing step-size is crucial for obtaining an unbiased CLT for numerical integration.*

As the main contribution of this section, we characterize the fluctuations of (RLMC) when it is used for computing the integral $\int_{\mathbb{R}^d} \varphi d\pi$ for a $\pi$-integrable function $\varphi$. Choosing the Langevin diffusion in (2) with the stationary distribution $\pi$, we have by Theorem 1 that it is ergodic, and $\lim_{t\to+\infty} \frac{1}{t}\int_0^t \varphi(X(s))ds = \int_{\mathbb{R}^d}\varphi d\pi := \pi(\varphi)$, almost surely. Motivated by this, we first discretize the diffusion using (RLMC) and then compute a discrete analogue of the average. The procedure consists of two successive phases:

(a) **Discretization:** The (RLMC) algorithm is run with a step size sequence $(\gamma_n)$ satisfying for all $n$, $\gamma_n > 0$, $\lim_{n\to+\infty} \gamma_n = 0$, and $\lim_{n\to+\infty} \Gamma_n = +\infty$, where $\Gamma_n := \sum_{k=1}^n \gamma_k$.

(b) **Averaging:** Using the (RLMC) iterates $(x_n)$, construct a weighted empirical measure via the same weight sequence $\gamma := (\gamma_n)$: For every $n \geq 1$ and every $\omega \in \Omega$, set

$$\pi_n^\gamma(\omega, dx) := \frac{\gamma_1\delta_{x_0(\omega)} + \cdots + \gamma_{k+1}\delta_{x_k(\omega)} + \cdots + \gamma_n\delta_{x_{n-1}(\omega)}}{\gamma_1 + \cdots + \gamma_n},$$

and use $\pi_n^\gamma(\omega, \varphi) := \int_{\mathbb{R}^d} \varphi \pi_n^\gamma(\omega, dx) = \frac{1}{\Gamma_n}\sum_{k=1}^n \gamma_k\varphi(x_{k-1}(\omega))$ to estimate the expectation (1).

For numerical purposes, for a fixed function $\varphi$, $\pi_n^\gamma(\omega, \varphi)$ can be recursively computed as follows:

$$\pi_{n+1}^\gamma(\omega, \varphi) = \pi_n^\gamma(\omega, \varphi) + \tilde{\gamma}_{n+1}\left(\varphi(x_n(\omega)) - \pi_n^\gamma(\omega, \varphi)\right) \quad \text{with } \tilde{\gamma}_{n+1} := \frac{\gamma_{n+1}}{\Gamma_{n+1}}.$$

We now provide the main result of this section, a central limit theorem for the algorithm (RLMC) when it is used to compute integrals of the form in (1).

**Theorem 2.** *Let $\pi$ be such that its potential $f$ satisfies Assumption 1.1. Consider a test function $\varphi : \mathbb{R}^d \to \mathbb{R}$ of the form $\varphi = \mathcal{A}\phi$ for some function $\phi : \mathbb{R}^d \to \mathbb{R}$, where $\mathcal{A}$ denotes the generator of the diffusion (2), i.e., $\mathcal{A}\phi := -\langle\nabla f, \nabla\phi\rangle + \Delta\phi$. Define $\hat{\gamma}_n := \frac{1}{\sqrt{\Gamma_n}}\sum_{k=1}^n \gamma_k^2$ and let $\hat{\gamma}_\infty = \lim_{n\to\infty}\hat{\gamma}_n$. Then for all $\phi \in \mathcal{C}^4(\mathbb{R}^d)$ with $D^2\phi$, $D^3\phi$ being bounded, and $D^4\phi$ being bounded and Lipschitz, and $\sup_{x\in\mathbb{R}^d}\|\nabla\phi(x)\|^2/(1+\|x\|^2) < +\infty$, we have the following central limit theorem for the numerical integration computed via (RLMC):*

(i) *If $\hat{\gamma}_\infty = 0$, then $\sqrt{\Gamma_n}\pi_n^\gamma(\varphi) \xrightarrow{d} \mathcal{N}(0, 2\int_{\mathbb{R}^d}\|\nabla\phi(x)\|^2\pi(dx))$,*

(ii) *If $\hat{\gamma}_\infty \in (0, +\infty)$, then $\sqrt{\Gamma_n}\pi_n^\gamma(\varphi) \xrightarrow{d} \mathcal{N}(\varrho\,\hat{\gamma}_\infty, 2\int_{\mathbb{R}^d}\|\nabla\phi(x)\|^2\pi(dx))$,*

(iii) *If $\hat{\gamma}_\infty = +\infty$, then $\frac{\sqrt{\Gamma_n}}{\hat{\gamma}_n}\pi_n^\gamma(\varphi) \xrightarrow{p} \varrho$,*

*where the mean $\varrho$ is given as*

$$\varrho = \int\int\langle D^3\phi(x), \nabla f(x) \otimes u \otimes u\rangle\mu(du)\pi(dx) - \frac{1}{2}\int\langle D^2 f(x), \nabla\phi(x) \otimes \nabla f(x)\rangle\pi(dx)$$
$$+ \frac{1}{2}\int\int\langle D^3 f(x), \nabla\phi(x) \otimes u \otimes u\rangle\mu(du)\pi(dx) - \frac{1}{2}\int\langle D^2\phi(x), \nabla f(x) \otimes \nabla f(x)\rangle\pi(dx)$$
$$- \frac{1}{6}\int\int\langle D^4\phi(x), u^{\otimes 4}\rangle\mu(du)\pi(dx),$$

*and $\mu$ is the distribution for a $d$-dimensional standard Gaussian measure.*

**Remark 3.** *First note that a CLT for the Euler discretization of Langevin diffusion follows from [21, Thm. 10]. The rates of the CLT established in Theorem 2 are similar to that case, with only the bias term $\rho$ being different. Specifically, following the same computation in [21], we see that the optimal*

*rate with polynomially decaying step-size choice $\gamma_k = k^{-\alpha}$, for some $\alpha > 0$, is $\mathcal{O}(n^{1/3})$. But in this case, the established CLT is biased. However, for any $0 < \alpha < 1/3$, we obtain an unbiased CLT as well. Hence, although the (RLMC) chain provides rate improvements for sampling (with respect to $W_2$ distance), as demonstrated in [40] and in Proposition 2.1, it does not seem to provide any improvements for CLT. In retrospect, this is expected as the rate improvements for sampling is achieved by the choice of constant step-size for which it is not possible to establish even a nearly unbiased CLT.*

The class of test functions that the above CLT can cover is intimately related to the solution of the *Stein equation* (or Poisson equation) $\varphi = \mathcal{A}\phi$. Given $\varphi$, there is an explicit characterization of $\phi$ that solves the Stein's equation, and various properties of $\varphi$ are translated to $\phi$ [18, 17].

# 3 Results for the Underdamped Langevin Diffusion

The underdamped Langevin diffusion is given by

$$d \begin{bmatrix} x(t) \\ v(t) \end{bmatrix} = \begin{bmatrix} v(t) \\ -(\beta v(t) + u\nabla f(x(t))) \end{bmatrix} dt + \sqrt{2\beta u} \begin{bmatrix} 0_d \\ I_d \end{bmatrix} dW(t), \qquad (4)$$

where $\beta > 0$ is the friction coefficient and $u > 0$ is the inverse mass. For simplicity, we will consider $\beta = 2$ in the later text. Under mild conditions, it is well-known that the continuous-time Markov process $(x(t), v(t))$ is positive recurrent, and its invariant distribution is given by $\nu(x, v) \propto \exp \left\{ -f(x) - \frac{1}{2u}\|v\|^2 \right\}$, $x \in \mathbb{R}^d$, $v \in \mathbb{R}^d$. This diffusion, with an additional Hamiltonian component, has gathered a lot of attention recently due to its improved convergence properties [7, 5, 40, 27, 12] and empirical performance [36, 2].

The *randomized midpoint discretization* of the underdamped Langevin diffusion (4) is given as:

$x_{n+\frac{1}{2}} = x_n + \frac{1}{2}(1 - e^{-2\alpha_{n+1}\gamma_{n+1}})v_n - \frac{u}{2}\left(\alpha_{n+1}\gamma_{n+1} - \frac{1}{2}(1 - e^{-2\alpha_{n+1}\gamma_{n+1}})\right)\nabla f(x_n) + \sqrt{u}\sigma_{n+1}^{(1)}U_{n+1}^{(1)},$

$x_{n+1} = x_n + \frac{1}{2}(1 - e^{-2\gamma_{n+1}})v_n - \frac{u}{2}\gamma_{n+1}(1 - e^{-2(1-\alpha_{n+1})\gamma_{n+1}})\nabla f(x_{n+\frac{1}{2}}) + \sqrt{u}\sigma_{n+1}^{(2)}U_{n+1}^{(2)},$

$v_{n+1} = v_n e^{-2\gamma_{n+1}} - u\gamma_{n+1}e^{-2(1-\alpha_{n+1})\gamma_{n+1}}\nabla f(x_{n+\frac{1}{2}}) + 2\sqrt{u}\sigma_{n+1}^{(3)}U_{n+1}^{(3)},$ \hfill (RULMC)

where $(\gamma_n)$ is the sequence of time steps, $\sigma_n^{(1)}$, $\sigma_n^{(2)}$ and $\sigma_n^{(3)}$ are positive with $(\sigma_n^{(1)})^2 = \alpha_n\gamma_n + \frac{1-e^{-4\alpha_n\gamma_n}}{4} - (1 - e^{-2\alpha_n\gamma_n})$, $(\sigma_n^{(2)})^2 = \gamma_n + \frac{1-e^{-4\gamma_n}}{4} - (1 - e^{-2\gamma_n})$ and $(\sigma_n^{(3)})^2 = \frac{1-e^{-4\gamma_n}}{4}$, and $(\alpha_n)$ is a sequence of identically distributed random variables following the distribution $\alpha_n \sim U[0, 1]$. $(U_n^{(1)}, U_n^{(2)}, U_n^{(3)})$ are independent centered Gaussian random vectors in $\mathbb{R}^{3d}$, also independent of $(\alpha_n)$ and initial point $(x_0, v_0)$, having the following pairwise covariances:

$$\text{cov}(\sigma_n^{(1)}U_n^{(1)}, \sigma_n^{(2)}U_n^{(2)}) = \left(\alpha_n\gamma_n - \left(e^{-\alpha_n\gamma_n} + e^{-2\gamma_n}\sinh(\alpha_n\gamma_n)\right)\sinh(\alpha_n\gamma_n)\right)I_{d\times d},$$
$$\text{cov}(\sigma_n^{(2)}U_n^{(2)}, \sigma_n^{(3)}U_n^{(3)}) = \left(e^{-2\gamma_n}\sinh(\gamma_n)^2\right)I_{d\times d},$$
$$\text{cov}(\sigma_n^{(1)}U_n^{(1)}, \sigma_n^{(3)}U_n^{(3)}) = \left(e^{-2\gamma_n}\sinh(\alpha_n\gamma_n)^2\right)I_{d\times d}.$$

The (RULMC) algorithm has emerged as an optimal sampling algorithm in the sense that it achieves the information theoretical lower bound in both tolerance $\epsilon$ and dimension $d$ for sampling from a strongly log-concave densities [1, 40]. Therefore, it is interesting to examine if (RULMC) based numerical integrator have any benefits in other MCMC-based tasks such as (1). Towards that, we characterize the order of bias with a constant step-size choice for (RULMC) iterates as proposed in [40]. Compared to the bias result in Proposition 2.2 for the (RLMC) discretization, we note that order of bias is increased (i.e. smaller bias). Next, in Theorem 4 we provide a CLT for numerical integration with (RULMC). Our results show that when it comes to computing expectations of the form in (1) using (RULMC) and characterizing its fluctuations, the (RULMC) discretization obtains rate improvements only for a class of constant test functions (as described in Remark 6).

## 3.1 Analysis of the Markov Chain generated by Constant Step-size RULMC

Recall that $\pi(x)$ is the marginal density function of $\nu(x, v)$ with respect to $x$. Similarly $\nu_h(x, v)$ be the stationary density function of the Markov chain generated by (RULMC) chain and $\pi_h(x)$

be the marginal density function of $\nu_h(x, v)$, with respect to $x$. Furthermore, the filtration $(\mathcal{F}_n)$ is defined as $\mathcal{F}_n := \sigma(\alpha_k, U_k^{(i)}; 1 \leq k \leq n, i = 1, 2, 3)$. When $f \in \mathcal{C}^2(\mathbb{R}^d)$ and is gradient Lipschitz with parameter $M$, then we can immediately see that the transition kernel of chain $(x_n, v_n)$: $P((x, v), (x', v')) \in \mathcal{C}(\mathbb{R}^{2d} \times \mathbb{R}^{2d})$ is positive everywhere. Therefore, it's easy to obtain that the chain $(x_n, v_n)$ is $\mu^{\text{Leb}}$-irreducible and aperiodic. Given all this information, we can give a sufficient condition to make sure that the chain has a unique invariant probability measure and is ergodic.

**Theorem 3.** *Let the potential function $f$ satisfy part (a) of Assumption 1.1, and let $\gamma_n := h$ be small enough. Then if $u \in (0, \frac{4}{2M-m})$, the (RULMC) Markov chain $(x_n, v_n)$ has a unique stationary probability measure $\nu_h$ and for every $(x, v) \in \mathbb{R}^{2d}$, we have*

$$\sup_{A \in \mathcal{B}(\mathbb{R}^{2d})} |P^n((x, v), A) - \nu_h(A)| \to 0 \quad as \quad n \to \infty.$$

We next derive an upper bound on the bias $W_2(\pi, \pi_h)$ of (RULMC) algorithm, under the additional strong convexity assumption on the potential function $f$.

**Proposition 3.1.** *Suppose that $f$ satisfies Assumption 1.1. If we run the (RULMC) algorithm with $u = 1/M$ and $\gamma_n := h$, for universal constants $C_1, C_2 > 0$, we have*

$$W_2^2(\pi, \pi_h) \leq \frac{C_1 h^3 (\kappa h^3 + 1)d}{1 - \frac{h}{4\kappa} - C_2 h^3 \kappa(1 + \kappa h^3)}.$$

**Remark 4.** *Note that we have $W_2(\pi, \pi_h) \to 0$ as $h \to 0$. Furthermore, as $h \to 0$, $W_2(\pi, \pi_h) < \mathcal{O}(h^{\frac{3}{2}})$. Hence, the bias order is increased for the underdamped Langevin diffusion compared to the overdamped case (cf. Proposition 2.2), providing a smaller bias for the same step-size.*

## 3.2 Wasserstein-2 rates and CLT with Decreasing Step-size

We now provide the rate of convergence in Wasserstein-2 metric with decreasing step-size for (RULMC). The specific choice for the decreasing step-size that we consider below, also is satisfied for our CLT result in Remark 6.

**Proposition 3.2.** *Suppose $f$ satisfies Assumption 1.1. Fix $u = 1/M$. Let $x_0 := \arg\min_x f(x)$ and choose $\gamma_n = \frac{16\kappa}{32\kappa^{\frac{5}{3}} + (n - K_1)^+}$, for a $K_1 \in (0, \infty)$ (where $(a)^+ := \max(0, a)$). After running (RULMC) for $K = \tilde{\mathcal{O}}\left(\kappa^{3/2}/\epsilon^{2/3}\right)$ steps, we obtain $W_2(\nu_K, \pi) \leq \epsilon\sqrt{d/m}$, where $\nu_K$ is the probability distribution of $x_K$.*

**Remark 5.** *Similar to the result in Proposition 2.3, there are two aspects of this result. The first aspect is again removing the logarithmic factor in $1/\epsilon$ compared to the result in Theorem 3 in [40], which is quite standard in the literature. The second aspect is that we never obtain the $\mathcal{O}(1/\epsilon^{1/3})$ part, as in Theorem 3 in [40] with the constant step-size choice.*

Similar to the previous case, we now describe the numerical integration procedure using the (RULMC) discretization. We denote the $n$-th iterate as $(x_n, v_n)$. The time-step we use is $(\gamma_n)$ such that $\forall n \in \mathbb{N}^*, \gamma_n \geq 0, \lim_n \gamma_n = 0$ and $\lim_n \Gamma_n^{(1)} = +\infty$, where $\Gamma_n^{(\ell)} := \sum_{i=1}^n \gamma_i^\ell$. Our averaging is a weighted empirical measure with $Y_n = (x_n, v_n)$ using the step size sequence $\gamma := (\gamma_n)$ as the weights. Let $\delta_x$ denote the Dirac mass at $x$. Then for every $n \geq 1$, set

$$\nu_n^\gamma(\omega, dx) := \frac{\gamma_1 \delta_{Y_0(\omega)} + \cdots + \gamma_{k+1} \delta_{Y_k(\omega)} + \cdots + \gamma_n \delta_{Y_{n-1}(\omega)}}{\gamma_1 + \cdots + \gamma_n}$$

and we can use $\nu_n^\gamma(\omega, \varphi)$ to approximate $\nu(\varphi) = \mathbb{E}_\nu[\varphi'(Y)]$, where $\varphi' : \mathbb{R}^{2d} \to \mathbb{R}$.

If we assume $g : \mathbb{R}^{2d} \to \mathbb{R}$ such that $\mathcal{L}g = \varphi'$, we can establish the following theorem, in which we state only the unbiased CLT result for simplicity.

**Theorem 4.** *Let $\pi$ be such that its potential function $f$ satisfies Assumption 1.1. Assume $u \in (0, \frac{4}{2M-m})$. Consider a test function $\varphi' = \mathcal{L}g$, for some function $g : \mathbb{R}^{2d} \to \mathbb{R}$, where $\mathcal{L} = 2u\Delta_v - 2\langle v, \nabla_v \rangle - u\langle \nabla f(x), \nabla_v \rangle + \langle v, \nabla_x \rangle$ denotes the generator of the diffusion (4). Suppose the step-size $(\gamma_k)$ is non-increasing, $\lim_{n \to +\infty}(1/\sqrt{\Gamma_n}) \sum_{k=1}^n \gamma_k^{3/2} = +\infty$. Then, if $\lim_{n \to +\infty}(1/\sqrt{\Gamma_n}) \sum_{k=1}^n \gamma_k^2 = 0$, for every $g \in \mathcal{C}^4(\mathbb{R}^{2d})$ function with $D^2g$ bounded, $D^3g$*

*bounded and Lipschitz and $\sup_{(x,v)\in\mathbb{R}^{2d}} \|\nabla g(x,v)\|/(1+\|x\|^2+\|v\|^2) < +\infty$, we have the following central limit theorem for the numerical integration computed using the* (RULMC) *iterates:*

$$\sqrt{\Gamma_n}\nu_n^\gamma(\mathcal{L}g) \xrightarrow{d} \mathcal{N}\big(0, 4u\int \|\nabla_v g(x,v)\|^2 \nu(dx,dv)\big).$$

The rate of convergence of the CLT in Theorem 4 follows exactly the same behavior in Theorem 2. Hence, for the class of general test functions, Theorem 4 does not exhibit a rate improvement. Towards that, we make the following remarks under a carefully constructed condition for the class of test functions.

**Remark 6.** *Let $\pi$ be such that its potential function $f$ satisfies Assumption 1.1. Assume $u \in (0, \frac{4}{2M-m})$. Consider a test function $\varphi = \mathcal{L}g$ which could be written as $\mathcal{L}g(v,\phi(x)) = \langle v, \nabla\phi(x)\rangle$, for some function $\phi : \mathbb{R}^d \to \mathbb{R}$, where $\mathcal{L} = 2u\Delta_v - 2\langle v, \nabla_v\rangle - u\langle\nabla f(x), \nabla_v\rangle + \langle v, \nabla_x\rangle$ denotes the generator of the diffusion (4). Suppose the time step-size $(\gamma_k)$ is non-increasing, and satisfies $\lim_{n\to\infty}(\gamma_{n-1}-\gamma_n)/\gamma_n^4 = 0$ and $\lim_{n\to\infty}\Gamma_n^{(4)} = +\infty$. Define $\hat{\gamma}_n := \Gamma_n^{(4)}/\sqrt{\Gamma_n^{(3)}}$ and let $\hat{\gamma}_\infty = \lim_{n\to\infty}\hat{\gamma}_n$. Then, for all $\phi \in \mathcal{C}^4(\mathbb{R}^d)$ with $D^2\phi$, $D^3\phi$ and $D^4\phi$ bounded and Lipschitz and $\sup_{(x,v)\in\mathbb{R}^{2d}} \|\nabla\phi(x)\|^2/(1+\|x\|^2+\|v\|^2) < +\infty$, we obtain the following central limit theorem for numerical integration computed using the* (RULMC) *algorithm:*

*(i) If $\hat{\gamma}_\infty = 0$, we have $\frac{\Gamma_n}{\sqrt{\Gamma_n^{(3)}}}\nu_n^\gamma(\mathcal{L}\phi) \xrightarrow{d} \mathcal{N}(0, \frac{10}{3}u\int_{\mathbb{R}^d}\|\nabla\phi(x)\|\pi(dx))$,*

*(ii) If $\hat{\gamma}_\infty \in (0, +\infty)$, we have $\frac{\Gamma_n}{\Gamma_n^{(4)}}\nu_n^\gamma(\mathcal{L}\phi) \xrightarrow{d} \mathcal{N}(\rho, \frac{10}{3}u\hat{\gamma}_\infty^{-2}\int_{\mathbb{R}^d}\|\nabla\phi(x)\|\pi(dx))$,*

*(iii) If $\hat{\gamma}_\infty = +\infty$, we have $\frac{\Gamma_n}{\Gamma_n^{(4)}}\nu_n^\gamma(\mathcal{L}\phi) \xrightarrow{p} \rho$,*

*where,*

$$\rho = \tfrac{5u}{12}\int\int\langle D^3\phi(x), \nabla f(x)\otimes v\otimes v\rangle\nu(dx,dv) + \tfrac{u}{24}\int\int\langle D^3 f(x), \nabla\phi(x)\otimes v\otimes v\rangle\nu(dx,dv)$$
$$+ \tfrac{7u}{12}\int\int(D^2\phi D^2 f)(x)v^{\otimes 2}\nu(dx,dv) - \tfrac{u^2}{4}\int\langle D^2\phi(x), \nabla f(x)^{\otimes 2}\rangle\pi(dx)$$
$$- \tfrac{u^2}{24}\int\langle D^2 f(x), \nabla\phi(x)\otimes\nabla f(x)\rangle\pi(dx).$$

**Remark 7.** *For polynomial time steps $\gamma_k := k^{-\alpha}$, since we require that $\Gamma_n^{(4)} \to +\infty$ as $n \to +\infty$, we need $0 < \alpha \le \frac{1}{4}$. Using L'Hospitals rule, it is straightforward to check that the condition $\lim_{n\to+\infty}\frac{\gamma_{n-1}-\gamma_n}{\gamma_n^4} = 0$ is satisfied when $\alpha \in (0, \frac{1}{4}]$. We then have the following order estimates:*

$$\Gamma_n \sim \frac{n^{1-\alpha}}{1-\alpha}, \quad \sqrt{\Gamma_n^{(3)}} \sim \frac{n^{\frac{1}{2}-\frac{3}{2}\alpha}}{\sqrt{1-3\alpha}}, \quad \Gamma_n^{(4)} \sim \begin{cases} \dfrac{n^{1-4\alpha}}{1-4\alpha}, & \text{if } \alpha \in (0, \frac{1}{4}), \\ \sqrt{\ln n}, & \text{if } \alpha = \frac{1}{4}. \end{cases}$$

*Hence, as $n \to +\infty$,*

$$\frac{\Gamma_n^{(4)}}{\sqrt{\Gamma_n^{(3)}}} \to \hat{\gamma}_\infty = \begin{cases} 0 & \text{if } \alpha \in (\frac{1}{5}, \frac{1}{4}], \\ \sqrt{10} & \text{if } \alpha = \frac{1}{5}, \\ +\infty & \text{if } \alpha \in (0, \frac{1}{5}). \end{cases}$$

*If $\alpha \in (\frac{1}{5}, \frac{1}{4}]$, the unbiased CLT holds at rate $\Gamma_n/\sqrt{\Gamma_n^{(3)}} = \mathcal{O}(n^{\frac{1}{2}(1+\alpha)}) \le \mathcal{O}(n^{\frac{5}{8}})$. The optimal rate is achieved when $\alpha = \frac{1}{4}$. If $\alpha = \frac{1}{5}$, the biased CLT holds at rate $\Gamma_n/\sqrt{\Gamma_n^{(3)}} = \mathcal{O}(n^{3\alpha}) = \mathcal{O}(n^{\frac{3}{5}})$. If $\alpha \in (0, \frac{1}{5})$, the rate of the convergence in probability is $\Gamma_n/\sqrt{\Gamma_n^{(3)}} = \mathcal{O}(n^{3\alpha}) < \mathcal{O}(n^{\frac{3}{5}})$. Therefore the optimal convergence rate $\mathcal{O}(n^{\frac{5}{8}})$ is obtained when an unbiased CLT holds. While the rate of this CLT is better than that of Theorem 2, the test functions that satisfy this condition is severely restricted.*

# 4 Discussion

In this work, we present several probablistic properties of the randomized midpoint discretization technique, focussing our attention on the overdamped and underdamped Langevin diffusions. Our results could be summarized as follows: To obtain optimal rates for sampling (in $W_2$ distance), one needs to have a constant choice of step-size. With such a constant step-size choice, the Markov chain generated by the discretization process is biased. This suggests that a decreasing step-size choice is required when using the randomized midpoint method for numerical integration. For several decreasing choices of step-sizes, we establish CLTs and highlight the relative merits and disadvantages of using randomized midpoint technique for numerical integration. In particular, our results have interesting consequence for computing confidence interval for numerical integration.

## Broader Impact

The paper predominantly concerns about theoretical results on numerical integration with MCMC based samplers. The presented results are of interest to researchers in machine learning community concerned with constructing confidence intervals for their numerical integration problems. Although not the main focus of this paper, the results might have positive consequence for building robust machine learning systems based on the obtained confidence intervals.

## Funding Disclosure

The research of YH and KB were supported in parts by UC Davis CeDAR (Center for Data Science and Artificial Intelligence Research) Innovative Data Science Seed Funding Program. The research of YH was also supported in parts by the NSF Grant CCF-1934568. The research of MAE is supported by NSERC Grant [2019-06167], Connaught New Researcher Award, CIFAR AI Chairs program, and CIFAR AI Catalyst grant.

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
