[Supplementary Material]

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

# 5 Additional Notations

We also use the following notations for the proofs. Due to the ease of presentation, whenever it is clear in the proof, we refer to the inner product between two compatible vectors $\langle a, b \rangle$ simply by $a \cdot b$. For any random variable $X$, $\|X\|_{L^2} := \mathbb{E}[\|X\|]$ where the expectation is taken over all randomness of $X$.

# 6 Proofs for Section 2

We now define the following condition, which is a consequence of Assumption 1.1

**Assumption 6.1.** *There exists a twice differentiable function* $V : \mathbb{R}^d \to [1, \infty)$ *such that:*
*(i)* $\lim_{\|x\| \to \infty} V(x) = +\infty$, *(ii) there exists* $\alpha > 0$ *and* $\beta > 0$: $\langle \nabla V(x), \nabla f(x) \rangle \geq \alpha V(x) - \beta$
*for every* $x$, *(iii) there exists* $c_V > 0$: $\|\nabla V(x)\|^2 + \|\nabla f(x)\|^2 \leq c_V V(x)$ *for every* $x$, *and (iv)*
$\left\| D^2 V \right\|_\infty := \sup_{x \in \mathbb{R}^d} \|D^2 V\|_{op} < \infty$ *(where* $\| \cdot \|_{op}$ *denotes the operator norm).*

**Lemma 1.** *Assumption 1.1 implies Assumption 6.1.*

*Proof.* Since $f \in \mathcal{C}^2(\mathbb{R}^d)$ is strongly convex, $\lim_{|x| \to +\infty} f(x) = +\infty$ and $f$ has a unique global minimizer $x^* \in \mathbb{R}^d$. It's easy to observe that $\nabla f(x^*) = 0$. We consider our $V(x) = f(x) - f(x^*) + 1$. Then it's easy to see $(i)$ is satisfied. $(iv)$ is also satisfied because $f$ is gradient Lipschitz. $(iii)$ is equivalent to that there exists a $C > 0$ such that

$$\frac{|\nabla f(x)|^2}{f(x) - f(x^*) + 1} \leq C \quad \text{for } \forall x \in \mathbb{R}^d$$

We Taylor expand the numerator and denominator:

$$|\nabla f(x)|^2 = \sum_{i=1}^d \left( f_i(x^*) + \nabla f_i(\xi)^T (x - x^*) \right)^2$$

$$\leq \sum_{i,j=1}^d |f_{ij}(\xi)|^2 |x - x^*|^2 = \left\| D^2 f(\xi) \right\|_F^2 |x - x^*|^2$$

$$\leq d^2 M^2 |x - x^*|^2$$

$$f(x) - f(x^*) + 1 = \nabla f(x^*)^T (x - x^*) + \frac{1}{2} D^2 f(\xi)(x - x^*)^{\otimes 2} + 1$$

$$= \frac{1}{2} D^2 f(\xi)(x - x^*)^{\otimes 2} + 1$$

$$\geq \frac{m}{2} |x - x^*|^2$$

Then

$$\frac{|\nabla f(x)|^2}{f(x) - f(x^*) + 1} \leq \frac{2d^2 M^2}{m} \quad \text{for } \forall x \in \mathbb{R}^d$$

$(ii)$ is equivalent to that there exists $\alpha, \beta > 0$ such that

$$|\nabla f(x)|^2 \geq \alpha(f(x) - f(x^*) + 1) - \beta \quad \text{for } \forall x \in \mathbb{R}^d$$

According to the strongly convexity of $f$, we have

$$f(x^*) - f(x) \geq \nabla f(x)^T (x^* - x) + \frac{m}{2} |x^* - x|^2$$

$$= \frac{m}{2} |x^* - x + \frac{1}{m} \nabla f(x)|^2 - \frac{1}{2m} |\nabla f(x)|^2$$

$$\iff |\nabla f(x)|^2 \geq 2m \left( f(x) - f(x^*) + 1 \right) - 2m \quad \text{for } \forall x \in \mathbb{R}^d$$

$(ii)$ is satisfied by choosing $\alpha = \beta = 2m > 0$. $\qquad \square$

**Remark 8.** *For the* $V(x)$ *we choose in the proof, under assumption 1.1, we can verify that:* $V(x) = O(|x|^2)$ *when* $|x| \to +\infty$. *We will use this fact later in the proof when we establish the CLT statement.*

## 6.1 Proofs for section 2.1

**Lemma 2.** *Let $x(t)$ be the solution to Langevin dynamics SDE with initial condition $x_0$ and $y(t)$ be the solution to Langevin dynamics SDE with initial condition $y_0$. Then we have the following estimates for Langevin dynamics when $f$ satisfies Assumption 1.1 and $Mh < \frac{1}{2}$:*

$$\mathbb{E}[\sup_{t \in [0,h]} \|\nabla f(x(t))\|^2] \leq 4 \|\nabla f(x_0)\|^2 + 8M^2 dh$$

$$\mathbb{E}[\sup_{t \in [0,h]} \|x(t) - x_0\|^2] \leq O(h^2 \|\nabla f(x_0)\|^2 + M^2 h^3 d + 2dh)$$

$$\mathbb{E}[\|x(t) - y(t)\|^2] \leq e^{-2mt} \|x_0 - y_0\|^2$$

*Proof.* By triangle inequality we have

$$\mathbb{E}[\sup_{t \in [0,h]} \|\nabla f(x(t))\|^2] \leq 2 \|\nabla f(x_0))\|^2 + 2M^2 \mathbb{E}[\sup_{t \in [0,h]} \|x(t) - x_0\|^2]$$

Furthermore, we have

$$\mathbb{E}[\sup_{t \in [0,h]} \|x(t) - x_0\|^2] = \mathbb{E}[\sup_{t \in [0,h]} \left\| -\int_0^t \nabla f(x(s))ds + \sqrt{2}W_t \right\|^2]$$

$$\leq h^2 \mathbb{E}[\sup_{t \in [0,h]} \|\nabla f(x(t))\|^2] + 2dh$$

Combining the two inequalities and $Mh < \frac{1}{2}$, we can obtain the first two estimates. The last estimate could be easily obtained by energy method. $\square$

*Proof of Theorem 2.1.* We denote $x_n = x_n(0)$ to be the algorithm iterate points, $y_n$ to be the $n$-th step of Langevin diffusion with $y_0 \sim \exp(-f(y))$, $x_{n+1}^* = x_n(h)$ to be one step solution of Langevin dynamics with initial values $x_n$. When $Mh < \frac{1}{2}$, apply lemma 2 and we get:

$$\mathbb{E}[\sup_{t \in [0,h]} \|x_{n-1}(\alpha_n h) - x_{n-1}(t)\|^2] \leq O(h^2 \|\nabla f(x_{n-1})\|_{L^2}^2 + M^2 h^3 d + 2dh)$$

$$\mathbb{E}[\left\| \nabla f(x_{n-\frac{1}{2}}) - \nabla f(x_{n-1}(\alpha_n h)) \right\|^2] \leq M^2 \mathbb{E} \left\| \int_0^{\alpha_n h} \nabla f(x_{n-1}(s)) - \nabla f(x_{n-1}(0))ds \right\|^2$$

$$\leq M4h^2 \mathbb{E}[\alpha_n^2 \sup_{t \in [0,\alpha_n h]} \|x_{n-1}(t) - x_{n-1}(0)\|^2]$$

$$\leq O(M^4 h^4 \|\nabla f(x_{n-1})\|_{L^2}^2 + dM^4 h^3 + dM^6 h^5)$$

Consider the distance between our iterates and the continuous process:

$$\mathbb{E}_{\alpha_K}[\|x_K - y_K\|^2] = \mathbb{E}_{\alpha_K}[\|x_K - x_K^* + x_K^* - y_K\|^2]$$

$$\leq \|y_K - x_K^*\|^2 + \mathbb{E}_{\alpha_K}[\|x_K - x_K^*\|^2] - 2(y_K - x_K^*)^T (\mathbb{E}_{\alpha_K} x_K - x_K^*)$$

$$\leq (1 + hm) \|y_K - x_K^*\|^2 + \frac{1}{hm} \|\mathbb{E}_{\alpha_K} x_K - x_K^*\|^2 + \mathbb{E}_{\alpha_K}[\|x_K - x_K^*\|^2]$$

Taking expectations over $\{\alpha_k, U_l, U_l'; 1 \leq k \leq K - 1, 1 \leq l \leq K\}$, applying lemma 2 again and using induction, we have

$$\|x_K - y_K\|_{L^2}^2 \leq (1 + hm) \|y_K - x_K^*\|_{L^2}^2 + \frac{1}{hm} \mathbb{E} \|\mathbb{E}_{\alpha_K} x_K - x_K^*\|^2 + \|x_K - x_K^*\|_{L^2}^2$$

$$\leq (1 + hm)e^{-2mh} \|x_{K-1} - y_{K-1}\|_{L^2}^2 + \frac{1}{hm} \mathbb{E} \|\mathbb{E}_{\alpha_K} x_K - x_K^*\|^2 + \|x_K - x_K^*\|_{L^2}^2$$

$$\leq (1 + hm)e^{-2mKh} \|x_0 - y_0\|_{L^2}^2 + \sum_{n=1}^K \frac{1}{hm} \mathbb{E} \|\mathbb{E}_{\alpha_n} x_n - x_n^*\|^2 + \sum_{n=1}^K \|x_n - x_n^*\|_{L^2}^2$$

$$\leq e^{-mKh} \|x_0 - y_0\|_{L^2}^2 + A + B$$

Next we bound part A and part B. For part A:

$$\|\mathbb{E}_{\alpha_n} x_n - x_n^*\|^2 = \left\|\mathbb{E}_{\alpha_n}[h\nabla f(x_{n-\frac{1}{2}})] - \int_0^h \nabla f(x_{n-1}(s))ds\right\|^2$$

$$\leq \mathbb{E}\left\|h\nabla f(x_{n-1}) - h\nabla f(x_{n-1}(\alpha_n h))\right\|^2 + 2\left\|\mathbb{E}_{\alpha_n}[h\nabla f(x_{n-1}(\alpha_n h))] - \int_0^h \nabla f(x_{n-1}(s))ds\right\|^2$$

$$\leq 2h^2\mathbb{E}_{\alpha_n}\left\|\nabla f(x_{n-\frac{1}{2}}) - \nabla f(x_{n-1}(\alpha_n h))\right\|^2 + 0$$

Therefore

$$\mathbb{E}\left\|\mathbb{E}_{\alpha_n} x_n - x_n^*\right\|^2 \leq 2h^2\mathbb{E}[\left\|\nabla f(x_{n-\frac{1}{2}}) - \nabla f(x_{n-1}(\alpha_n h))\right\|^2]$$

$$\leq O(M^4 h^6 \|\nabla f(x_{n-1})\|_{L^2}^2 + dM^4 h^5)$$

For part B, use our previous estimates:

$$\|x_n - x_n^*\|_{L^2}^2 = \left\|h\nabla f(x_{n-\frac{1}{2}}) - \int_0^h \nabla f(x_{n-1}(s))ds\right\|_{L^2}^2$$

$$\leq 2\left\|h\nabla f(x_{n-\frac{1}{2}}) - h\nabla f(x_{n-1}(\alpha_n h))\right\|_{L^2}^2 + 2\left\|\int_0^h \nabla f(x_{n-1}(s)) - \nabla f(x_{n-1}(\alpha_n h))ds\right\|_{L^2}^2$$

$$\leq 2h^2\left\|\nabla f(x_{n-\frac{1}{2}}) - \nabla f(x_{n-1}(\alpha_n h))\right\|_{L^2}^2 + 2M^2 h^2\mathbb{E}[\sup_{t\in[0,h]}\|x_{n-1}(\alpha_n h) - x_{n-1}(t)\|^2]$$

$$\leq O(M^2 h^4 \|\nabla f(x_{n-1})\|_{L^2}^2 + dM^2 h^3)$$

Plug the estimates on A and B into the inequality we have

$$\|x_K - y_K\|_{L^2}^2 \leq e^{-mKh}\|x_0 - y_0\|_{L^2}^2 + O(m^{-1}M^4 h^5 \sum_{n=0}^{K-1}\|\nabla f(x_n)\|_{L^2}^2 + dm^{-1}M^4 Kh^4)$$

$$+ O(M^2 h^4 \sum_{n=0}^{K-1}\|\nabla f(x_n)\|_{L^2}^2 + dM^2 Kh^3)$$

Next we need to estimate $\sum_{n=0}^{K-1}\|\nabla f(x_n)\|_{L^2}^2$. Since

$$f(x_n(h)) = f(x_n(0)) + \int_0^h df(x_n(t))$$

$$= f(x_n(0)) - \int_0^h |\nabla f(x_n(t))|^2 dt + \sqrt{2}W_h + \int_0^h \Delta f(x_n(t))dt$$

we have

$$\mathbb{E}[f(x_{n+1}(0))] - \mathbb{E}[f(x_n(h))] = \mathbb{E}[f(x_{n+1}(0)) - f(x_n(0))] + \mathbb{E}[\int_0^h |\nabla f(x_n(t))|^2 dt] - \mathbb{E}[\int_0^t \Delta f(x_n(t))dt]$$

When $Mh < \frac{1}{4}$,

$$\mathbb{E}[\inf_{t\in[0,h]}\|\nabla f(x(t))\|^2] \geq \frac{1}{2}\|\nabla f(x(0))\|_{L^2}^2 - \mathbb{E}[\sup_{t\in[0,h]}\|\nabla f(x(t)) - \nabla f(x(0))\|^2]$$

$$\geq \frac{1}{2}\|\nabla f(x(0))\|_{L^2}^2 - M^2\mathbb{E}[\sup_{t\in[0,h]}\|x(t) - x(0)\|^2]$$

$$\geq \frac{1}{4}\|\nabla f(x(0))\|_{L^2}^2 + O(dM^2 h)$$

$$|\Delta f(x_n(t))| \leq d\left\|\nabla^2 f(x_n(t))\right\| \leq Md$$

Plug these two estimates into our previous identity and we obtain,

$$\mathbb{E}[f(x_{n+1}(0)) - f(x_n(h))] \geq \mathbb{E}[f(x_{n+1}) - f(x_n)] + \frac{h}{4}\|\nabla f(x_n)\|_{L^2}^2 - dMh + O(dM^2h^2)$$

Next we consider that

$$\mathbb{E}_{\alpha_{n+1}}[f(x_{n+1}(0))] \leq f(x_n(h)) + \nabla f(x_n(h))^T(\mathbb{E}_{\alpha_{n+1}}[x_{n+1}(0)] - x_n(h)) + \frac{M}{2}\mathbb{E}_{\alpha_{n+1}}[\|x_{n+1}(0) - x_n(h)\|^2]$$

$$\leq f(x_n(h)) + Mh^2\|\nabla f(x_n(h))\|_{L^2}^2 + M^{-1}h^{-2}\left\|\mathbb{E}_{\alpha_{n+1}}[x_{n+1}] - x_n(h)\right\|^2$$

$$+ \frac{M}{2}\mathbb{E}_{\alpha_{n+1}}[\|x_{n+1}(0) - x_n(h)\|^2]$$

where

$$Mh^2\mathbb{E}[\|\nabla f(x_n(h))\|^2] \leq O(Mh^4\|\nabla f(x_n)\|_{L^2}^2 + dMh^3)$$

$$M^{-1}h^{-2}\left\|\mathbb{E}_{\alpha_{n+1}}[x_{n+1}] - x_n(h)\right\|^2 \leq O(M^3h^4\|\nabla f(x_n)\|_{L^2}^2 + dM^3h^3)$$

$$\frac{M}{2}\mathbb{E}[\|x_{n+1} - x_n(h)\|^2] \leq O(M^3h^4\|\nabla f(x_n)\|_{L^2}^2 + dM^3h^3)$$

Hence we have

$$\mathbb{E}[f(x_{n+1}(0)) - f(x_n(h))] \leq O(M^3h^4\|\nabla f(x_n)\|_{L^2}^2 + dM^3h^3)$$

and

$$O(M^3h^4\|\nabla f(x_n)\|_{L^2}^2 + dM^3h^3) \geq \mathbb{E}[f(x_{n+1}) - f(x_n)] + \frac{h}{4}\|\nabla f(x_n)\|_{L^2}^2 + O(dM^2h^2) - dMh$$

sum up over $k$ from 0 to $K-1$:

$$O(M^3h^4\sum_{k=0}^{K-1}\|\nabla f(x_n)\|_{L^2}^2 + dM^3Kh^3) \geq \mathbb{E}[f(x_K) - f(x_0)] + \frac{h}{4}\sum_{k=1}^{K-1}\|\nabla f(x_n)\|_{L^2}^2 + O(dM^2Nh^2) - dMKh$$

Picking $x_0 = \arg\min f(x)$, we can ensure $\mathbb{E}[f(x_n) - f(x_0)] \geq 0$, when $Mh < \frac{1}{2}$, we have

$$\frac{h}{8}\sum_{k=0}^{K-1}\|\nabla f(x_n)\|_{L^2}^2 \leq dKMh - O(dKM^2h^2) + O(dKM^3h^3)$$

$$\implies \sum_{k=0}^{K-1}\|\nabla f(x_n)\|_{L^2}^2 \leq O(dKM)$$

Therefore

$$\|x_K - y_K\|_{L^2}^2 \leq e^{-mKh}\|x_0 - y_0\|_{L^2}^2 + O(m^{-1}M^5h^5Kd + m^{-1}M^4h^4Kd) + O(M^3h^4Kd + M^2h^3Kd)$$

$$\leq e^{-mKh}\|x_0 - y_0\|_{L^2}^2 + O(\kappa M^3h^4Kd) + O(M^2h^3Kd)$$

Therefore we have

$$W_2(\nu_K, \pi)^2 \leq e^{-mKh}\|x_0 - y_0\|_{L^2}^2 + O(M^3h^4Kd)\max\{\kappa, \frac{1}{Mh}\}$$

a) When $\kappa > \frac{1}{Mh}$, by choosing $h \sim O(\frac{\epsilon^{2/3}}{\kappa^{1/3}M})$, we can ensure $W_2(\nu_K, \pi)^2 \leq \epsilon^2 d/m$ after $K$ steps when $K \sim \tilde{O}(\frac{\kappa^{4/3}}{\epsilon^{2/3}})$.

b) When $\kappa \leq \frac{1}{Mh}$, by choosing $h \sim O(\frac{\epsilon}{M})$, we can ensure $W_2(\nu_K, \pi)^2 \leq \epsilon^2 d/m$ after $K$ steps when $K \sim \tilde{O}(\frac{\kappa}{\epsilon})$.

$\square$

## 6.2 Proofs for Section 2.2

*Proof of Theorem 1.* Under the assumption 6.1, we can show that the following Lyapunov condition is satisfied for small $h$.

**(Lyapunov Condition):** There exists a function $V : \mathbb{R}^d \to [1, \infty)$ such that:

0) $\lim_{|x| \to \infty} V(x) = +\infty$,

1) There exists $\hat{\alpha} \in (0, 1)$ and $\hat{\beta} \geq 0$: $\mathbb{E}[V(x_{n+1})|\mathcal{F}_n] \leq \hat{\alpha}V(x_n) + \hat{\beta}$.

**Proof:** To show that assumption 6.1 implies Lyapunov condition, we first do Taylor expansion of $V(x_{n+1})$ at $x_n$:

$$V(x_{n+1}) = V(x_n) - h\langle \nabla V(x_n), \nabla f(x_n)\rangle + \alpha_{n+1}h^2\langle D^2 f(x_n); \nabla f(x_n), \nabla V(x_n)\rangle$$
$$- \sqrt{2\alpha_{n+1}}h^{\frac{3}{2}}\langle D^2 f(x_n); \nabla V(x_n), U'_{n+1}\rangle + \sqrt{2h}\nabla V(x_n) \cdot U_{n+1}$$
$$+ \frac{1}{2}D^2 V(\theta_n)(-h\nabla f(x_n) + \alpha_{n+1}h^2 D^2 f(x_n)\nabla f(x_n) - \sqrt{2\alpha_{n+1}}h^{\frac{3}{2}}U'_{n+1} + \sqrt{2h}U_{n+1})^{\otimes 2}$$

where $\theta_n$ is a random point on the line segment joining $x_n$ and $x_{n+1}$. Using the fact that $f$ is $M$-gradient Lipschitz, we have:

$$\mathbb{E}[V(x_{n+1})|\mathcal{F}_n] \leq V(x_n) - h\langle \nabla V(x_n), \nabla f(x_n)\rangle + \frac{1}{4}Mh^2(|\nabla f(x_n)|^2 + |\nabla V(x_n)|^2) + h^{\frac{3}{2}}Md + h^{\frac{3}{2}}M|\nabla V(x_n)|^2$$
$$+ 2\left\|D^2 V\right\|_\infty (h^2|\nabla f(x_n)|^2 + \frac{1}{3}M^2 h^4|\nabla f(x_n)|^2 + h^3 d + 2hd)$$
$$\leq (1 - \alpha h + \frac{1}{4}Mh^2 c_V + Mh^{\frac{3}{2}}c_V + 2\left\|D^2 V\right\|_\infty h^2 c_V + \frac{2}{3}c_V\left\|D^2 V\right\|_\infty M^2 h^4 c_V)V(x_n)$$
$$+ \beta h + dMh^{\frac{3}{2}} + 2d\left\|D^2 V\right\|_\infty h^3 + 4d\left\|D^2 V\right\|_\infty h$$
$$\leq \hat{\alpha}V(x_n) + \hat{\beta}$$

for some $\hat{\alpha} \in (0, 1)$ and $\hat{\beta} \geq 0$ when $h$ is small. $\qquad\square$

Once we have the Lyapunov condition, we can define the stopping time $\tau_C = \inf\{n > 0 : x_n \in C\}$ and show that $\sup_{x \in C}\mathbb{E}_x[\tau_C] \leq M_C < \infty$ for all small set C. Then uniqueness of stationary probability measure and ergodicity all follow by Theorem 1.3.1 in [33]. Next we prove that $\sup_{x \in C}\mathbb{E}_x[\tau_C] \leq M_C < \infty$ given Lyapunov condition. To do so, note that we have

$$\mathbb{E}_x[\tau_C] = \sum_{k=1}^{\infty} k\mathbb{P}(\tau_C = k)$$
$$= \sum_{k \geq 1}\mathbb{P}(\tau_C > k - 1)$$

Under Lyapunov condition, for any stopping time $N$, according to Lemma A.3 and Corollary A.4 in [31], we have

$$\mathbb{P}(\tau_C > k - 1) \leq \mathbb{E}[V(x_n)1_{\tau_C > k-1}]$$
$$\leq \frac{\kappa[\gamma^{k-1}V(x_0) + 1]}{1 - \gamma}$$
$$\leq \kappa\gamma^{n-1}[V(x_0) + 1]$$

for some $\gamma \in (\hat{\alpha}, 1)$ and constant $\kappa$. Therefore we have

$$\mathbb{E}_x[\tau_C] \leq \sum_{k \geq 1}\kappa\gamma^{n-1}[V(x_0) + 1]$$
$$= \frac{\kappa[V(x) + 1]}{1 - \gamma}$$

and

$$\sup_{x \in C}\mathbb{E}_x[\tau_C] \leq \frac{\kappa}{1 - \gamma}\sup_{x \in C}V(x) + \frac{\kappa}{1 - \gamma} \leq M_C < \infty$$

So as a conclusion, the statement of the theorem follows. $\qquad\square$

*Proof of Proposition 2.2.* Consider that $x_n \sim \pi_h$ and $x_n^* \sim \pi$ are two independent random variables. Define $x_{n+1}$ to be the one step RLMC result starting from $x_n$ and $x_n^*(h)$ to be the solution of Langevin dynamics with initial value $x_n^*$. Therefore, $x_{n+1} \sim \pi_h$ and $x_n^*(h) \sim \pi$ are also independent and $\|x_n^* - x_n\|_{L^2} = \|x_n^*(h) - x_{n+1}\|_{L^2}$. We can compute the diffenrence between $x_{n+1}$ and $x_n^*(h)$:

$$x_n^*(h) - x_{n+1} = (x_n^* - x_n) - \int_0^h \nabla f(x_n^*(s))ds + h\nabla f(x_n^*(\alpha_{n+1}h)) - h(-\nabla f(x_{n+\frac{1}{2}}) + \nabla f(x_n^*(\alpha_{n+1}h)))$$

It's easy to see that $\mathbb{E}_{\alpha_{n+1}}[\int_0^h \nabla f(x_n^*(s))ds - h\nabla f(x_n^*(\alpha_{n+1}h))] = 0$. And we can rewrite the last term as

$$
\begin{aligned}
h(-\nabla f(x_{n+\frac{1}{2}}) + \nabla f(x_n^*(\alpha_{n+1}h))) = & \; h(\nabla f(x_{n+\frac{1}{2}} + x_n^* - x_n) - \nabla f(x_{n+\frac{1}{2}})) \\
& + h\nabla f(x_n^* - \int_0^{\alpha_{n+1}h} \nabla f(x_n^*(s))ds + \sqrt{2}W_{\alpha_{n+1}h}) \\
& - h\nabla f(x_n^* - \alpha_{n+1}h\nabla f(x_n) + \sqrt{2\alpha_{n+1}h}U'_{n+1})
\end{aligned}
$$

Take $L_2$-norm on other randomness:

$$
\begin{aligned}
\|x_n^*(h) - x_{n+1}\|_{L^2} = & \left\| (x_n^* - x_n) - h(\nabla f(x_{n+\frac{1}{2}} + x_n^* - x_n) - \nabla f(x_{n+\frac{1}{2}})) \right\|_{L^2} \\
& + h \left\| \nabla f(x_n^* - \int_0^{\alpha_{n+1}h} \nabla f(x_n^*(s))ds + \sqrt{2}W_{\alpha_{n+1}h}) - \nabla f(x_n^* - \alpha_{n+1}h\nabla f(x_n) + \sqrt{2\alpha_{n+1}h}U'_{n+1}) \right\|_{L^2} \\
& + \left\| \int_0^h \nabla f(x_n^*(s))ds - h\nabla f(x_n^*(\alpha_{n+1}h)) \right\|_{L^2}
\end{aligned}
$$

Since $f$ is twice differentiable and $f$ is also $M$-gradient Lipschitz and strongly convex with parameter $m$,

$$\left\| (x_n^* - x_n) - h(\nabla f(x_{n+\frac{1}{2}} + x_n^* - x_n) - \nabla f(x_{n+\frac{1}{2}})) \right\|_{L^2} \le \rho \|x_n^* - x_n\|_{L^2}$$

where $\rho = \max(1 - mh, Mh - 1) = 1 - mh$.
For the second term:

$$
\begin{aligned}
h & \left\| \nabla f(x_n^* - \int_0^{\alpha_{n+1}h} \nabla f(x_n^*(s))ds + \sqrt{2}W_{\alpha_{n+1}h}) - \nabla f(x_n^* - \alpha_{n+1}h\nabla f(x_n) + \sqrt{2\alpha_{n+1}h}U'_{n+1}) \right\|_{L^2} \\
& \le Mh \left\| \int_0^{\alpha_{n+1}h} \nabla f(x_n^*(s)) - \nabla f(x_n)ds \right\|_{L^2} \\
& \le \frac{\sqrt{3}}{3}M^2h^2 \|x_n^* - x_n\|_{L^2} + \frac{\sqrt{3}}{3}M^2h^2 \sup_{0<s<h} \|x_n^*(s) - x_n^*\|_{L^2} \\
& \le \frac{\sqrt{3}}{3}M^2h^2 \|x_n^* - x_n\|_{L^2} + \frac{\sqrt{3}}{3}M^2h^2(4h^2\|\nabla f(x_n^*)\|^2 + 8M^2dh^3 + 2dh)^{\frac{1}{2}} \\
& \le \frac{\sqrt{3}}{3}M^2h^2 \|x_n^* - x_n\|_{L^2} + \frac{\sqrt{3}}{3}M^2h^2(2dh + 4Mdh^2 + 8M^2dh^3)^{\frac{1}{2}}
\end{aligned}
$$

For the third term:

$$\left\| \int_0^h \nabla f(x_n^*(s))ds - h\nabla f(x_n^*(\alpha_{n+1}h)) \right\|_{L^2} = \{\mathbb{E}\mathbb{E}_{\alpha_{n+1}}[(\int_0^h \nabla f(x_n^*(s))ds - h\nabla f(x_n^*(\alpha_{n+1}h)))^2]\}^{\frac{1}{2}}$$

$$= \{\mathbb{E}[h\int_0^h |\nabla f(x_n^*(s))|^2 ds - (\int_0^h \nabla f(x_n^*(s))ds)]\}^{\frac{1}{2}}$$

$$= \{\mathbb{E}[h\int_0^h |\nabla f(x_n^*(s)) - \frac{1}{h}\int_0^h \nabla f(x_n^*(s'))ds'|^2 ds]\}^{1/2}$$

$$\leq \{\mathbb{E}[h\int_0^h \frac{1}{h}\int_0^h \|\nabla f(x_n^*(s)) - \nabla f(x_n^*(s'))\|^2 ds'ds]\}^{1/2}$$

$$\leq 2Mh\{\sup_{s\in(0,h)} \|x_n^*(s) - x_n^*\|^2\}^{1/2}$$

$$\leq 2Mh(4h^2 \|\nabla f(x_n^*)\|^2 + 8M^2dh^3 + 2dh)^{1/2}$$

$$\leq 2Mh(2dh + 4Mdh^2 + 8M^2dh^3)^{\frac{1}{2}}$$

Combine all the bounds:

$$\|x_n^* - x_n\|_{L^2} \leq \frac{\frac{\sqrt{3}}{3}M^2h^2(2dh + 4Mdh^2 + 8M^2dh^3)^{\frac{1}{2}} + 2Mh(2dh + 4Mdh^2 + 8M^2dh^3)^{\frac{1}{2}}}{mh - \frac{\sqrt{3}}{3}M^2h^2}$$

The final statement follows by the fact that $W_2(\pi, \pi_h) \leq \|x_n - x_n^*\|_{L^2}$.  □

## 6.3  Proofs for Section 2.3

*Proof of Theorem 2.3.* From previous analysis, if we keep track of the coefficients in all those bounds and assume that $M\gamma_n \leq \frac{1}{2}$ for all $n$, we have:

$$\mathbb{E}[\|x_{n+1} - y_{n+1}\|^2] \leq (1 + m\gamma_{n+1})\mathbb{E}[\|y_{n+1} - x_{n+1}^*\|^2] + \frac{1}{m\gamma_{n+1}}\mathbb{E}\|\mathbb{E}_{\alpha_{n+1}}x_{n+1} - x_{n+1}^*\|^2 + \mathbb{E}\|x_{n+1} - x_{n+1}^*\|^2$$

$$\leq (1 + m\gamma_{n+1})e^{-2m\gamma_{n+1}}\mathbb{E}\|x_n - y_n\|^2 + \frac{1}{m\gamma_{n+1}}\mathbb{E}\|\mathbb{E}_{\alpha_{n+1}}x_{n+1} - x_{n+1}^*\|^2 + \mathbb{E}\|x_{n+1} - x_{n+1}^*\|^2$$

$$\leq (1 + m\gamma_{n+1})e^{-2m\gamma_{n+1}}\mathbb{E}\|x_n - y_n\|^2 + \frac{2\gamma_{n+1}^2}{m\gamma_{n+1}}\mathbb{E}\left\|\nabla f(x_{n+\frac{1}{2}}) - \nabla f(x_n(\alpha_{n+1}\gamma_{n+1}))\right\|^2$$

$$+ 2\gamma_{n+1}^2\mathbb{E}\left\|\nabla f(x_{n+\frac{1}{2}}) - \nabla f(x_n(\alpha_{n+1}\gamma_{n+1}))\right\|^2 + 2M^2\gamma_{n+1}^2\mathbb{E}\sup_{t\in[0,\gamma_{n+1}]}\|x_n(\alpha_{n+1}\gamma_{n+1}) - x_n(t)\|^2$$

$$\leq (1 + m\gamma_{n+1})e^{-2m\gamma_{n+1}}\mathbb{E}\|x_n - y_n\|^2$$

$$+ 2\gamma_{n+1}^4(1 + \frac{1}{m\gamma_{n+1}})M^4(\frac{1}{5}\gamma_{n+1}^2\|\nabla f(x_n)\|_{L^2}^2 + \frac{1}{6}M^2d\gamma_{n+1}^3 + \frac{2}{3}d\gamma_{n+1})$$

$$+ 4M^2\gamma_{n+1}^2\left(\frac{4\gamma_{n+1}^2}{1 - 2M^2\gamma_{n+1}}\|\nabla f(x_n)\|_{L^2}^2 + \frac{8M^2d\gamma_{n+1}^3}{1 - 2M^2\gamma_{n+1}^2} + 4d\gamma_{n+1}\right)$$

$$\leq (1 + m\gamma_{n+1})e^{-2m\gamma_{n+1}}\mathbb{E}\|x_n - y_n\|^2 + (33 + \kappa)M^2\gamma_{n+1}^4\|\nabla f(x_n)\|_{L^2}^2 + (33 + \kappa)M^2d\gamma_{n+1}^3$$

We can further bound $\|\nabla f(x_n)\|_{L^2}^2$:

$$\|\nabla f(x_n)\|_{L^2}^2 \leq 2\|\nabla f(y_n)\|_{L^2}^2 + 2\|\nabla f(y_n) - \nabla f(x_n)\|_{L^2}^2$$

$$\leq 2\|\nabla f(y_n)\|_{L^2}^2 + 2M^2\|x_n - y_n\|_{L^2}^2$$

$$\leq 2Md + 2M^2\|x_n - y_n\|_{L^2}^2$$

Therefore we have the following iterative inequality:

$$\mathbb{E}[\|x_{n+1} - y_{n+1}\|^2] \leq (1 + m\gamma_{n+1})e^{-2m\gamma_{n+1}}\mathbb{E}\|x_n - y_n\|^2 + 2(33 + \kappa)M^2d\gamma_{n+1}^3 + 2(33 + \kappa)M^4\gamma_{n+1}^4\mathbb{E}\|x_n - y_n\|^2$$

$$\leq \left[1 - m\gamma_{n+1} + (\frac{m^2}{2} + \frac{M^2(33 + \kappa)}{2})\gamma_{n+1}^2\right]\mathbb{E}\|x_n - y_n\|^2 + 2(33 + \kappa)M^2d\gamma_{n+1}^3$$

Since $(\gamma_n)$ is fast decreasing, we can assume that $\gamma_{n+1} \leq \frac{m}{m^2+M^2(33+\kappa)} \leq \frac{1}{m+34M}$ for large $n$, and for those $n$ we have

$$\mathbb{E}[\|x_{n+1} - y_{n+1}\|^2] \leq (1 - \frac{1}{2}m\gamma_{n+1})\mathbb{E}\|x_n - y_n\|^2 + 2(33+\kappa)M^2 d\gamma_{n+1}^3$$

Our strategy of choosing $(\gamma_n)$: for the first $K_1$ steps, we choose constant step size $h = \frac{1}{m+34M}$, $K_1$ is the first time so that $\mathbb{E}[\|x_{K_1} - y_{K_1}\|^2] \leq 5\kappa(\kappa+33)M(\frac{d^{\frac{1}{2}}}{m+34M})^2$. such $K_1$ exists because

$$\mathbb{E}[\|x_{K_1} - y_{K_1}\|^2] \leq (1 - \frac{m}{2m+68M})^{K_1}\mathbb{E}[\|x_0 - y_0\|^2] + \frac{2M^2(\kappa+33)d}{(m+34M)^2}\frac{2(m+34M)}{m}$$

$$= (1 - \frac{m}{2m+68M})^{K_1}\mathbb{E}[\|x_0 - y_0\|^2] + 4\kappa(\kappa+33)M(\frac{d^{\frac{1}{2}}}{m+34M})^2$$

**Claim:** There exists $\lambda > 0$ such that if we choose $\gamma_{n+1} = \frac{1}{m+34M+\lambda(n-K_1)}$ for all $n \geq K_1$, we can ensure that $\mathbb{E}[\|x_k - y_k\|^2] \leq 5\kappa(\kappa+33)M(\frac{d^{\frac{1}{2}}}{m+34M+\lambda(n-K_1)})^2$ for all $n \geq K_1$.
**Proof of Claim:** Simply use induction:

$$\mathbb{E}[\|x_{n+1} - y_{n+1}\|^2] \leq (1 - \frac{1}{2}m\gamma_{n+1})5\kappa(\kappa+33)Md\gamma_{n+1}^2 + 2M^2(\kappa+33)d\gamma_{n+1}^3$$

$$= 5\kappa(\kappa+33)Md\gamma_{n+1}^2(1 - \frac{m}{10}\gamma_{n+1})$$

Our goal is to ensure $5\kappa(\kappa+33)Md\gamma_{n+1}^2(1 - \frac{m}{10}\gamma_{n+1}) < 5\kappa(\kappa+33)M(\frac{d^{\frac{1}{2}}}{m+34M+\lambda(n+1-K_1)})^2$. It boils down to discuss the following polynomial inequality relates to $\lambda$:

$$G(\lambda) = (K - \frac{1}{10}m(K+1)^2)\lambda^2 + (X - \frac{1}{5}mX(K+1))\lambda - \frac{1}{10}mX^2 \leq 0$$

where $X = m + 34M$ and $K = n - K_1 > 0$. It's not hard to see that there's always positive $\lambda$ satisfying the inequality.
At last to get small error, we require $\mathbb{E}\|x_n - y_n\|^2 \leq \frac{d\epsilon^2}{m}$, i.e

$$5\kappa(\kappa+33)M\frac{d}{(m+34M+\lambda(n-K_1))^2} \leq \frac{d\epsilon^2}{m}$$

Then we have

$$n \geq K_1 + \lambda^{-1}m^{\frac{1}{2}}M^{\frac{1}{2}}\kappa^{\frac{1}{2}}(\kappa+33)^{\frac{1}{2}}/\epsilon - \lambda^{-1}(m+34M) \sim O(\kappa^{\frac{3}{2}}/\epsilon)$$

$\square$

### 6.3.1 Proof of Theorem 2

Before we prove Theorem 2, we need several intermediate results on the tightness of the (RLMC) chain.
**Lemma 3.** *Under assumption 6.1, for every continuous function $\varphi$ satisfying $\varphi(x) = o(V^k(x))$ for some $k \in \mathbb{N}$, $\lim_n \pi_n^\gamma(\varphi) = \pi(\varphi)$.*

*Proof of Lemma 3.* The proof is divided into three steps:

1) For all $p \geq 1$, there exists $\tilde{\alpha} \in (0,1)$ and $\tilde{\beta}, n_0 \in \mathbb{N}$ such that $\mathbb{E}[V^p(x_{n+1})|\mathcal{F}_n] \leq V^p(x_n) + \gamma_{n+1}V^{p-1}(x_n)(\tilde{\beta} - \tilde{\alpha}V(x_n))$ for all $n \geq n_0$.
When $p = 1$, the statement follows by the Lyapunov condition.
When $p > 1$, first we Taylor expand $V^p(x_{n+1})$ at $x_n$:

$$V^p(x_{n+1}) = V^p(x_n) + pV^{p-1}(x_n)\nabla V(x_n) \cdot (x_{n+1} - x_n) + \frac{1}{2}D^2(V^p)(\xi_{n+1})(x_{n+1} - x_n)^{\otimes 2}$$

$$= V^p(x_n) - \gamma_{n+1}pV^{p-1}(x_n)\nabla V(x_n) \cdot \nabla f(x_{n+\frac{1}{2}}) + \sqrt{2\gamma_{n+1}}pV^{p-1}\nabla V(x_n) \cdot U_{n+1}$$

$$+ \frac{1}{2}D^2(V^p)(\xi_{n+1})\left(-\gamma_{n+1}\nabla f(x_{n+\frac{1}{2}}) + \sqrt{2\gamma_{n+1}}U_{n+1}\right)^{\otimes 2}$$

$$\leq V^p(x_n) - \gamma_{n+1}pV^{p-1}(x_n)\nabla V(x_n) \cdot \nabla f(x_{n+\frac{1}{2}}) + \sqrt{2\gamma_{n+1}}pV^{p-1}(x_n)\nabla V(x_n) \cdot U_{n+1}$$

$$+ p\lambda_p V^{p-1}(\xi_{n+1})|-\gamma_{n+1}\nabla f(x_{n+\frac{1}{2}}) + \sqrt{2\gamma_{n+1}}U_{n+1}|^2$$

where $\xi_{n+1}$ is a point on the line segment joining $x_n$ and $x_{n+1}$ and $\lambda_p := \frac{1}{2}\lambda_{D^2V+(p-1)(\nabla V \otimes \nabla V)/V} < +\infty$.

Due to $\nabla(\sqrt{V}) = \frac{\nabla V}{2\sqrt{V}}$ and $|\nabla V|^2 \leq c_V V$, we have $\sqrt{V}$ is Lipschitz continuous and the Lipschitz constant $[\sqrt{V}]_1 = \frac{1}{4}c_V < +\infty$. Hence for a point $\xi_{n+1}$ on the line segment between $x_n$ and $x_{n+1}$,

$$V^{p-1}(\xi_{n+1}) = (\sqrt{V})^{2(p-1)}(\xi_{n+1}) \leq \left(\sqrt{V}(x_n) + [\sqrt{V}]_1 |x_{n+1} - x_n|\right)^{2(p-1)}$$

$$\leq \begin{cases} V^{p-1}(x_n) + [\sqrt{V}]_1^{2(p-1)}|x_{n+1} - x_n|^{2(p-1)}, & 2(p-1) \leq 1 \\ V^{p-1}(x_n) + c\left(V^{(2p-3)/2}(x_n)|x_{n+1} - x_n| + |x_{n+1} - x_n|^{2(p-1)}\right), & 2(p-1) > 1 \end{cases}$$

We can further bound

$$|x_{n+1} - x_n| = |-\gamma_{n+1}\nabla f(x_n) + \sqrt{2\gamma_{n+1}}U_{n+1} - \gamma_{n+1}\left(\nabla f(x_{n+\frac{1}{2}}) - \nabla f(x_n)\right)|$$

$$\leq \gamma_{n+1}|\nabla f(x_n)| + \sqrt{2\gamma_{n+1}}|U_{n+1}| + M\gamma_{n+1}| - \tilde{\gamma}_{n+1}\nabla f(x_n) + \sqrt{2\alpha_{n+1}\gamma_{n+1}}U'_{n+1}|$$

$$\leq \gamma_{n+1}(1 + M\alpha_{n+1}\gamma_{n+1})|\nabla f(x_n)| + \sqrt{2\gamma_{n+1}}|U_{n+1}| + \sqrt{2}M\gamma_{n+1}\alpha_{n+1}^{\frac{1}{2}}\gamma_{n+1}^{\frac{1}{2}}|U'_{n+1}|$$

$$\leq C\sqrt{V}(x_n)\gamma_{n+1}^{\frac{1}{2}}(1 + |U_{n+1}| + |U'_{n+1}|)$$

Plug these results into the last term in the first inequality we obtained from Taylor expansion:

$$p\lambda_p V^{p-1}(\xi_{n+1})|x_{n+1} - x_n|^2 \leq p\lambda_p V^{p-1}(x_n)|x_{n+1} - x_n|^2$$

$$+ Cp\lambda_p \begin{cases} |x_{n+1} - x_n|^{2p}, & 2p \leq 3 \\ V^{(2p-3)/2}(x_n)|x_{n+1} - x_n|^3 + |x_{n+1} - x_n|^{2p}, & 2p > 3 \end{cases}$$

$$\leq p\lambda_p V^{p-1}(x_n)|x_{n+1} - x_n|^2 + C\gamma_{n+1}^{p \wedge \frac{3}{2}}V^p(x_n)(1 + |U_{n+1}|^{2p} + |U'_{n+1}|^{2p})$$

We then take conditional expectation, there exists $\alpha > 0$ and $\beta \geq 0$ such that for all $n \geq n_0$:

$$\mathbb{E}[V^p(x_{n+1})|\mathcal{F}_n] \leq V^p(x_n) - pV^{p-1}(x_n)(\alpha V(x_n) - \beta)$$

$$- p\gamma_{n+1}V^{p-1}(x_n)\mathbb{E}[\nabla V(x_n) \cdot \left(\nabla f(x_{n+\frac{1}{2}}) - \nabla f(x_n)\right)|\mathcal{F}_n]$$

$$+ 2p\lambda_p V^{p-1}(x_n)\mathbb{E}[\gamma_{n+1}^2|\nabla f(x_{n+\frac{1}{2}})|^2 + 2\gamma_{n+1}|U_{n+1}|^2|\mathcal{F}_n]$$

$$+ CV^p(x_n)(1 + \mathbb{E}|U_{n+1}|^{2p} + \mathbb{E}|U'_{n+1}|^{2p})\gamma_{n+1}^{p \wedge \frac{3}{2}}$$

$$\leq V^p(x_n) - pV^{p-1}(x_n)(\alpha V(x_n) - \beta) + 2p\lambda_p \mathbb{E}|U_{n+1}|^2\gamma_{n+1}V^{p-1}(x_n)$$

$$+ CV^p(x_n)(1 + \mathbb{E}|U_{n+1}|^{2p} + \mathbb{E}|U'_{n+1}|^{2p})\gamma_{n+1}^{p \wedge \frac{3}{2}}$$

$$+ c_V Mp\gamma_{n+1}^2 V^p(X_n) + \sqrt{2}c_V Mp\gamma_{n+1}^{\frac{3}{2}}\mathbb{E}|U'_{n+1}|V^{p-1/2}(x_n)$$

$$+ c_V p\lambda_p \gamma_{n+1}^2 V^{p-1}(x_n)\mathbb{E}[V(x_{n+\frac{1}{2}})|\mathcal{F}_n]$$

From $x_n$ to $x_{n+\frac{1}{2}}$, it's simply the Euler discretization with time step $\alpha_{n+1}\gamma_{n+1}$, we could use the result in [21]: there exists a $\bar{\alpha} > 0$ and $\bar{\beta} \in \mathbb{R}$ such that for all $n \geq n_0$:

$$\mathbb{E}[V(x_{n+\frac{1}{2}})|\mathcal{F}_n] \leq V(x_n)(1 - \bar{\alpha}\tilde{\gamma}_{n+1}) + \bar{\beta}\tilde{\gamma}_{n+1}$$

Therefore we have

$$\mathbb{E}[V^{x_{n+1}}|\mathcal{F}_n] \leq (1 - \alpha p\gamma_{n+1} + o(\gamma_{n+1}))V^p(x_n) + \gamma_{n+1}V^{p-1}(x_n)(p\beta + 2p\lambda_p \mathbb{E}|U_{n+1}|^2 + c_V Mp\mathbb{E}|U'_{n+1}|^2)$$

There exists $\hat{\alpha} > 0$ and $\hat{\beta} \in \mathbb{R}$ such that for all $n \geq n_0$:

$$\mathbb{E}[V^p(x_{n+1})|\mathcal{F}_n] \leq V^p(x_n) + \gamma_{n+1}V^{p-1}(x_n)\left(\hat{\beta} - \hat{\alpha}V(x_n)\right)$$

2) From step 1), we derive

$$\sup_{n \geq n_0} \mathbb{E}[V^p(x_n)] \leq (\frac{\hat{\beta}}{\hat{\alpha}})^p \vee \mathbb{E}[V^p(x_{n_0})]$$

Hence $\sup_n \mathbb{E}[V^p(x_n)] < +\infty$ for all $p \geq 1$. Therefore $\sup_n \pi_n^\gamma(\omega, V^p) < +\infty$ $\mathbb{P} - a.s$ for all $p \geq 1$.

3) Identification of the weak limit: To identify the limit, we essentially follow the same steps in [21] and hence we omit the proof.

    (a) **(Echeverrría-Weiss Theorem)** Let $E$ be a locally compact Polish space and $A$ a linear operator satisfying the positive maximum principle. Assume that its domain $\mathcal{D}(A)$ is an algebra everywhere dense in $(\mathcal{C}_0(E), \| \ \|_\infty)$ containing a sequence $(f_n)_{n\in\mathbb{N}}$ satisfying

$$\sup_{n\in\mathbb{N}} (\|f_n\|_\infty + \|\mathcal{L}f_n\|_\infty) < +\infty, \ \ \forall x \in E, \ f_n(x) \to 1 \ \text{and} \ Af_n(x) \to 0.$$

If a distribution on $(E, \mathcal{B}(E))$ satisfies $\int_E Af d\nu = 0$ for every $f \in \mathcal{D}(A)$, then there exists a stationary solution for the martingale problem $(A, \nu)$ (this means that there exists a stationary continuous-time homogeneous Markov process with infinitesimal generator $A$ and invariant distribution $\nu$).

    (b) The generator of the Langevin dynamics, $\mathcal{A}$, satisfies the assumptions of the Echeverrría-Weiss theorem.

    (c) Under assumption 6.1, for every bounded Lipschitz continuous function $\varphi : \mathbb{R}^d \to \mathbb{R}$, $\lim_n \frac{1}{\Gamma_n} \sum_{k=1}^n \mathbb{E}[\varphi(x_k) - \varphi(x_{k-1})|\mathcal{F}_{k-1}] = 0 \ \mathbb{P} - a.s.$

    (d) Under assumption 6.1, for every twice continuously differentiable function $\varphi$ with compact support, $\lim_n \left( \frac{1}{\Gamma_n} \sum_{k=1}^n \mathbb{E}[\varphi(x_k) - \varphi(x_{k-1})|\mathcal{F}_{k-1}] - \pi_n^\gamma(\mathcal{A}\varphi) \right) = 0 \ \mathbb{P} - a.s.$

$a), b), c), d)$ together implies that the weak limit of the empirical distribution $\pi_n^\gamma$ is $\pi$, i.e the stationary distribution of the Langevin dynamics.

$\square$

*Proof of Theorem 2.* Since $f$ satisfies assumption 1.1, we can show that the Langevin dynamics satisfies assumption 6.1. Therefore lemma 3 is true. Then we may use the following method to discuss the CLT of (RLMC).

$$
\begin{aligned}
x_k - x_{k-1} &= -\gamma_k \left( \nabla f(x_{k-1}) + D^2 f(x_{k-1})(x_{k-\frac{1}{2}} - x_{k-1}) + r_2(x_{k-\frac{1}{2}}, x_{k-1}) \right) + \sqrt{2\gamma_k} U_k \\
&= -\gamma_k \nabla f(x_{k-1}) + \sqrt{2\gamma_k} U_k - \gamma_k D^2 f(x_{k-1})(x_{k-\frac{1}{2}} - x_{k-1}) - \gamma_k r_2(x_{k-\frac{1}{2}}, x_{k-1}) \\
&= -\gamma_k \nabla f(x_{k-1}) + \sqrt{2\gamma_k} U_k + \alpha_k \gamma_k^2 D^2 f(x_{k-1})\nabla f(x_{k-1}) - \sqrt{2\alpha_k} \gamma_k^{\frac{3}{2}} \nabla^2 f(x_{k-1}) U_k' - \gamma_k r_2(x_{k-\frac{1}{2}}, x_{k-1})
\end{aligned}
$$

where

$$
\begin{aligned}
r_2(x_{k-\frac{1}{2}}, x_{k-1}) &= \nabla f(x_{k-\frac{1}{2}}) - \nabla f(x_{k-1}) - D^2 f(x_{k-1})(x_{k-1} - x_{k-\frac{1}{2}}) \\
&= \frac{1}{2} D^3 f(x_{k-1})(x_{k-\frac{1}{2}} - x_{k-1})^{\otimes 2} + \frac{1}{6} D^4 f(x_{k-1})(x_{k-\frac{1}{2}} - x_{k-1})^{\otimes 3} + O(\gamma_k^2) \\
&= \alpha_k \gamma_k D^3 f(x_{k-1}) U_k'^{\otimes 2} - \sqrt{2}\alpha_k^{\frac{3}{2}} \gamma_k^{\frac{3}{2}} \langle D^3 f(x_{k-1}); \nabla f(x_{k-1}), U_k' \rangle \\
&\quad + \frac{\sqrt{2}}{3} \alpha_k^{\frac{3}{2}} D^4 f(x_{k-1}) U_k'^{\otimes 4} + O(\gamma_k^2)
\end{aligned}
$$

Then

$$
\begin{aligned}
x_k - x_{k-1} &= -\gamma_k \nabla f(x_{k-1}) + \sqrt{2\gamma_k} U_k - \sqrt{2\alpha_k} \gamma_k^{\frac{3}{2}} \nabla^2 f(x_{k-1}) U_k' \\
&\quad + \alpha_k \gamma_k^2 D^2 f(x_{k-1})\nabla f(x_{k-1}) - \alpha_k \gamma_k^2 D^3 f(x_{k-1}) U_k'^{\otimes 2} + O(\gamma_k^{\frac{5}{2}})
\end{aligned}
$$

We can decompose $\phi(x_k)$:

$$
\begin{aligned}
\phi(x_k) - \phi(x_{k-1}) &= \nabla \phi(x_{k-1})(x_k - x_{k-1}) + \frac{1}{2} D^2 \phi(x_{k-1})(x_k - x_{k-1})^{\otimes 2} + \frac{1}{6} D^3 \phi(x_{k-1})(x_k - x_{k-1})^{\otimes 3} \\
&\quad + \frac{1}{24} D^4 \phi(x_{k-1})(x_k - x_{k-1})^{\otimes 4} + O(\gamma_k^{\frac{5}{2}}) \\
&= \nabla \phi(x_{k-1})(\sqrt{2}\gamma_k^{\frac{1}{2}} U_k - \gamma_k \nabla f(x_{k-1}) - \sqrt{2\alpha_k}\gamma_k^{\frac{3}{2}} D^2 f(x_{k-1}) U_k' \\
&\quad + \alpha_k \gamma_k^2 D^2 f(x_{k-1})\nabla f(x_{k-1}) - \alpha_k \gamma_k^2 D^3 f(x_{k-1}) U_k'^{\otimes 2}) \\
&\quad + \frac{1}{2} D^2 \phi(x_{k-1}) \left( \sqrt{2}\gamma_k^{\frac{1}{2}} U_k - \gamma_k \nabla f(x_{k-1}) - \sqrt{2\alpha_k}\gamma_k^{\frac{3}{2}} D^2 f(x_{k-1}) U_k' \right)^{\otimes 2} \\
&\quad + \frac{1}{6} D^3 \phi(x_{k-1}) \left( \sqrt{2}\gamma_k^{\frac{1}{2}} U_k - \gamma_k \nabla f(x_{k-1}) \right)^{\otimes 3} + \frac{1}{24} D^4 \phi(x_{k-1})(\sqrt{2}\gamma_k^{\frac{1}{2}} U_k)^{\otimes 4} + O(\gamma_k^{\frac{5}{2}})
\end{aligned}
$$

If $\mathcal{A}$ is the generator of Langevin dynamics and summing up over $k$:

$$\sum_{k=1}^{n}\gamma_k\mathcal{A}\phi(x_{k-1}) = \phi(x_n) - \phi(x_0) - \sqrt{2}\sum_{k=1}^{n}\gamma_k^{\frac{1}{2}}\nabla\phi(x_{k-1})U_k - \sum_{k=1}^{n}\gamma_k\left(D^2\phi(x_{k-1})U_k^{\otimes 2} - \mathbb{E}[D^2\phi(x_{k-1})U_k^{\otimes 2}|\mathcal{F}_{k-1}]\right)$$

$$+ \sqrt{2}\sum_{k=1}^{n}\gamma_k^{\frac{3}{2}}\langle D^2\phi(x_{k-1}); \nabla f(x_{k-1}), U_k\rangle - \frac{\sqrt{2}}{3}\sum_{k=1}^{n}\gamma_k^{\frac{3}{2}}D^3\phi(x_{k-1})U_k^{\otimes 3}$$

$$+ \sum_{k=1}^{n}\sqrt{2\alpha_k}\gamma_k^{\frac{3}{2}}\langle D^2 f(x_{k-1}); \nabla\phi(x_{k-1}), U_k'\rangle + \sum_{k=1}^{n}\gamma_k^2\langle D^3\phi(x_{k-1}); \nabla f(x_{k-1}), U_k^{\otimes 2}\rangle$$

$$- \sum_{k=1}^{n}\alpha_k\gamma_k^2\langle D^2 f(x_{k-1}); \nabla\phi(x_{k-1}), \nabla f(x_{k-1})\rangle + \sum_{k=1}^{n}\alpha_k\gamma_k^2\langle D^3 f(x_{k-1}); \nabla\phi(x_{k-1}), U_k'^{\otimes 2}\rangle$$

$$- \frac{1}{2}\sum_{k=1}^{n}\gamma_k^2 D^2\phi(x_{k-1})\nabla f(x_{k-1})^{\otimes 2} + \sum_{k=1}^{n}2\alpha_k^{\frac{1}{2}}\gamma_k^2\langle D^2\phi(x_{k-1}); D^2\phi(x_{k-1})U_k', U_k\rangle$$

$$- \frac{1}{6}\sum_{k=1}^{n}\gamma_k^2 D^4\phi(x_{k-1})U_k^{\otimes 4} + \sum_{k=1}^{n}O(\gamma_k^{\frac{5}{2}})$$

$$:= N_n^{(0)} + N_n^{(\frac{1}{2})} + N_n^{(1)} + N_n^{(\frac{3}{2})} + N_n^{(2)} + N_n^{(\frac{5}{2})}$$

In the fast decreasing time step situation($\sum_{k=1}^{n}\gamma_k^2/\sqrt{\Gamma_n} \to 0$), the CLT for (RLMC) is the same as that of LMC. In the slowly decreasing time step situation, when $\sum_{k=1}^{n}\gamma_k^2/\sqrt{\Gamma_n} \to \hat{\gamma} \in (0, +\infty]$:

a) $\frac{\phi(x_n) - \phi(x_0)}{\Gamma_n^{(2)}} \to 0$ because $(x_n)$ is tight and $\phi$ is continuous.

b) $\frac{-\sqrt{2}\sum_{k=1}^{n}\gamma_k^{\frac{1}{2}}\nabla\phi(x_{k-1})U_k}{\sqrt{\Gamma_n}} \implies \mathcal{N}(0, 2\int_{\mathbb{R}^d}|\nabla\phi(x)|^2\pi(dx))$. Therefore,

$$\frac{-\sqrt{2}\sum_{k=1}^{n}\gamma_k^{\frac{1}{2}}\nabla\phi(x_{k-1})U_k}{\Gamma_n^{(2)}} \implies \begin{cases} \mathcal{N}(0, 2\hat{\gamma}^{-2}\int_{\mathbb{R}^d}|\nabla\phi(x)|^2\pi(dx)), & \text{when } \hat{\gamma} < +\infty \\ 0, & \text{when } \hat{\gamma} = +\infty \end{cases}$$

c) $\frac{-\sum_{k=1}^{n}\gamma_k\left(D^2\phi(x_{k-1})U_k^{\otimes 2} - \mathbb{E}[D^2\phi(x_{k-1})U_k^{\otimes 2}|\mathcal{F}_{k-1}]\right)}{\sqrt{\Gamma_n}} \to 0$ in $L^2$.

d) $\frac{\sqrt{2}\sum_{k=1}^{n}\gamma_k^{\frac{3}{2}}\langle D^2\phi(x_{k-1}); \nabla f(x_{k-1}), U_k\rangle}{\sqrt{\Gamma_n}} \to 0$ in $L^2$.

$\frac{-\frac{\sqrt{2}}{3}\sum_{k=1}^{n}\gamma_k^{\frac{3}{2}}D^3\phi(x_{k-1})U_k^{\otimes 3}}{\sqrt{\Gamma_n}} \to 0$ in probability because $\mathbb{E}[U_k^{\otimes 3}] = 0$.

$\frac{\sum_{k=1}^{n}\sqrt{2\alpha_k}\gamma_k^{\frac{3}{2}}\langle D^2 f(x_{k-1}); \nabla\phi(x_{k-1}), U_k'\rangle}{\sqrt{\Gamma_n}} \to 0$ in $L^2$.

Therefore $\frac{N_n^{(\frac{3}{2})}}{\Gamma_n^{(2)}} \to 0$ in probability.

e) $\frac{\sum_{k=1}^{n}\gamma_k^2\langle D^3\phi(x_{k-1}); \nabla f(x_{k-1}), U_k^{\otimes 2}\rangle}{\Gamma_n^{(2)}} \to \int_{\mathbb{R}^d}\int_{\mathbb{R}^d}\langle D^3\phi(x); \nabla f(x), u^{\otimes 2}\rangle\mu(du)\pi(dx)$ in probability.

$\frac{-\sum_{k=1}^{n}\alpha_k\gamma_k^2\langle D^2 f(x_{k-1}); \nabla\phi(x_{k-1}), \nabla f(x_{k-1})\rangle}{\Gamma_n^{(2)}} \to -\frac{1}{2}\int_{\mathbb{R}^d}\langle D^2 f(x); \nabla\phi(x), \nabla f(x)\rangle\pi(dx)$ in probability.

$\frac{\sum_{k=1}^{n}\alpha_k\gamma_k^2\langle D^3 f(x_{k-1}); \nabla\phi(x_{k-1}), U_k'^{\otimes 2}\rangle}{\Gamma_n^{(2)}} \to \frac{1}{2}\int_{\mathbb{R}^d}\int_{\mathbb{R}^2}\langle D^3 f(x); \nabla\phi(x), u^{\otimes 2}\rangle\mu'(du)\pi(dx)$ in probability.

$\frac{-\frac{1}{2}\sum_{k=1}^{n}\gamma_k^2 D^2\phi(x_{k-1})\nabla f(x_{k-1})^{\otimes 2}}{\Gamma_n^{(2)}} \to -\frac{1}{2}\int_{\mathbb{R}^d}D^2\phi(x)\nabla f(x)^{\otimes 2}\pi(dx)$ in probability.

$\frac{\sum_{k=1}^{n}2\alpha_k^{\frac{1}{2}}\gamma_k^2\langle D^2\phi(x_{k-1}); D^2\phi(x_{k-1})U_k', U_k\rangle}{\Gamma_n^{(2)}} \to 0$ in $L^2$.

$\frac{-\frac{1}{6}\sum_{k=1}^{n}\gamma_k^2 D^4\phi(x_{k-1})U_k^{\otimes 4}}{\Gamma_n^{(2)}} \to -\frac{1}{6}\int_{\mathbb{R}^d}\int_{\mathbb{R}^d}D^4\phi(x)u^{\otimes 4}\mu(du)\pi(dx)$ in probability.

Therefore
$$\frac{N_n^{(2)}}{\Gamma_n^{(2)}} \to \varrho \qquad \text{in probability}$$

where

$$\varrho = \int_{\mathbb{R}^d} \int_{\mathbb{R}^d} \langle D^3\phi(x); \nabla f(x), u^{\otimes 2}\rangle \mu(du)\pi(dx) - \frac{1}{2}\int_{\mathbb{R}^d} \langle D^2 f(x); \nabla\phi(x), \nabla f(x)\rangle \pi(dx)$$

$$+ \frac{1}{2}\int_{\mathbb{R}^d}\int_{\mathbb{R}^2} \langle D^3 f(x); \nabla\phi(x), u^{\otimes 2}\rangle \mu(du)\pi(dx) - \frac{1}{2}\int_{\mathbb{R}^d} D^2\phi(x)\nabla f(x)^{\otimes 2}\pi(dx)$$

$$- \frac{1}{6}\int_{\mathbb{R}^d}\int_{\mathbb{R}^d} D^4\phi(x)u^{\otimes 4}\mu(du)\pi(dx)$$

and $\mu$ is the distribution for a $d$-dimensional standard Gaussian random variable.

f) $\frac{N_n^{\frac{5}{2}}}{\Gamma_n^{(2)}} \to 0$ in $L^1$.

As a conclusion, we obtain the proof of part (1) of the theorem:

$$\frac{\sum_{k=1}^n \gamma_k \mathcal{A}\phi(x_{k-1})}{\Gamma_n^{(2)}} \to \begin{cases} \mathcal{N}(\varrho, 2\hat{\gamma}^{-2}\int_{\mathbb{R}^d} |\nabla\phi(x)|^2\pi(dx)), & \text{when } \hat{\gamma} < +\infty \\ \varrho \quad , & \text{when } \hat{\gamma} = +\infty \end{cases}$$

For the fast decreasing step, i.e., part (2) of the theorem, the proof follows by the same arguments in the corresponding part of Theorem 10 in [21] and hence we omit it. $\qquad\square$

# 7 Proofs for Section 3

In this section, we would denote the drift function that appears in 4 as $b(x, v)$, i.e.

$$b(x, v) = \begin{bmatrix} v \\ -2v - u\nabla f(x) \end{bmatrix}$$

**Assumption 7.1.** *There exists a twice differentiable function $V : \mathbb{R}^{2d} \to [1, \infty)$ such that: (0) $\lim_{\|(x,v)\|\to\infty} V(x, v) = +\infty$, (1) there exists $\alpha > 0$ and $\beta > 0$: $\langle\nabla V(x, v), b(x, v)\rangle \leq -\alpha V(x, v) + \beta$ for every $(x, v)$, (2) there exists $c_V > 0$: $\|\nabla V(x, v)\|^2 + \|b(x, v)\|^2 \leq c_V V(x, v)$ for every $(x, v)$, and (3) $\left\|D^2 V\right\|_\infty := \sup_{(x,v)\in\mathbb{R}^{2d}} \|D^2 V\|_{op} < \infty$.*

**Lemma 4.** *Assumption 1.1 implies Assumption 7.1 when $u \in (0, \frac{4}{2M-m})$.*

*Proof of Lemma 4.* For simplicity, We choose $V(x, v) = \|x - x_*\|^2 + \|x - x_* + v\|^1 + 1$ with $f(x_*) = \min f(x)$. Now we check conditions 0), 1), 2), 3) in $(\mathcal{L}_{V,\infty})$ are satisfied.

0) It's onvious that $\lim_{|(x,v)|\to+\infty} V(x, v) = +\infty$ and $V(x, v) \geq 1$ for all $(x, v) \in \mathbb{R}^d$.
3) The Hessian of $V$ we choose is

$$D^2 V(x, v) = \begin{bmatrix} 4I_d & 2I_d \\ 2I_d & 2I_d \end{bmatrix}$$

For arbitrary $(x, v)^T, (y, w)^T \in \mathbb{R}^{2d}$:

$$\left\|D^2 V(x, v)(y, w)^T\right\|^2 = \left\|\begin{bmatrix} 4y + 2w \\ 2y + 2w \end{bmatrix}\right\|^2$$

$$\leq 40\left\|(y, w)^T\right\|^2$$

Therefore $\left\|D^2 V\right\|_\infty < \infty$.

2) Take gradient of the $V$ we choose:

$$\nabla V(x,v) = \begin{bmatrix} 2(x - x_*) + 2(x - x_* + v) \\ 2(x - x_* + v) \end{bmatrix}$$

Then for all $(x,v) \in \mathbb{R}^{2d}$,

$$
\begin{aligned}
|\nabla V(x,v)|^2 + |b(x,v)|^2 &\leq 2(4\|x - x_*\|^2 + 4\|x - x_* + v\|^2) + 4\|x - x_* + v\|^2 \\
&\quad + \|v\|^2 + 2(4\|v\|^2 + u^2\|\nabla f(x)\|^2) \\
&\leq 8\|x - x_*\|^2 + 12\|x - x_* + v\|^2 + 9\|v\|^2 + 2u^2 M^2 \|x - x_*\|^2 \\
&\leq \max\{26 + 2u^2 M^2, 30\} V(x,v)
\end{aligned}
$$

1) Last we consider

$$
\begin{aligned}
\langle \nabla V(x,v), b(x,v)\rangle &= 2(x - x_*) \cdot v + 2(x - x_* + v) \cdot v - 4(x - x_* + v) \cdot v \\
&\quad - 2u(x - x_* + v) \cdot \nabla f(x) \\
&\leq -2\|v\|^2 - 2u\left[ f(x) - f(x_* - v) + \frac{m}{2}\|x - x_* + v\|^2 \right] \\
&\leq -2\|v\|^2 - um\|x - x_* + v\|^2 - 2u\left( f(x_*) + \frac{m}{2}\|x - x_*\|^2 \right) \\
&\quad + 2u\left( f(x_*) + \frac{M}{2}\|v\|^2 \right) \\
&= -um\|x - x_* + v\|^2 - um\|x - x_*\|^2 - (2 - uM)\|v\|^2
\end{aligned}
$$

The second inequality follows from the fact that $f$ is $m$-strongly convex.
When $u \in (0, \frac{2}{M}]$, $\langle \nabla V(x,v), b(x,v)\rangle \leq -umV(x,v) + um$ for all $(x,v) \in \mathbb{R}^{2d}$. Therefore 1) is satisfied.
When $u > \frac{2}{M}$, we can use triangle inequality to further bound our result:

$$
\begin{aligned}
\langle \nabla V(x,v), b(x,v)\rangle &\leq -um\|x - x_* + v\|^2 - um\|x - x_*\|^2 + (uM - 2)\|v\|^2 \\
&\leq [-um + 2(uM - 2)](\|x - x_* + v\|^2 + \|x - x_*\|^2) \\
&\leq -[4 - u(2M - m)]V(x,v) - [4 - u(2M - m)]
\end{aligned}
$$

When $u \in (\frac{2}{M}, \frac{4}{2M - m})$, 1) is satisfied because $4 - u(2M - m) > 0$.
Therefore, 1) holds when $u \in (0, \frac{4}{2M - m})$.

$\square$

**Remark 9.** *For the $V(x,v)$ we choose in the proof, under assumption 1.1, we can verify that: $V(x,v) = O(|x|^2 + |v|^2)$ when $|(x,v)| \to +\infty$. We will use this fact later in the proof when we establish the CLT statement.*

### 7.1 Proofs for Section 3.1

*Proof of Theorem 3.* Under the assumption 7.1, we can show that the following Lyapunov condition is satisfied for small $h$.
**(Lyapunov Condition):** There exists a function $V : \mathbb{R}^{2d} \to [1, \infty)$ such that:

0) $\lim_{|(x,v)| \to \infty} V(x,v) = +\infty$,
1) There exists $\hat{\alpha} \in (0,1)$ and $\hat{\beta} \geq 0$: $\mathbb{E}[V(x_{n+1}, v_{n+1})|\mathcal{F}_n] \leq \hat{\alpha}V(x_n, v_n) + \hat{\beta}$.

**Proof:** To show that assumption 7.1 implies Lyapunov condition, we first do Taylor expansion of $V(x_{n+1}, v_{n+1})$ at $(x_n, v_n)$:

$$V(x_{n+1}, v_{n+1}) = V(x_n, v_n) + \nabla V(x_n, v_n) \cdot (x_{n+1} - x_n, v_{n+1} - v_n)^T + \frac{1}{2}D^2 V(\theta_n)[(x_{n+1} - x_n, v_{n+1} - v_n)^T]^{\otimes 2}$$

where $\theta_n$ is a random point on the line segment joining $(x_n, v_n)$ and $(x_{n+1}, v_{n+1})$. Use the RULMC algorithm and part (a) of Assumption 1.1:

$$\mathbb{E}[V(x_{n+1}, v_{n+1})|\mathcal{F}_n] \leq V(x_n, v_n) + \nabla V(x_n, v_n) \cdot \begin{bmatrix} \frac{1-e^{-2h}}{2}v_n - \frac{u}{2}(h - \frac{1-e^{-2h}}{2})\nabla f(x_n) \\ -2\frac{1-e^{-2h}}{2}v_n - u\frac{1-e^{-2h}}{2}\nabla f(x_n) \end{bmatrix}$$

$$- \nabla V(x_n.v_n) \cdot \begin{bmatrix} \frac{u}{2}(h - \frac{1-e^{-2h}}{2})\mathbb{E}[\nabla f(x_{n+\frac{1}{2}}) - \nabla f(x_n)|\mathcal{F}_n] \\ u\frac{1-e^{-2h}}{2}\mathbb{E}[\nabla f(x_{n+\frac{1}{2}}) - \nabla f(x_n)|\mathcal{F}_n] \end{bmatrix}$$

$$+ \frac{3M}{2}\left[5(\frac{1-e^{-2h}}{2})^2|v_n|^2 + u^2h^2|\nabla f(x_n)^2| + (\sigma_{n+1}^{(2)}{}^2 + 4\sigma_{n+1}^{(3)}{}^2)ud\right]$$

$$+ \frac{3M}{2}u^2h^2\mathbb{E}[|\nabla f(x_{n+\frac{1}{2}})|^2 - |\nabla f(x_n)|^2|\mathcal{F}_n]$$

where we can further estimate

$$\mathbb{E}[\nabla f(x_{n+\frac{1}{2}}) - \nabla f(x_n)|\mathcal{F}_n] \leq M\mathbb{E}[x_{n+\frac{1}{2}} - x_n|\mathcal{F}_n]$$

$$\leq M\frac{1}{2h}(h - \frac{1-e^{-2h}}{2})|v_n| + \sqrt{ud}M\sigma_{n+1}^{(1)}$$

$$+ \frac{u}{2}(\frac{h}{2} - \frac{h - \frac{1-e^{-2h}}{2}}{2h})|\nabla f(x_n)|$$

and there exists $\xi_n$ such that $|\nabla f(x_{n+\frac{1}{2}})|^2 - |\nabla f(x_n)|^2| = 2D^2f(\xi_n)\nabla f(\xi_n)$ and $\xi_n$ is on the line segment joining $x_n$ and $x_{n+\frac{1}{2}}$. Therefore $|\xi_n - x_n| \leq |x_{n+\frac{1}{2}} - x_n|$. then we have

$$\mathbb{E}[|\nabla f(x_{n+\frac{1}{2}})|^2 - |\nabla f(x_n)|^2|\mathcal{F}_n] \leq 2M\mathbb{E}[|\nabla f(\xi_n)||\mathcal{F}_n]$$

$$\leq 2M|\nabla f(x_n)| + 2M^2\mathbb{E}[[x_{n+\frac{1}{2}} - x_n|\mathcal{F}_n]]$$

$$\leq 2M^3\frac{1}{2h}(h - \frac{1-e^{-2h}}{2})|v_n| + 2\sqrt{ud}M^3\sigma_{n+1}^{(1)}$$

$$+ \left[2M + uM^2(\frac{h}{2} - \frac{h - \frac{1-e^{-2h}}{2}}{2h})|\nabla f(x_n)|\right]$$

When $h$ is small, we can use polynomials of $h$ to bound those exponential coefficients. We can obtain that there exists $C > 0$:

$$\mathbb{E}[V(x_{n+1}, v_{n+1})|\mathcal{F}_n] \leq V(x_n, v_n) + h\nabla V(x_n, v_n) \cdot b(x_n, v_n)^T + Ch^2(d + |v_n|^2 + |\nabla f(x_n)|^2)$$

then assumption 7.1 imples that there exists $\alpha > 0, \beta > 0$ such that

$$\mathbb{E}[V(x_{n+1}, v_{n+1})|\mathcal{F}_n] \leq (1 - \alpha h + Cc_V h^2)V(x_n, v_n) + Ch^2 d + \beta$$

When $h$ is small, there exists $\hat{\alpha} = 1 - \alpha h + Cc_V h^2 \in (0, 1)$ and $\hat{\beta} = Ch^2 d + \beta > 0$ such that $\mathbb{E}[V(x_{n+1}, v_{n+1})|\mathcal{F}_n] \leq \hat{\alpha}V(x_n, v_n) + \hat{\beta}$. $\qquad\square$

Once we have the Lyapunov condition, we can define the stopping time $\tau_C = \inf\{n > 0 : (x_n, v_n) \in C\}$ and show that $\sup_{(x,v) \in C} \mathbb{E}_{(x,v)}[\tau_C] \leq M_C < \infty$ for all small set C. Then uniqueness of stationary probability measure and ergodicity all follow by Theorem 1.3.1 in [33]. Next we prove that $\sup_{(x,v) \in C} \mathbb{E}_{(x,v)}[\tau_C] \leq M_C < \infty$ given Lyapunov condition. To do so, note that we have

$$\mathbb{E}_{(x,v)}[\tau_C] = \sum_{n=1}^{\infty} n\mathbb{P}(\tau_C = n)$$

$$= \sum_{n \geq 1} \mathbb{P}(\tau_C > n - 1)$$

Under Lyapunov condition, for any stopping time $N$, according to Lemma A.3 and Corollary A.4 in [31], we have

$$\mathbb{P}(\tau_C > n - 1) \le \mathbb{E}[V(x_n, v_n) 1_{\tau_C > n-1}]$$
$$\le \frac{\kappa[\gamma^{n-1} V(x_0, v_0) + 1]}{1 - \gamma}$$
$$\le \kappa \gamma^{n-1} [V(x_0, v_0) + 1]$$

for some $\gamma \in (\hat{\alpha}, 1)$ and constant $\kappa$. Therefore, we have

$$\mathbb{E}_{(x,v)}[\tau_C] \le \sum_{k \ge 1} \kappa \gamma^{n-1} [V(x_0, v_0) + 1]$$
$$= \frac{\kappa[V(x, v) + 1]}{1 - \gamma}$$

and

$$\sup_{(x,v) \in C} \mathbb{E}_{(x,v)}[\tau_C] \le \frac{\kappa}{1 - \gamma} \sup_{(x,v) \in C} V(x, v) + \frac{\kappa}{1 - \gamma} \le M_C < \infty$$

So as a conclusion, the statement of the theorem follows. $\qquad\square$

*Proof of Proposition 3.1 .* Denote $A_n^2 = \mathbb{E}[\|x_n - y_n\|^2 + \|(x_n + v_n) - (y_n + w_n)\|^2]$. Using triangle inequality we have

$$\mathbb{E}_\alpha[\|x_n - y_n\|^2 + \|(x_n + v_n) - (y_n + w_n)\|^2] \le (1 + \frac{h}{2\kappa})(\|x_k^* - y_n\|^2 + \|(x_k^* + v_k^*) - (y_n + w_n)\|^2)$$
$$+ \frac{2\kappa}{h}(\|\mathbb{E}_\alpha[x_n] - x_k^*\|^2 + \|\mathbb{E}_\alpha[x_n + v_n] - (x_n^* + v_n^*)\|^2)$$
$$+ \mathbb{E}_\alpha \|x_n - x_n^*\|^2 + \mathbb{E}_\alpha \|(x_n + v_n) - (x_n^* + v_n^*)\|^2$$

Furthermore, we can take expectation on $\omega$ and use the contraction of Underdamped Langevin dynamics:

$$A_n^2 \le (1 + \frac{h}{2\kappa}) e^{-\frac{h}{\kappa}} A_{n-1}^2 + \frac{2\kappa}{h}(\mathbb{E}\|\mathbb{E}_\alpha x_n - x_n^*\|^2 + \mathbb{E}\|\mathbb{E}_\alpha[x_n + v_n] - (x_n^* + v_n^*)\|^2)$$
$$+ \mathbb{E}\|x_n^* - x_n\|^2 + \mathbb{E}\|(x_n + v_n) - (x_n^* + v_n^*)\|^2$$
$$\le e^{-\frac{h}{2\kappa}} A_{n-1}^2 + \frac{2\kappa}{h}(3\mathbb{E}\|\mathbb{E}_\alpha x_n - x_n^*\|^2 + 2\mathbb{E}\|\mathbb{E}_\alpha v_n - v_n^*\|^2)$$
$$+ 3\mathbb{E}\|x_n - x_n^*\|^2 + 2\mathbb{E}\|v_n - v_n^*\|^2$$

When $h < \frac{1}{2}$, $u = \frac{1}{M}$ and $m = 1$, use the results in 2):

$$A_n^2 \le e^{-\frac{h}{2\kappa}} A_{n-1}^2 + 8250 \left[ (\kappa h^7 + h^4) \mathbb{E}\|v_{n-1}\|^2 + (\kappa^{-1} h^8 + \kappa^{-2} h^4) \mathbb{E}\|\nabla f(x_{n-1})\|^2 + (\kappa^{-1} h^5 + h^7) \right]$$

Our next step is to bound $\mathbb{E}\|v_{n-1}\|^2$ and $\mathbb{E}\|\nabla f(x_{n-1})\|^2$. First for Underdamped Langevin dynamics with $f$ satisfying Assumption 1.1, it's easy to compute that:

$$\mathbb{E}\|w_{n-1}\|^2 = d/M$$
$$\mathbb{E}\|\nabla f(y_{n-1})\|^2 = \frac{1}{\int e^{-f(x)} dx} \int |\nabla f(x)|^2 e^{-f(x)} dx$$
$$= -\frac{1}{\int e^{-f(x)} dx} \int (\nabla f(x))^T \nabla e^{-f(x)} dx$$
$$= \frac{1}{\int e^{-f(x)} dx} \int \Delta f(x) e^{-f(x)} dx$$
$$\le \|\Delta f(x)\|_\infty \le Md$$

Therefore, we have

$$\mathbb{E}\|v_{n-1}\|^2 \le 2d/M + 2\mathbb{E}\|v_{n-1} - w_{n-1}\|^2 \le 2d/M + 4A_{n-1}^2$$
$$\mathbb{E}\|\nabla f(x_{n-1})\|^2 \le 2Md + 2M^2 \mathbb{E}\|x_{n-1} - y_{n-1}\|^2 \le 2Md + 2M^2 A_{n-1}^2$$

Plug the upper bounds into our previous result:

$$A_n^2 \leq e^{-\frac{h}{2\kappa}}A_{n-1}^2 + 8250\left[(\kappa h^7 + h^4)(2d/M + 4A_{n-1}^2) + (\kappa^{-1}h^8 + \kappa^{-2}h^4)(2Md + 2M^2A_{n-1}^2) + (\kappa^{-1}h^5 + h^7)\right]$$

$$\leq \left[1 - \frac{h}{2\kappa} + \frac{h^2}{8\kappa^2} + 49500(h^4 + \kappa h^7)\right]A_{n-1}^2 + 41250d(h^7 + \kappa^{-1}h^4)$$

If we choose $(x_{n-1}, v_{n-1}) \sim \pi_h^*(x,v)$ and $(y_{n-1}, w_{n-1}) \sim \pi^*(x,v)$ such that

$$A_{n-1}^2 = \min_{X \sim \pi_h^*,\, Y \sim \pi^*} \mathbb{E}\|X - Y\|^2$$

Then we have

$$W_2(\pi, \pi_h)^2 \leq A_{n-1}^2 \leq \frac{82500h^3(\kappa h^3 + 1)d}{1 - \frac{h}{4\kappa} - 99000h^3\kappa(1 + \kappa h^3)}$$

We can see that $W_2(\pi, \pi_h) \to 0$ as $h \to 0$. Furthermore, as $h \to 0$, $W_2(\pi, \pi_h) < O(h^{\frac{3}{2}})$. $\qquad\square$

## 7.2  Proofs for Section 3.2

Before proving Theorem 3.2, we require some preliminary estimtes from [40], that we present below. First, let $(y_n, w_n)$ be the solution of Underdamped Langevin dynamics evaluated at $t = \sum_{k=1}^n \gamma_k$ with initial value $(y_0, w_0)$. $(x_n, v_n)$ is the $n$th iterates in the (RULMC) algorithm with initial value $(x_0, v_0)$. $(x_n^*(t), v_n^*(t))$ is the solution of Underdamped Langevin dynamics with initial value $(x_{n-1}, v_{n-1})$ and $(x_n^*, v_n^*) = (x_{n-1}^*(\gamma_n), v_{n-1}^*(\gamma_n))$. Then, we have the following results from Lemma 2 in [40]. When $\gamma_{n+1} < \frac{1}{2}$ and $u = \frac{1}{M}$, we have:

$$\mathbb{E}\left\|\mathbb{E}_\alpha x_{n+1} - x_{n+1}^*\right\|^2 \leq 45(\gamma_{n+1}^{10}\mathbb{E}\|v_n\|^2 + M^{-2}\gamma_{n+1}^{12}\mathbb{E}\|\nabla f(x_n)\|^2 + M^{-1}d\gamma_{n+1}^{11})$$

$$\mathbb{E}\left\|x_{n+1} - x_{n+1}^*\right\|^2 \leq 1800(\gamma_{n+1}^6\mathbb{E}\|v_n\|^2 + M^{-2}\gamma_{n+1}^4\mathbb{E}\|\nabla f(x_n)\|^2 + M^{-1}d\gamma_{n+1}^7)$$

$$\mathbb{E}\left\|\mathbb{E}_\alpha v_{n+1} - v_{n+1}^*\right\|^2 \leq 45(\gamma_{n+1}^8\mathbb{E}\|v_n\|^2 + M^{-2}\gamma_{n+1}^{10}\mathbb{E}\|\nabla f(x_n)\|^2 + M^{-1}d\gamma_{n+1}^9)$$

$$\mathbb{E}\left\|v_{n+1} - v_{n+1}^*\right\|^2 \leq 1300(\gamma_{n+1}^4\mathbb{E}\|v_n\|^2 + M^{-2}\gamma_{n+1}^4\mathbb{E}\|\nabla f(x_n)\|^2 + M^{-1}d\gamma_{n+1}^5)$$

*Proof of Theorem 3.2.* Define $A_n^2 = \mathbb{E}[\|x_n - y_n\|^2 + \|(x_n + v_n) - (y_n + w_n)\|^2]$. From the proof of proposition 3.1, we know that

$$A_n^2 \leq \left[1 - \frac{\gamma_n}{2\kappa} + \frac{\gamma_n^2}{8\kappa^2} + 49500(\gamma_n^4 + \kappa\gamma_n^7)\right]A_{n-1}^2 + 41250d(\gamma_n^7 + \kappa^{-1}\gamma_n^4)$$

When time step $h$ is a constant, apply the inequality repeatedly to get

$$A_n^2 \leq \left[1 - \frac{h}{2\kappa} + \frac{h^2}{8\kappa^2} + 49500(h^4 + \kappa h^7)\right]^k A_0^2 + \frac{82500h^3(\kappa h^3 + 1)d}{1 - \frac{h}{4\kappa} - 99000h^3\kappa(1 + \kappa h^3)}$$

Denote $\nu_n$ to be the density function of $x_n$, then $W_2(\nu_n, \pi) \leq A_n$. By choosing $\gamma_n = h \sim O(\epsilon^{\frac{2}{3}})$, we can guarantee that $W_2(\nu_n, \pi) < \epsilon\sqrt{\frac{d}{m}}$ for all $n > K \sim \tilde{O}(\epsilon^{-\frac{2}{3}})$.

When the time step $\gamma_n$ is variant, the inequality we correspondingly have

$$A_n^2 \leq \left[1 - \frac{\gamma_n}{2\kappa} + \frac{\gamma_n^2}{8\kappa^2} + 49500(\gamma_n^4 + \kappa\gamma_n^7)\right]A_{n-1}^2 + 41250d(\gamma_n^7 + \kappa^{-1}\gamma_n^4)$$

When $\gamma_n < 1$, $\frac{\gamma_n^2}{8\kappa^2} < \frac{\gamma_n}{8\kappa}$. When $\gamma_n < (\frac{99000}{8\kappa^2}) < 24\kappa^{-\frac{2}{3}}$, we have $49500(\gamma_n^4 + \kappa\gamma_n^7) < \frac{\gamma_n}{8\kappa}$. Similarly, when $\gamma_n < 1$, we have $41250d(\gamma_n^7 + \kappa^{-1}\gamma_n^4) < 82500d\gamma_n^4$. Therefore, when $\gamma_n < \min\{1/2, 24\kappa^{-\frac{2}{3}}\}$, we have

$$A_n^2 < (1 - \frac{\gamma_n}{4\kappa})A_{n-1}^2 + 82500d\gamma_n^4$$

If we choose $\gamma_n = \frac{16\kappa}{32\kappa^{\frac{5}{3}} + (n - K_1)^+}$, where $K_1$ is the smallest integer such that

$$A_{K_1}^2 < (1 - \frac{4}{\kappa^{\frac{5}{3}}})^{K_1}A_0^2 + (82500)d\frac{1}{2\kappa} < 2\frac{82500d}{\kappa}$$

Then we claim that for all $n \geq K_1$, we have

$$A_n^2 < \frac{82500(16)^4 d\kappa^4}{(32\kappa^{\frac{5}{3}} + n - K_1)^3}$$

The claim can be proved by induction: Assume that the claim hold for $A_n^2$ and denote $b = 32\kappa^{\frac{5}{3}} + n - K_1$, then

$$
\begin{aligned}
A_{n+1}^2 &< (1 - \frac{4}{1+b})\frac{82500(16)^4 d\kappa^4}{b^3} + \frac{82500d(16)^4\kappa^4}{(b+1)^4} \\
&= \frac{82500(16)^4 d\kappa^4}{(b+1)^3}\left[\frac{(b-3)(b+1)^2}{b^3} + \frac{1}{b+1}\right] \\
&< \frac{82500(16)^4 d\kappa^4}{(b+1)^3} \\
&= \frac{82500(16)^4 d\kappa^4}{(32\kappa^{\frac{5}{3}} + n + 1 - K_1)^3}
\end{aligned}
$$

Therefore, under our choice of time step $(\gamma_n)$, we can guarantee $W_2(\nu_n, \pi) < \epsilon\sqrt{\frac{d}{m}}$ for all $n > K \sim O(\epsilon^{-\frac{2}{3}})$. Compared to the running time of constant step size RULMC, vanishing step size help reduce the factor $\log(\frac{1}{\epsilon})$ in the guarantees. □

Now we introduce the CLT statement for another sampling algorithm related to (RULMC) and give a complete proof of the statement. The proof of Remark 6 can be done in the same way. In the following theorem, we give a central limit result with specific choice of weights and time step-size. The Euler-discretization of the underdamped Langevin diffusion (which we call as KLMC, following [7]) is given by the following algorithm:

$$
\begin{aligned}
x_{n+1} &= x_n + \frac{1 - e^{-2\gamma_{n+1}}}{2}v_n - \frac{u}{2}(\gamma_{n+1} - \frac{1 - e^{-2\gamma_{n+1}}}{2})\nabla f(x_n) + \sqrt{u}\sigma_{n+1}^{(1)}U_{n+1}^{(1)} \\
v_{n+1} &= v_n e^{-2\gamma_{n+1}} - u\frac{1 - e^{-2\gamma_{n+1}}}{2}\nabla f(x_n) + 2\sqrt{u}\sigma_{n+1}^{(2)}U_{n+1}^{(2)}
\end{aligned} \tag{KLMC}
$$

where $\{\gamma_n\}$ are the time steps. $\sigma_n^{(1)}$ and $\sigma_n^{(2)}$ are positive with $\sigma_n^{(1)^2} = \gamma_n + \frac{1-e^{-4\gamma_n}}{4} - (1 - e^{-2\gamma_n})$, $\sigma_n^{(2)^2} = \frac{1-e^{-4\gamma_n}}{4}$. $\{(U_n^{(1)}, U_n^{(2)})\}_n$ are independent Centered Gaussian random vectors in $\mathbb{R}^{2d}$ with $(U_n^{(1)}, U_n^{(2)}) \sim \mathcal{N}(0, \sigma_n^2 I_d)$ and $\sigma_n^2 = \frac{1+e^{-4\gamma_n}-2e^{-2\gamma_n}}{4\sigma_n^{(1)}\sigma_n^{(2)}}$. Numerical integration with the above sampler follows the same steps as described in Section 3.2. We now provide the following CLT.

**Theorem 5.** *Assume potential function $f$ satisfies Assumption 1.1. Let $\{(x_k, v_k)\}$ and $\{(U_k^{(1)}, U_k^{(2)})\}$ be the same as what we have in the (KLMC) algorithm and the time step-size $\{\gamma_k\}$ is non-increasing and $\lim_k(\gamma_{k-1} - \gamma_k)/\gamma_k^4 = 0$. If $\lim_n(1/\sqrt{\Gamma_n^{(3)}})\sum_{k=1}^n \gamma_k^4 = \hat{\gamma} \in (0, +\infty]$ and $\lim_n \Gamma_n^{(4)} = +\infty$, then for all $\phi \in \mathcal{C}^3$ with $D^2\phi$, $D^3\phi$ and $D^4\phi$ bounded and Lipschitz and $\sup_{(x,v)\in\mathbb{R}^{2d}} |\nabla\phi(x)|^2/V(x,v) < +\infty$, we have*

$$
\begin{aligned}
\frac{\Gamma_n}{\Gamma_n^{(4)}}\nu_n^\gamma(\mathcal{L}\phi) &\to \mathcal{N}(\rho, \frac{10}{3}u\hat{\gamma}^{-2}\int_{\mathbb{R}^d}|\nabla\phi(x)|\pi(dx)) && \text{if } \hat{\gamma} < +\infty \\
\frac{\Gamma_n}{\Gamma_n^{(4)}}\nu_n^\gamma(\mathcal{L}\phi) &\to \rho && \text{if } \hat{\gamma} = +\infty,
\end{aligned}
$$

*where*

$$
\begin{aligned}
\rho &= \frac{u}{6}\int\int\langle D^3\phi(x); \nabla f(x), v^{\otimes 2}\rangle\nu(dx, dv) + \frac{u}{24}\int\int\langle D^3 f(x); \nabla\phi(x), v^{\otimes 2}\rangle\nu(dx, dv) \\
&\quad + \frac{u}{12}\int\int(D^2\phi D^2 f)(x)v^{\otimes 2}\nu(dx, dv) - \frac{1}{12}\int\int D^4\phi(x)v^{\otimes 4}\nu(dx, dv) \\
&\quad - \frac{u^2}{24}\int\langle D^2 f(x); \nabla\phi(x), \nabla f(x)\rangle\pi(dx).
\end{aligned}
$$

In the following context we'll discuss the weak convergence of empirical measure $\nu_n^\eta$ and build a central limit theorem under certain assumptions.

1) (Lyapunov Conditions) The underdamped Langevin dynamics can be rewritten as

$$dY_t = b(Y_t)dt + \sigma(Y_t)dW_t$$

where $Y_t = [X_t, V_t]^T$, $b(y) = b(x,v) = [v, -2v - u\nabla f(x)]^T$, $\sigma(y) = 2\sqrt{u}[0_d, I_d]^T$ for all $x, v \in \mathbb{R}^d$. $\{W_t\}$ is a 2d-dimensional Brownian motion.
The Lyapunov condition is similar to the one that's introduced in Lamberton's paper.

**Assumption** $(\mathcal{L}_{V,\infty})$: There's a $\mathcal{C}^2$ function $V : \mathbb{R}^{2d} \to [v_*, +\infty)$ for some $v_* > 0$ satisfying the following conditions:
a) $\|D^2V\|_\infty = \sup_{(x,v)^T \in \mathbb{R}^{2d}} \|D^2V(x,v)\|_{op} < +\infty$ and $\lim_{|(x,v)| \to +\infty} V(x,v) = +\infty$;
b) $|\nabla V(x,v)|^2 + |b(x,v)|^2 \leq c_V V(x,v)$ for all $(x,v)^T \in \mathbb{R}^{2d}$ and some $c_V > 0$;
c) $\langle \nabla V(x,v), b(x,v) \rangle \leq -\alpha V(x,v) + \beta$ for some $\alpha > 0$ and $\beta \in \mathbb{R}$.

**Assumption** $(\mathcal{L}_{V,p})$: There's a $\mathcal{C}^2$ function $V : \mathbb{R}^{2d} \to [v_*, +\infty)$ for some $v_* > 0$ satisfying for some $p \geq 1$:
a) $\|D^2V\|_\infty = \sup_{(x,v)^T \in \mathbb{R}^{2d}} \|D^2V(x,v)\|_{op} < +\infty$ and $\lim_{|(x,v)| \to +\infty} V(x,v) = +\infty$;
b) $|\nabla V(x,v)|^2 + |b(x,v)|^2 + \text{Tr}(\sigma(x,v)\sigma(x,v)^T) \leq c_V V(x,v)$ for all $(x,v)^T \in \mathbb{R}^{2d}$ and some $c_V > 0$;
c) $\langle \nabla V(x,v), b(x,v) \rangle + \lambda_p \text{Tr}(\sigma(x,v)\sigma(x,v)^T) \leq -\alpha V(x,v) + \beta$ for some $\alpha > 0$ and $\beta \in \mathbb{R}$, where $\lambda_p = \frac{1}{2}\lambda_{D^2V+(p-1)(\nabla V \otimes \nabla V)/V}$.

**Remark 10.** *1) We can show that: $(\mathcal{L}_{V,p'}) \implies (\mathcal{L}_{V,p})$ if $p' \geq p \geq 1$. Especially $(\mathcal{L}_{V,\infty}) \implies (\mathcal{L}_{V,p})$ for all $p \geq 1$.*

*2) If we choose $b$ and $\sigma$ the same as those in the Underdamped Langevin dynamics, then $(\mathcal{L}_{V,\infty})$ is almost the same as assumption 7.1. We can instantly obtain that assumption 7.1 implies $(\mathcal{L}_{V,\infty})$. Therefore, according to lemma 4, assumption 1.1 implies $(\mathcal{L}_{V,\infty})$.*

2) (Tightness Result) We now establish the almost sure tightness of the weighted empirical measures. The filtration $\{\mathcal{F}_n\}$ we consider is $\mathcal{F}_n = \sigma(Y_0, (U_1^{(1)}, U_1^{(2)}), \cdots, (U_n^{(1)}, U_n^{(2)}))$.

**Lemma 5.** *(a) If $(\mathcal{L}_{V,1})$ holds, then for every $a \geq \frac{1}{2}$,*

$$|V^a(Y_{n+1}) - V^a(Y_n)| \leq c_a \sqrt{\gamma_{n+1}} V^a(Y_n)(1 + |U_{n+1}^{(1)}|^{2a} + |U_{n+1}^{(2)}|^{2a})$$

*(b) If $(\mathcal{L}_{V,p})$ holds for some $p \geq 1$, then there exists real numbers $\tilde{\alpha} > 0$ and $\tilde{\beta}$ and $n_0 \in \mathbb{N}$ such that*

$$\mathbb{E}[V^p(Y_{n+1})|\mathcal{F}_n] \leq V^(Y_n) + \gamma_{n+1}V^{p-1}(Y_n)(\tilde{\beta} - \tilde{\alpha}V(Y_n)), \quad \forall n \geq n_0$$

*and furthermore*

$$\sup_{n \in \mathbb{N}} \mathbb{E}[V^p(Y_n)] < +\infty$$

*Proof of Lemma 5.* (a) Using mean value theorem and $(\mathcal{L}_{V,1})$:

$$|V^a(Y_{n+1}) - V^a(Y_n)| = a|V^{a-1}(\xi_{n+1})\langle \nabla V(\xi_{n+1}), Y_{n+1} - Y_n \rangle|$$
$$\leq CV^{a-\frac{1}{2}}(\xi_{n+1})|Y_{n+1} - Y_n|$$

From $(\mathcal{L}_{V,1})$-b) we get that $\nabla\sqrt{V}$ is bounded, i.e $\sqrt{V}$ is Lipschitz with parameter $[\sqrt{V}]_1$. Hence

$$V^{a-\frac{1}{2}}(\xi_{n+1}) \leq (\sqrt{V}(Y_n) + [\sqrt{V}]_1|Y_{n+1} - Y_n|)^{2a-1}$$
$$\leq 2^{2a-1}\left(V^{a-\frac{1}{2}}(Y_n) + [\sqrt{V}]_1^{2a-1}|Y_{n+1} - Y_n|^{2a-1}\right)$$

Meanwhile,

$$|Y_{n+1} - Y_n|^2 = \left| \begin{bmatrix} \frac{1-e^{-2\gamma_{n+1}}}{2}v_n - \frac{u}{2}(\gamma_{n+1} - \frac{1-e^{-2\gamma_{n+1}}}{2})\nabla f(x_n) + \sqrt{u}\sigma_{n+1}^{(1)}U_{n+1}^{(1)} \\ -2\frac{1-e^{-2\gamma_{n+1}}}{2}v_n - u\frac{1-e^{-2\gamma_{n+1}}}{2}\nabla f(x_n) + 2\sqrt{u}\sigma_{n+1}^{(2)}U_{n+1}^{(2)} \end{bmatrix} \right|^2$$
$$\leq 15(\frac{1-e^{-2\gamma_{n+1}}}{2})^2|v_n^2| + [\frac{3u^2}{4}(\gamma_{n+1} - \frac{1-e^{-2\gamma_{n+1}}}{2})^2 + 3u^2(\frac{1-e^{-2\gamma_{n+1}}}{2})^2]|\nabla f(x_n)|^2$$

$$+ 3u\sigma_{n+1}^{(1)}{}^2 |U_{n+1}^{(1)}|^2 + 12u\sigma_{n+1}^{(2)}{}^2 |U_{n+1}^{(2)}|^2$$

Since $\gamma_n \to 0$ as $\nu \to \infty$ and $\frac{1-e^{-2\gamma_n}}{2} \sim O(\gamma_n)$, $\gamma_n - \frac{1-e^{-2\gamma_n}}{2} \sim O(\gamma_n^2)$, $\sigma_n^{(1)} \sim O(\gamma_n^{\frac{3}{2}})$ and $\sigma_n^{(2)} \sim O(\gamma_n^{\frac{1}{2}})$, there exist $C_1, C_2, C_3 > 0$ such that

$$|Y_{n+1} - Y_n|^2 \leq C_1 \left[ \gamma_{n+1}^2(|v_n|^2 + |\nabla f(x_n)|^2) + \gamma_{n+1}(|U_{n+1}^{(1)}|^2 + |U_{n+1}^{(2)}|^2) \right]$$

$$\leq C_2 \left[ \gamma_{n+1}^2 V(Y_n) + \gamma_{n+1}(|U_{n+1}^{(1)}|^2 + |U_{n+1}^{(2)}|^2 + 1) \right]$$

$$\implies |Y_{n+1} - Y_n| \leq C_3 \sqrt{\gamma_{n+1}} \sqrt{V(Y_n)}(|U_{n+1}^{(1)}| + |U_{n+1}^{(2)}| + 1)$$

Combining our estimations, since $a \geq 1/2$. we get

$$|V^a(Y_{n+1}) - V^a(Y_n)| \leq C2^{2a-1} \left( V^{a-\frac{1}{2}}(Y_n) + [\sqrt{V}]_1^{2a-1} |Y_{n+1} - Y_n|^{2a-1} \right) |Y_{n+1} - Y_n|$$

$$\leq c_a' \left( \sqrt{\gamma_{n+1}} V^a(Y_n)(|U_{n+1}^{(1)}| + |U_{n+1}^{(2)}| + 1) + \gamma_{n+1}^a V^a(Y_n)(|U_{n+1}^{(1)}| + |U_{n+1}^{(2)}| + 1)^{2a} \right)$$

$$\leq c_a \sqrt{\gamma_{n+1}} V^a(Y_n)(|U_{n+1}^{(1)}|^{2a} + |U_{n+1}^{(2)}|^{2a} + 1)$$

(b) We Taylor expand $V^p(Y_{n+1})$ at $Y_n$:

$$V^p(Y_{n+1}) = V^p(Y_n) + pV^{p-1}(Y_n)\langle \nabla V(Y_n), Y_{n+1} - Y_n \rangle + \frac{1}{2} D^2(V^p)(\xi_{n+1})(Y_{n+1} - Y_n)^{\otimes 2}$$

Since $D^2(V^p) = pV^{p-1}D^2V + p(p-1)V^{p-1}\nabla V \nabla V^T$, by the definition of $\lambda_p$:

$$D^2(V^p)(\xi_{n+1})(Y_{n+1} - Y_n)^{\otimes 2} \leq 2p\lambda_p V^{p-1}(\xi_{n+1})|Y_{n+1} - Y_n|^2$$

Therefore

$$V^p(Y_{n+1}) \leq V^p(Y_n) + pV^{p-1}(Y_n)\langle \nabla V(Y_n), Y_{n+1} - Y_n \rangle + p\lambda_p V^{p-1}(\xi_{n+1})|Y_{n+1} - Y_n|$$

When $p = 1$, take conditional expectation on $\mathcal{F}_n$:

$$\mathbb{E}[V(Y_{n+1})|\mathcal{F}_n] \leq V(Y_n) + \frac{1-e^{-2\gamma_{n+1}}}{2}\langle \nabla V(x_n, v_n), b(x_n, v_n) \rangle$$

$$- \frac{u}{2}(\gamma_{n+1} - \frac{1-e^{-2\gamma_{n+1}}}{2})\nabla_x V(x_n, v_n) \cdot \nabla f(x_n)$$

$$+ \lambda_1(\frac{1-e^{-2\gamma_{n+1}}}{2})^2[5|v_n|^2 + u^2|\nabla f(x_n)|^2 + 4u\nabla f(x_n) \cdot v_n]$$

$$- \frac{u}{2}\frac{1-e^{-2\gamma_{n+1}}}{2}(\gamma_{n+1} - \frac{1-e^{-2\gamma_{n+1}}}{2})\nabla f(x_n) \cdot v_n$$

$$+ \frac{u^2}{4}(\gamma_{n+1} - \frac{1-e^{-2\gamma_{n+1}}}{2})^2|\nabla f(x_n)|^2 + u(\sigma_{n+1}^{(1)}{}^2 + 4\sigma_{n+1}^{(2)}{}^2)d$$

There exists $n_0 \in \mathbb{N}$ such that for all $n \geq n_0$

$$\frac{1-e^{-2\gamma_{n+1}}}{2}\langle \nabla V(x_n, v_n), b(x_n, v_n) \rangle \leq \gamma_{n+1}(-\alpha V(Y_n) + \beta), \quad \text{for some } \alpha > 0, \beta \in \mathbb{R};$$

$$-\frac{u}{2}(\gamma_{n+1} - \frac{1-e^{-2\gamma_{n+1}}}{2})\nabla_x V(x_n, v_n) \cdot \nabla f(x_n) \leq C\gamma_{n+1}^2(|\nabla V(Y_n)|^2 + |b(Y_n)|^2) \leq C\gamma_{n+1}^2 V(Y_n);$$

$$\lambda_1(\frac{1-e^{-2\gamma_{n+1}}}{2})^2[5|v_n|^2 + u^2|\nabla f(x_n)|^2 + 4u\nabla f(x_n) \cdot v_n] \leq C\gamma_{n+1}^2|b(Y_n)|^2 \leq C\gamma_{n+1}^2 V(Y_n);$$

$$-\frac{u}{2}\frac{1-e^{-2\gamma_{n+1}}}{2}(\gamma_{n+1} - \frac{1-e^{-2\gamma_{n+1}}}{2})\nabla f(x_n) \cdot v_n \leq C\gamma_{n+1}^3|b(Y_n)|^2 \leq C\gamma_{n+1}^3 V(Y_n);$$

$$\frac{u^2}{4}(\gamma_{n+1} - \frac{1-e^{-2\gamma_{n+1}}}{2})^2|\nabla f(x_n)|^2 \leq C\gamma_{n+1}^4|b(Y_n)|^2 \leq C\gamma_{n+1}^4 V(Y_n);$$

$$u(\sigma_{n+1}^{(1)}{}^2 + 4\sigma_{n+1}^{(2)}{}^2)d \leq C\gamma_{n+1}.$$

Therefore, for all $n \geq n_0$, there exist $\tilde{\alpha} > 0$, $\tilde{\beta} \in \mathbb{R}$ such that

$$\mathbb{E}[V(Y_{n+1})|\mathcal{F}_n] \leq V(Y_n)(1 - \alpha\gamma_{n+1} + C(2\gamma_{n+1}^2 + \gamma_{n+1}^3 + \gamma_{n+1}^4)) + \gamma_{n+1}(\beta + C)$$
$$\leq V(Y_n)(1 - \tilde{\alpha}\gamma_{n+1}) + \tilde{\beta}\gamma_{n+1}$$

and $1 - \tilde{\alpha}\gamma_{n+1} > 0$. This leads to

$$\mathbb{E}[V(Y_{n+1})] \leq \mathbb{E}[V(Y_n)](1 - \tilde{\alpha}\gamma_{n+1}) + \tilde{\beta}\gamma_{n+1}$$

We could use induction to prove:

$$\sup_{n \geq n_0} \mathbb{E}[V(Y_n)] \leq \frac{\tilde{\beta}}{\tilde{\alpha}} \vee \mathbb{E}[V(Y_{n_0})]$$

Assume now $p > 1$. Due to $(\mathcal{L}_{V,p})$-b), we derive that $\sqrt{V}$ is Lipschitz with parameter $[\sqrt{V}]_1$. Consequently,

$$V^{p-1}(\xi_{n+1}) = \sqrt{V}^{2(p-1)}(\xi_{n+1}) \leq \left(\sqrt{V}(Y_n) + [\sqrt{V}]_1|Y_{n+1} - Y_n|\right)^{2(p-1)}$$
$$\leq \begin{cases} V^{p-1}(Y_n) + ([\sqrt{V}]_1|Y_{n+1} - Y_n|)^{2(p-1)} & \text{if } 2(p-1) \leq 1, \\ V^{p-1}(Y_n) + C\left(V^{(2p-3)/2}(Y_n)|Y_{n+1} - Y_n| + |Y_{n+1} - Y_n|^{2(p-1)}\right) & \text{if } 2(p-1) > 1. \end{cases}$$

Using the fact we've proved in part a):

$$|Y_{n+1} - Y_n|^2 \leq C_2\left[\gamma_{n+1}^2 V(Y_n) + \gamma_{n+1}(|U_{n+1}^{(1)}|^2 + |U_{n+1}^{(2)}|^2 + 1)\right]$$

We derive

$$V^{p-1}(\xi_{n+1})|Y_{n+1} - Y_n|^2 \leq V^{p-1}(Y_n)|Y_{n+1} - Y_n|^2 + C\gamma_{n+1}^{p \wedge \frac{3}{2}} V^p(Y_n)(1 + |U_{n+1}^{(1)}|^{2p} + |U_{n+1}^{(2)}|^{2p})$$

Then we take conditional expectation:

$$\mathbb{E}[V^p(Y_{n+1})|\mathcal{F}_n] \leq V^p(Y_n) + pV^{p-1}\frac{1 - e^{-2\gamma_{n+1}}}{2}\langle \nabla V(x_n, v_n), b(x_n, v_n)\rangle$$
$$- pV^{p-1}(Y_n)\frac{u}{2}(\gamma_{n+1} - \frac{1 - e^{-2\gamma_{n+1}}}{2})\nabla_x V(x_n, v_n) \cdot \nabla f(x_n)$$
$$+ CV^{p-1}(Y_n)\left[\gamma_{n+1}^2 V(Y_n) + \gamma_{n+1}(|U_{n+1}^{(1)}|^2 + |U_{n+1}^{(2)}|^2 + 1)\right]$$
$$+ C\gamma_{n+1}^{p \wedge \frac{3}{2}} V^p(Y_n)(1 + |U_{n+1}^{(1)}|^{2p} + |U_{n+1}^{(2)}|^{2p})$$

There exists $n_0 \in \mathbb{N}$ such that for all $n \geq n_0$

$$\frac{1 - e^{-2\gamma_{n+1}}}{2}\langle \nabla V(x_n, v_n), b(x_n, v_n)\rangle \leq \gamma_{n+1}(-\alpha V(Y_n) + \beta), \quad \text{for some } \alpha > 0, \beta \in \mathbb{R};$$

$$-\frac{u}{2}(\gamma_{n+1} - \frac{1 - e^{-2\gamma_{n+1}}}{2})\nabla_x V(x_n, v_n) \cdot \nabla f(x_n) \leq C\gamma_{n+1}^2(|\nabla V(Y_n)|^2 + |b(Y_n)|^2) \leq C\gamma_{n+1}^2 V(Y_n).$$

Since $\gamma_n^{p \wedge \frac{3}{2}}, \gamma_n^2 \sim o(\gamma_n)$, there exists $\tilde{\alpha} > 0, \tilde{\beta} \in \mathbb{R}$, such that for all $n \geq n_0$:

$$\mathbb{E}[V^p(Y_n)|\mathcal{F}_n] \leq V^p(Y_n) + \gamma_{n+1} V^{p-1}(Y_n)(\tilde{\beta} - \tilde{\alpha}V(Y_n))$$

Same as the proof for $p = 1$, we can show

$$\sup_{n \in \mathbb{N}} \mathbb{E}[V^p(Y_n)] < +\infty$$

$\square$

**Theorem 6.** *Let $p \in [0, +\infty)$. Assume $(\mathcal{L}_{V,p})$, If there exists $s \in (0,1]$ such that*

$$\sum_{n \geq 1} \frac{1}{H_n}(\Delta \frac{\eta_n}{\gamma_n})_+ < +\infty \quad \text{and} \quad \sum_{n \geq 1} (\frac{\eta_n}{H_n\sqrt{\gamma_n}})^{1+s} < +\infty$$

*then*

$$\mathbb{P}(d\omega) - a.s \quad \sup_{n \in \mathbb{N}} \nu_n^\eta(\omega, V^{p/(1+s)}) < +\infty$$

Based on Lemma 5, the proof of Theorem 6 immediately follows, by using the same steps in the proof of Theorem 4 in [21]. Hence we don't replicate the proof here.

3) (Identification of the limit)

**Theorem 7** (Echeverrría-Weiss Theorem). *Let $E$ be a locally compact Polish space and $\mathcal{L}$ a linear operator satisfying the positive maximum principle. Assume that its domain $\mathcal{D}(A)$ is an algebra everywhere dense in $(\mathcal{C}_0(E), \| \ \|_\infty)$ containing a sequence $(f_n)_{n \in \mathbb{N}}$ satisfying*

$$\sup_{n \in \mathbb{N}} (\|f_n\|_\infty + \|\mathcal{L}f_n\|_\infty) < +\infty, \ \ \forall x \in E, \ \ f_n(x) \to 1 \ \ and \ \ \mathcal{L}f_n(x) \to 0.$$

*If a distribution on $(E, \mathcal{B}(E))$ satisfies $\int_E \mathcal{L}f \, d\nu = 0$ for every $f \in \mathcal{D}(A)$, then there exists a stationary solution for the martingale problem $(\mathcal{L}, \nu)$ (this means that there exists a stationary continuous-time homogeneous Markov process with infinitesimal generator $\mathcal{L}$ and invariant distribution $\nu$).*

**Lemma 6.** *If the potential function $f$ is Gradient Lipschitz and strongly convex, then the generator of kinetic, $\mathcal{L}$, satisfies the assumptions of the Echeverrría-Weiss theorem.*

*Proof of lemma 6.* First it's well-known that the infinitesimal generator of a Fellerian semigroup satisfies the maximum principle. We can choose our $f_n(y) = \phi(y/n)$ for any $y \in \mathbb{R}^{2d}$ where $\phi$ is $\mathcal{C}^2$ with compact support and $\phi(0) = 1$. It's easy to check that $\forall y \in \mathbb{R}^{2d}$, $f_n(y) \to 0$ and $\mathcal{L}f_n(y) \to 0$. It's also straightforward that $\sup_{n \in \mathbb{N}} \|f_n\|_\infty < +\infty$. The last thing to check is $\sup_{n \in \mathbb{N}} \|\mathcal{L}f_n\|_\infty < +\infty$. Since $\mathcal{L}$ can also be written as $b(x,v) \cdot [\nabla_x, \ \nabla_v]^T + 2u\Delta_v$ and we've shown that under our assumptions on $f$, $(\mathcal{L}_{V,\infty})$ is satisfied, we have the Lyapunov function $V(y) = O(|y|^2)$ and $|b(x,v)| \leq C(1 + |(x,v)|)$. Therefore we get $\sup_{n \in \mathbb{N}} \|\mathcal{L}f_n\|_\infty < +\infty$. $\qquad\square$

**Theorem 8.** *Assume that $f$ is gradient Lipschitz and strongly convex. Assume also*

$$\lim_n \frac{1}{H_n} \sum_{k=1}^n |\Delta \frac{\eta_n}{\gamma_n}| = 0 \ \ and \ \ \sum_{n \geq 1} (\frac{\eta_n}{\sqrt{\gamma_n} H_n})^2 < +\infty$$

*Let $a \geq \frac{1}{2}$. Assume $\sup_n \nu_n^\eta(V^a) < +\infty \ \mathbb{P} - a.s$. If $a < 1$, assume also that $\sum_{n \geq 1} \eta_n \gamma_n / H_n < +\infty$. Then $\mathbb{P} - a.s$, every limiting distribution $\nu_\infty(\omega, dx)$ of the sequence $(\nu_n^\eta(\omega, dx))$ is an invariant distribution of the underdamped Langevin dynamics introduced in the previous section.*

The proof of theorem 8 follows immediately from Theorem 7, lemma 6, lemma 7 and lemma 8.

**Lemma 7.** *Under the assumptions in theorem 8, then for every bounded Lipschitz continuous function $g : \mathbb{R}^{2d} \to \mathbb{R}$,*

$$\mathbb{P} - a.s \ \ \lim_n \frac{1}{H_n} \sum_{k=1}^n \frac{\eta_k}{\gamma_k} \mathbb{E}[g(Y_k) - g(Y_{k-1})|\mathcal{F}_{k-1}] = 0$$

*Proof of lemma 7.* Setting $\eta_0/\gamma_0 = 0$ gives

$$\frac{1}{H_n} \sum_{k=1}^n \mathbb{E}[g(Y_k) - g(Y_{k-1})|\mathcal{F}_{k-1}] = \frac{1}{H_n} \sum_{k=1}^n \frac{\eta_k}{\gamma_k}(g(Y_k) - g(Y_{k-1})) - \frac{1}{H_n} \sum_{k=1}^n \frac{\eta_k}{\gamma_k}(g(Y_k) - \mathbb{E}[g(Y_k)|\mathcal{F}_{k-1}]).$$

As $g$ is bounded, it follows by lemma 3-b) in Lamberton's paper that

$$\mathbb{P} - a.s \ \ \lim_n \frac{1}{H_n} \sum_{k=1}^n (g(Y_k) - g(Y_{k-1})) = 0.$$

Then

$$\frac{1}{H_n} \sum_{k=1}^n \frac{\eta_k}{\gamma_k}(g(Y_k) - \mathbb{E}[g(Y_k)|\mathcal{F}_{k-1}])$$

will converge to 0 once the martingale

$$M_n^g := \sum_{k=1}^n \frac{\eta_k}{\gamma_k H_k}(g(Y_k) - \mathbb{E}[g(Y_k)|\mathcal{F}_{k-1}])$$

converge a.s in $\mathbb{R}$.

$$\mathbb{E}\langle M_n^g\rangle_\infty = \sum_{n\geq 1}(\frac{\eta_n}{\gamma_n H_n})^2\,\|g(Y_n) - \mathbb{E}[g(Y_n)|\mathcal{F}_{n-1}]\|_2^2 \leq \sum_{n\geq 1}(\frac{\eta_n}{\gamma_n H_n})^2\,\|g(Y_n) - g(Y_{n-1})\|_2^2$$

$$\leq [f]_1^2 \sum_{n\geq 1}(\frac{\eta_n}{\gamma_n H_n})^2\,\|Y_n - Y_{n-1}\|_2^2$$

Since $(\mathcal{L}_{V,1})$ holds under our assumptions on $f$ and by lemma 2-b)

$$\|Y_n - Y_{n-1}\|_2^2 \leq C'\mathbb{E}[\gamma_n^2 V(Y_{n-1}) + (2d+1)\gamma_n] \leq C\gamma_n$$

Therefore

$$\mathbb{E}\langle M_n^g\rangle_\infty \leq C\sum_{n\geq 1}(\frac{\eta_n}{\sqrt{\gamma_n}H_n})^2 < +\infty$$

$\square$

**Lemma 8.** *Under the assumptions in theorem* 8, *then for every* $g \in \mathcal{C}^2(\mathbb{R}^{2d})$ *with compact support,*

$$\lim_n \left( \frac{1}{H_n}\sum_{k=1}^n \frac{\eta_k}{\gamma_k}\mathbb{E}[g(Y_k) - g(Y_{k-1})|\mathcal{F}_{k-1}] - \nu_n^\eta(\mathcal{L}g) \right) = 0 \ a.s$$

*Proof of lemma* 8. Setting $R_2(y_1,y_2) := g(y_2) - g(y_1) - \langle \nabla g(y_1), y_2 - y_1 \rangle - \frac{1}{2}D^2 g(y_1)(y_2-y_1)^{\otimes 2}$, we obtain for every $k \in \mathbb{N}$,

$$g(Y_k) - g(Y_{k-1}) = \langle \nabla g(Y_{k-1}), Y_k - Y_{k-1}\rangle + \frac{1}{2}D^2 g(Y_{k-1})(Y_k - Y_{k-1})^{\otimes 2} + R_2(Y_{k-1}, Y_k)$$

$$= \nabla_x g(x_{k-1}, v_{k-1}) \cdot [\frac{1-e^{-2\gamma_k}}{2}v_{k-1} - \frac{u}{2}(\gamma_k - \frac{1-e^{-2\gamma_k}}{2})\nabla f(x_{k-1}) + \sqrt{u}\sigma_k^{(1)}U_k^{(1)}]$$

$$+ \nabla_v g(x_{k-1}, v_{k-1}) \cdot [-2\frac{1-e^{-2\gamma_k}}{2}v_{k-1} - u\frac{1-e^{-2\gamma_k}}{2}\nabla f(x_{k-1}) + 2\sqrt{u}\sigma_k^{(2)}U_k^{(2)}]$$

$$+ \frac{1}{2}D_x^2 g(x_{k-1}, v_{k-1})[\frac{1-e^{-2\gamma_k}}{2}v_{k-1} - \frac{u}{2}(\gamma_k - \frac{1-e^{-2\gamma_k}}{2})\nabla f(x_{k-1}) + \sqrt{u}\sigma_k^{(1)}U_k^{(1)}]^{\otimes 2}$$

$$+ \frac{1}{2}D_v^2 g(x_{k-1}, v_{k-1})[-2\frac{1-e^{-2\gamma_k}}{2}v_{k-1} - u\frac{1-e^{-2\gamma_k}}{2}\nabla f(x_{k-1}) + 2\sqrt{u}\sigma_k^{(2)}U_k^{(2)}]^{\otimes 2}$$

$$+ \langle D_{xv}g(x_{k-1}, v_{k-1}); \frac{1-e^{-2\gamma_k}}{2}v_{k-1} - \frac{u}{2}(\gamma_k - \frac{1-e^{-2\gamma_k}}{2})\nabla f(x_{k-1}) + \sqrt{u}\sigma_k^{(1)}U_k^{(1)},$$

$$- 2\frac{1-e^{-2\gamma_k}}{2}v_{k-1} - u\frac{1-e^{-2\gamma_k}}{2}\nabla f(x_{k-1}) + 2\sqrt{u}\sigma_k^{(2)}U_k^{(2)}\rangle$$

$$+ R_2(Y_{k-1}, Y_k)$$

$$= \gamma_k \mathcal{L}g(Y_{k-1}) - (\gamma_k - \frac{1-e^{-2\gamma_k}}{2})\nabla_x g(Y_{k-1})\cdot v_{k-1} - \frac{u}{2}(\gamma_k - \frac{1-e^{-2\gamma_k}}{2})\nabla_x g(Y_{k-1})\cdot \nabla f(x_{k-1})$$

$$+ 2(\gamma_k - \frac{1-e^{-2\gamma_k}}{2})\nabla_v g(Y_{k-1})\cdot v_{k-1} + u(\gamma_k - \frac{1-e^{-2\gamma_k}}{2})\nabla_v g(Y_{k-1})\cdot \nabla f(x_{k-1})$$

$$+ \sqrt{u}\sigma_k^{(1)}\nabla g(Y_{k-1})\cdot U_k^{(1)} + 2\sqrt{u}\sigma_k^{(2)}\nabla g(Y_{k-1})\cdot U_k^{(2)}$$

$$+ \frac{1}{2}(\frac{1-e^{-2\gamma_k}}{2})^2 D_x^2 g(Y_{k-1})v_{k-1}^{\otimes 2} + \frac{u^2}{8}(\gamma_k - \frac{1-e^{-2\gamma_k}}{2})^2 D_x^2 g(Y_{k-1})\nabla f(x_{k-1})^{\otimes 2}$$

$$+ \frac{u}{2}\sigma_k^{(1)2} D_x^2 g(Y_{k-1})U_k^{(1)\otimes 2} - \frac{u}{2}\frac{1-e^{-2\gamma_k}}{2}(\gamma_k - \frac{1-e^{-2\gamma_k}}{2})\langle D_x^2 g(Y_{k-1}); v_{k-1}, \nabla f(x_{k-1})\rangle$$

$$+ \sqrt{u}\frac{1-e^{-2\gamma_k}}{2}\sigma_k^{(1)}\langle D_x^2 g(Y_{k-1}); v_{k-1}, U_k^{(1)}\rangle$$

$$- \frac{u^{3/2}}{2}(\gamma_k - \frac{1-e^{-2\gamma_k}}{2})\sigma_k^{(1)}\langle D_x^2 g(Y_{k-1}); \nabla f(x_{k-1}), U_k^{(1)}\rangle$$

$$+ 2(\frac{1-e^{-2\gamma_k}}{2})^2 D_v^2 g(Y_{k-1})v_{k-1}^{\otimes 2} + \frac{u^2}{2}(\frac{1-e^{-2\gamma_k}}{2})^2 D_v^2 g(Y_{k-1})\nabla f(x_{k-1})^{\otimes 2}$$

$$+ 2u\left(\sigma_k^{(2)^2} D_v^2 g(Y_{k-1}) U_k^{(2)\otimes 2} - \gamma_k \mathbb{E}[D_v^2 g(Y_{k-1}) U_k^{(2)\otimes 2} | \mathcal{F}_{k-1}]\right)$$

$$+ 2u(\frac{1-e^{-2\gamma_k}}{2})^2 \langle D_v^2 g(Y_{k-1}); v_{k-1}, \nabla f(x_{k-1})\rangle - 4\sqrt{u}\frac{1-e^{-2\gamma_k}}{2}\sigma_k^{(2)}\langle D_v^2 g(Y_{k-1}); v_{k-1}, U_k^{(2)}\rangle$$

$$- 2u^{3/2}\frac{1-e^{-2\gamma_k}}{2}\sigma_k^{(2)}\langle D_v^2 g(Y_{k-1}); \nabla f(x_{k-1}), U_k^{(2)}\rangle + R_2(Y_{k-1}, Y_k)$$

Take conditional expectation:

$$\mathbb{E}[g(Y_k) - g(Y_{k-1})|\mathcal{F}_{k-1}] - \gamma_k \mathcal{L}g(Y_{k-1}) = -(\gamma_k - \frac{1-e^{-2\gamma_k}}{2})\nabla_x g(Y_{k-1}) \cdot v_{k-1}$$

$$- \frac{u}{2}(\gamma_k - \frac{1-e^{-2\gamma_k}}{2})\nabla_x g(Y_{k-1}) \cdot \nabla f(x_{k-1})$$

$$+ 2(\gamma_k - \frac{1-e^{-2\gamma_k}}{2})\nabla_v g(Y_{k-1}) \cdot v_{k-1}$$

$$+ u(\gamma_k - \frac{1-e^{-2\gamma_k}}{2})\nabla_v g(Y_{k-1}) \cdot \nabla f(x_{k-1})$$

$$+ \frac{1}{2}(\frac{1-e^{-2\gamma_k}}{2})^2 D_x^2 g(Y_{k-1}) v_{k-1}^{\otimes 2}$$

$$+ \frac{u^2}{8}(\gamma_k - \frac{1-e^{-2\gamma_k}}{2})^2 D_x^2 g(Y_{k-1}) \nabla f(x_{k-1})^{\otimes 2}$$

$$+ \frac{u}{2}\sigma_k^{(1)^2}\Delta_x g(Y_{k-1})$$

$$- \frac{u}{2}\frac{1-e^{-2\gamma_k}}{2}(\gamma_k - \frac{1-e^{-2\gamma_k}}{2})\langle D_x^2 g(Y_{k-1}); v_{k-1}, \nabla f(x_{k-1})\rangle$$

$$+ 2(\frac{1-e^{-2\gamma_k}}{2})^2 D_v^2 g(Y_{k-1}) v_{k-1}^{\otimes 2} + \frac{u^2}{2}(\frac{1-e^{-2\gamma_k}}{2})^2 D_v^2 g(Y_{k-1}) \nabla f(x_{k-1})^{\otimes 2}$$

$$+ 2u\left(\sigma_k^{(2)^2} - \gamma_k\right)\Delta_v g(Y_{k-1})$$

$$+ 2u(\frac{1-e^{-2\gamma_k}}{2})^2 \langle D_v^2 g(Y_{k-1}); v_{k-1}, \nabla f(x_{k-1})\rangle$$

$$+ \mathbb{E}[R_2(Y_{k-1}, Y_k)|\mathcal{F}_{k-1}]$$

Observe that for all the terms, except for $R_2(Y_{k-1}, Y_k)$, on the right hand side of the equation, their coefficients are of order $O(\gamma_k^2)$ or $o(\gamma_k^2)$. Furthermore, $\nabla g$ and $D^2 g$ are bounded because $g$ is $\mathcal{C}^2$ and compact supported. Since $(\mathcal{L}_{V,\infty})$ is satisfied under our assumptions, $\sup_{n\in\mathbb{N}} \mathbb{E}[|v_n|^2 + |\nabla f(x_n)|^2] < C \sup_{n\in\mathbb{N}} \mathbb{E}[V(Y_n)]) < +\infty$. Therefore, we obtain that as $n \to 0$,

$$\frac{1}{H_n}\sum_{k=1}^n \frac{\eta_k}{\gamma_k}\mathbb{E}[g(Y_k) - g(Y_{k-1})|\mathcal{F}_{k-1}] - \eta_k \mathcal{L}g(Y_{k-1}) - \frac{\eta_k}{\gamma_k}\mathbb{E}[R_2(Y_{k-1}, Y_k)|\mathcal{F}_{k-1}] \to 0$$

because $\frac{1}{H_n}\sum_{k=1}^n \eta_k \gamma_k \to 0$ as $n \to 0$.

Now we deal with $\mathbb{E}[R_2(Y_{k-1}, Y_k)|\mathcal{F}_{k-1}]$. For any $x, y \in \mathbb{R}^{2d}$, define

$$r_2(x, y) := \frac{1}{2}\sup_{t\in(0,1)}\left\|D^2 g(x + t(y-x)) - D^2 g(x)\right\|$$

It's easy to see that $r_2$ is a bounded continuous function on $\mathbb{R}^d \times \mathbb{R}^d$, $r_2(x, x) = 0$ and

$$|R_2(x, y)| \leq r_2(x, y)|x - y|^2$$

Therefore we obtain

$$\frac{\eta_k}{\gamma_k}|\mathbb{E}[R_2(Y_{k-1}, Y_k)|\mathcal{F}_{k-1}]| \leq C\left(\eta_k \gamma_k \|r_2\|_\infty V(Y_{k-1}) + (2d+1)\eta_k\mathbb{E}[r_2(Y_{k-1}, Y_k)(|U_k^{(1)}|^2 + |U_k^{(2)}|^2)|\mathcal{F}_{k-1}]\right)$$

If $a \geq 1, \mathbb{P} - a.s$

$$\frac{1}{H_n}\sum_{k=1}^n C\eta_k \gamma_k \|r_2\|_\infty V(Y_{k-1}) \leq C'\frac{1}{H_n}\sum_{k=1}^n \eta_k \gamma_k V(Y_{k-1}) \to 0 \quad \text{as } \sup_{n\in\mathbb{N}}\nu_n^\eta(V) < +\infty \text{ and } \gamma_n \to 0$$

If $a \in [1/2, 1)$, the same limit follows from the Kronecker lemma and $\sum_{n \geq 1} \eta_n \gamma_n / H_n < +\infty$. Meanwhile,

$$J(\gamma, x, v) = \int_{\mathbb{R}^d \times \mathbb{R}^d} r_2((x, v), (x', v'))(|r_1|^2 + |r_2|^2)\mu(dr_1, dr_2)$$

**where**

$$(x', v') = (x + \frac{1 - e^{-2\gamma}}{2}v - \frac{u}{2}(\gamma - \frac{1 - e^{-2\gamma}}{2})\nabla f(x) + \sqrt{u}\sigma^{(1)}r_1, e^{-2\gamma}v - u\frac{1 - e^{-2\gamma}}{2}\nabla f(x) + 2\sqrt{u}\sigma^{(2)}r_2)$$

$$\sigma^{(1)} = \left(\gamma + \frac{1 - e^{-4\gamma}}{4} - (1 - e^{-2\gamma})\right)^{1/2}, \quad \sigma^{(2)} = \left(\frac{1 - e^{-4\gamma}}{4}\right)^{1/2}$$

and $(U^{(1)}, U^{(2)}) \sim \mu = \mathcal{N}(0, \frac{1 + e^{-4\gamma} - 2e^{-2\gamma}}{4\sigma^{(1)}\sigma^{(2)}}I_{2d})$

We can see that $J$ is a bounded continuous function on $\mathbb{R}_+ \times \mathbb{R}^d \times \mathbb{R}^d$ and $J(0, x, v) = 0$. Since $\lim_{|y| \to \infty} V(y) = +\infty$. We can also write

$$(2d + 1)\eta_k \mathbb{E}[r_2(Y_{k-1}, Y_k)(|U_k^{(1)}|^2 + |U_k^{(2)}|^2)|\mathcal{F}_{k-1}] = \eta_k V^a((x_{k-1}, v_{k-1}))\theta((x_{k-1}, v_{k-1}))J(\gamma_k, x_{k-1}, v_{k-1})$$

where $\lim_{|(x_{k-1}, v_{k-1})| \to \infty} \theta((x_{k-1}, v_{k-1})) = 0$ It remains to show that

$$\mathbb{P} - a.s \quad \lim_n \frac{1}{H_n} \sum_{k=1}^n \eta_k V^a((x_{k-1}, v_{k-1}))\theta((x_{k-1}, v_{k-1}))J(\gamma_k, x_{k-1}, v_{k-1}) = 0$$

For a fixed number $A > 0$, $J$ is uniformly continuous on $[0, \sup_n \gamma_n] \times \bar{B}_{2d}(0, A)$, then

$$J(\gamma_k, x_{k-1}, v_{k-1})1_{|(x_{k-1}, v_{k-1})| \leq A} \to 0 \quad \mathbb{P} - a.s.$$

And $V^a((x_{k-1}, v_{k-1}))\theta((x_{k-1}, v_{k-1}))$ is bounded on $\bar{B}_{2d}(0, A)$. Therefore

$$\mathbb{P} - a, s \quad \lim_n \frac{1}{H_n} \sum_{k=1}^n \eta_k V^a((x_{k-1}, v_{k-1}))\theta((x_{k-1}, v_{k-1}))J(\gamma_k, x_{k-1}, v_{k-1})1_{|(x_{k-1}, v_{k-1})| \leq A} = 0$$

On the other hand side

$$\limsup_n \frac{1}{H_n} \sum_{k=1}^n \eta_k V^a((x_{k-1}, v_{k-1}))\theta((x_{k-1}, v_{k-1}))J(\gamma_k, x_{k-1}, v_{k-1})1_{|(x_{k-1}, v_{k-1})| > A}$$

$$\leq \sup_{|(x, v)| > A} |\theta(x, v)| \|J\|_\infty \sup_n \nu_n^\eta(V^a) \to 0 \quad \text{as } A \to +\infty$$

So taking $A \to +\infty$ completes the proof. $\qquad \square$

**Theorem 9.** *Let $p \in [1, +\infty)$. Assume $(\mathcal{L}_{V,p})$. Let $s \in (0, 1]$. Assume that*

$$\sum_{n \geq 1} \frac{1}{H_n}\left(\Delta\frac{\eta_n}{\gamma_n}\right)_+ < +\infty. \quad \lim_n \frac{1}{H_n} \sum_{k=1}^n |\Delta\frac{\eta_k}{\gamma_k}| = 0 \text{ and } \sum_{n \geq 1} \left(\frac{\eta_n}{H_n\sqrt{\gamma_n}}\right)^{1+s} < +\infty$$

*(a) Then*

$$\mathbb{P} - a.s \quad \sup_{n \in \mathbb{N}} \nu_n^\eta(\omega, V^{p/(1+s)}) < +\infty.$$

*(b) When $p \leq 1 + s$, assume also $\sum_{n \geq 1} \eta_n \gamma_n / H_n < +\infty$. Then with probability 1, any weak limit of the sequence $(\nu_m^\eta)$ is an invariant distribution of the underdamped Langevin dynamics.*

Theorem 9 follows directly from theorem 6 and theorem 8.

*Proof of Theorem 5.* First we try to decompose $\sum_{k=1}^{n} \gamma_k \mathcal{L}\phi(x_{k-1})$ using Taylor expansion.

$$\phi(x_k) = \phi(x_{k-1}) + \nabla\phi(x_{k-1}) \cdot (x_k - x_{k-1}) + \frac{1}{2}D^2\phi(x_{k-1})(x_k - x_{k-1})^{\otimes 2} + R_2^{(k)}$$

where $R_2^{(k)} = \phi(x_k) - \phi(x_{k-1}) - \nabla\phi(x_{k-1}) \cdot (x_k - x_{k-1}) - \frac{1}{2}D^2\phi(x_{k-1})(x_k - x_{k-1})^{\otimes 2}$. We can plug our discretization into the equation and obtain:

$$\begin{aligned}
\phi(x_k) - \phi(x_{k-1}) = {}& \gamma_k \mathcal{L}\phi(x_{k-1}) - (\gamma_k - \frac{1 - e^{-2\gamma_k}}{2})v_{k-1} \cdot \nabla\phi(x_{k-1}) \\
& - \frac{u}{2}(\gamma_k - \frac{1 - e^{-2\gamma_k}}{2})\nabla f(x_{k-1}) \cdot \nabla\phi(x_{k-1}) + \sqrt{u}\sigma_k^{(1)}\nabla\phi(x_{k-1}) \cdot U_k^{(1)} \\
& + \frac{1}{2}(\frac{1 - e^{-2\gamma_k}}{2})^2 D^2\phi(x_{k-1})v_{k-1}^{\otimes 2} + \frac{u}{2}{\sigma_k^{(1)}}^2 D^2\phi(x_{k-1})U_k^{(1)\otimes 2} \\
& + \frac{u^2}{8}(\gamma_k - \frac{1 - e^{-2\gamma_k}}{2})^2 D^2\phi(x_{k-1})\nabla f(x_{k-1})^{\otimes 2} \\
& - \frac{u}{2}\frac{1 - e^{-2\gamma_k}}{2}(\gamma_k - \frac{1 - e^{-2\gamma_k}}{2})\langle D^2\phi(x_{k-1}); v_{k-1}, \nabla f(x_{k-1})\rangle \\
& + \sqrt{u}\sigma_k^{(1)}\frac{1 - e^{-2\gamma_k}}{2}\langle D^2\phi(x_{k-1}); v_{k-1}, U_k^{(1)}\rangle \\
& - \frac{u^{3/2}}{2}\sigma_k^{(1)}(\gamma_k - \frac{1 - e^{-2\gamma_k}}{2})\langle D^2\phi(x_{k-1}); \nabla f(x_{k-1}), U_k^{(1)}\rangle \\
& + R_2^{(k)}
\end{aligned}$$

where

$$\begin{aligned}
R_2^{(k)} = {}& \frac{1}{6}(\frac{1 - e^{-2\gamma_k}}{2})^3 D^3\phi(x_{k-1})v_{k-1}^{\otimes 3} - \frac{u}{4}(\frac{1 - e^{-2\gamma_k}}{2})^2(\gamma_k - \frac{1 - e^{-2\gamma_k}}{2})\langle D^3\phi(x_{k-1}); v_{k-1}^{\otimes 2}, \nabla f(x_{k-1})\rangle \\
& + \frac{\sqrt{u}}{2}\sigma_k^{(1)}(\frac{1 - e^{-2\gamma_k}}{2})^2\langle D^3\phi(x_{k-1}); v_{k-1}^{\otimes 2}, U_k^{(1)}\rangle + \frac{u}{2}{\sigma_k^{(1)}}^2\frac{1 - e^{-2\gamma_k}}{2}\langle D^3\phi(x_{k-1}); v_{k-1}, U_k^{(1)\otimes 2}\rangle \\
& + \frac{1}{24}(\frac{1 - e^{-2\gamma_k}}{2})^4 D^4\phi(x_{k-1})v_{k-1}^{\otimes 4} + r^{(k)}
\end{aligned}$$

Since $f$ is gradient Lipschitz and strongly convex, we've shown $(\mathcal{L}_{V,\infty})$ holds. Using $(\mathcal{L}_{V,\infty})$ the fact that $D^4\phi$ is bounded and Lipschitz, we can show there exists a constant $C > 0$ such that

$$|r_k| \le C\gamma_k^{9/2}V^2(x_{k-1}, v_{k-1})$$

Apply theorem 6 for $p = 4$ and $s = 1$, we have $\sup_n \nu_n^\gamma(V^2) < +\infty$ $\mathbb{P} - a.s.$ Therefore

$$\frac{1}{\Gamma_n^{(4)}}\sum_{k=1}^{n} r^{(k)} \to 0 \qquad \text{in } \mathbb{L}^1$$

In the following proof, we will use $o(\gamma_k^4)$ to denote the sum of those terms $b_k$ such that $\frac{1}{\Gamma_n^{(4)}}\sum_{k=1}^{n} b_k \to 0$ $\mathbb{P} - a.s.$ According to our decomposition, we can pull out polynomials of $\gamma_k$ from factors $\frac{1 - e^{-2\gamma_k}}{2}$, $\gamma_k - \frac{1 - e^{-2\gamma_k}}{2}$ and $\sigma_k^{(1)}$ so that the terms left could be included in $o(\gamma_k^4)$. Then we obtain

$$\begin{aligned}
\sum_{k=1}^{n} \gamma_k \mathcal{L}\phi(x_{k-1}) = \sum_{k=1}^{n} \Big\{ {}& [\phi(x_k) - \phi(x_{k-1})] + (\gamma_k^2 - \frac{2}{3}\gamma_k^3 + \frac{1}{3}\gamma_k^4)v_{k-1} \cdot \nabla\phi(x_{k-1}) \\
& + \frac{u}{2}(\gamma_k^2 - \frac{2}{3}\gamma_k^3 + \frac{1}{3}\gamma_k^4)\nabla f(x_{k-1}) \cdot \nabla\phi(x_{k-1}) \\
& - \frac{2\sqrt{3u}}{3}\gamma_k^{\frac{3}{2}}\nabla\phi(x_{k-1}) \cdot U_k^{(1)} \\
& - \frac{1}{2}(\gamma_k^2 - 2\gamma_k^3 + \frac{7}{3}\gamma_k^4)D^2\phi(x_{k-1})v_{k-1}^{\otimes 2}
\end{aligned}$$

$$-\frac{u^2}{8}\gamma_k^4 D^2\phi(x_{k-1})\nabla f(x_{k-1})^{\otimes 2}$$

$$-\frac{u}{2}(\frac{4}{3}\gamma_k^3 - 2\gamma_k^4)D^2\phi(x_{k-1})U_k^{(1)\otimes 2}$$

$$+\frac{u}{2}(\gamma_k^3 - \frac{5}{3}\gamma_k^4)\langle D^2\phi(x_{k-1}); v_{k-1}, \nabla f(x_{k-1})\rangle$$

$$-\frac{1}{6}(\gamma_k^3 - 3\gamma_k^4)D^3\phi(x_{k-1})v_{k-1}^{\otimes 3}$$

$$+\frac{u}{4}\gamma_k^4\langle D^3\phi(x_{k-1}); v_{k-1}^{\otimes 2}, \nabla f(x_{k-1})\rangle$$

$$-\frac{2u}{3}\gamma_k^4\langle D^3\phi(x_{k-1}); v_{k-1}, U_k^{(1)\otimes 2}\rangle$$

$$\left. -\frac{1}{24}\gamma_k^4 D^4\phi(x_{k-1})v_{k-1}^{\otimes 4} + o(\gamma_k^4)\right\}$$

$$:= Z_n^{(0)} + Z_n^{(2)} + Z_n^{(3)} + Z_n^{(4)} + N_n + r_n$$

where

$$Z_n^{(0)} = \phi(x_n) - \phi(x_0)$$

$$Z_n^{(2)} = \sum_{k=1}^n \gamma_k^2 \left[v_{k-1}\cdot\nabla\phi(x_{k-1}) + \frac{u}{2}\nabla f(x_{k-1})\cdot\nabla\phi(x_{k-1}) - \frac{1}{2}D^2\phi(x_{k-1})v_{k-1}^{\otimes 2}\right]$$

$$:= \sum_{k=1}^n \gamma_k^2 z_{k-1}^{(2)}$$

$$Z_n^{(3)} = \sum_{k=1}^n \gamma_k^3 \left[-\frac{2}{3}v_{k-1}\cdot\nabla\phi(x_{k-1}) - \frac{u}{3}\nabla f(x_{k-1})\cdot\nabla\phi(x_{k-1}) + D^2\phi(x_{k-1})v_{k-1}^2\right.$$

$$\left. -\frac{2u}{3}D^2\phi(x_{k-1})U_k^{(1)\otimes 2} + \frac{u}{2}\langle D^2\phi(x_{k-1}); v_{k-1}, \nabla f(x_{k-1})\rangle - \frac{1}{6}D^3\phi(x_{k-1})v_{k-1}^{\otimes 3}\right]$$

$$:= -\sum_{k=1}^n \gamma_k^3 z_{k-1}^{(3)}$$

$$Z_n^{(4)} = \sum_{k=1}^n \gamma_k^4 \left[\frac{1}{3}v_{k-1}\cdot\nabla\phi(x_{k-1}) + \frac{u}{6}\nabla f(x_{k-1})\cdot\nabla\phi(x_{k-1}) - \frac{7}{6}D^2\phi(x_{k-1})v_{k-1}^{\otimes 2}\right.$$

$$-\frac{u^2}{8}D^2\phi(x_{k-1})\nabla f(x_{k-1})^{\otimes 2} + uD^2\phi(x_{k-1})U_k^{(1)\otimes 2} - \frac{5u}{6}\langle D^2\phi(x_{k-1}); v_{k-1}, \nabla f(x_{k-1})\rangle$$

$$+\frac{1}{2}D^3\phi(x_{k-1})v_{k-1}^{\otimes 3} + +\frac{u}{4}\langle D^3\phi(x_{k-1}); v_{k-1}^{\otimes 2}, \nabla f(x_{k-1})\rangle - \frac{2u}{3}\langle D^3\phi(x_{k-1}); v_{k-1}, U_k^{(1)\otimes 2}\rangle$$

$$\left. -\frac{1}{24}D^4\phi(x_{k-1})v_{k-1}^{\otimes 4}\right] := \sum_{k=1}^n \gamma_k^4 z_{k-1}^{(4)}$$

$$N_n = \sum_{k=1}^n \frac{2\sqrt{3u}}{3}\gamma_k^{\frac{3}{2}}\nabla\phi(x_{k-1})\cdot U_k^{(1)}$$

$$r_n = \sum_{k=1}^n o(\gamma_k^4)$$

First, it's easy to see that $r_n/\Gamma_n^{(4)} \to 0 \ \mathbb{P} - a.s$ as $n \to +\infty$. Apply lemma 5 and we obtain $\sup_n \mathbb{E}[V(x_n, v_n)] < +\infty$. Therefore we can further obtain the tightness of sequence $\{x_n\}$ and it follows from the continuity of $\phi$ that $\{\phi(x_n)\}$ is also tight. According to the tightness, $Z_n^{(0)}/\Gamma_n^{(4)} \to 0 \ \mathbb{P} - a.s$. For $Z_n^{(4)}$, under our assumptions on $\phi$ and $f$, we can show that

$$\lim_{|(x_n, v_n)|\to+\infty} z_n^{(4)}/V^4(x_n, v_n) = 0$$

Therefor apply theorem 9 with $p = 8$, $s = 1$ and we obtain:

$$\mathbb{P} - a.s \quad Z_n^{(4)}/\Gamma_n^{(4)} \to \frac{u}{4}\int_{\mathbb{R}^{2d}} \langle D^3\phi(x); \nabla f(x), v^{\otimes 2}\rangle \nu(dx, dv) - \frac{u^2}{8}\int_{\mathbb{R}^d} D^2\phi(x)\nabla f(x)^{\otimes 2}\pi(dx)$$

$$- \frac{1}{24}\int_{\mathbb{R}^{2d}} D^4\phi(x)v^{\otimes 4}\nu(dx, dv)$$

To consider the limit of $Z_n^{(i)}/\Gamma_n^{(4)}$ for $i = 2, 3$, We first Taylor expand $\mathcal{L}\phi(x_{k-1})$ and $z_{k-1}^{(3)}$ at $x_{k-2}$:

$$\mathcal{L}\phi(x_{k-1}) = v_{k-2} \cdot \nabla\phi(x_{k-2}) + \langle D^2\phi(x_{k-2}); v_{k-2}, x_{k-1} - x_{k-2}\rangle + \nabla\phi(x_{k-2}) \cdot (v_{k-1} - v_{k-2})$$

$$+ \frac{1}{2}\langle D^3\phi(x_{k-2}); v_{k-2}, (x_{k-1} - x_{k-2})^{\otimes 2}\rangle + \langle D^2\phi(x_{k-2}); v_{k-1} - v_{k-2}, x_{k-1} - x_{k-2}\rangle$$

$$+ \frac{1}{6}\langle D^4\phi(x_{k-2}); v_{k-2}, (x_{k-1} - x_{k-2})^{\otimes 3}\rangle$$

$$+ \frac{1}{2}\langle D^3\phi(x_{k-2}); v_{k-1} - v_{k-2}, (x_{k-1} - x_{k-2})^{\otimes 2}\rangle$$

$$+ o(\gamma_k^3)$$

Plug the discretization into the Taylor expansions and preserve the "large" terms, then we obtain:

$$\mathcal{L}\phi(x_{k-1}) = \mathcal{L}\phi(x_{k-2}) + (\gamma_{k-1} - \gamma_{k-1}^2 + \frac{2}{3}\gamma_{k-1}^3)D^2\phi(x_{k-2})v_{k-2}^{\otimes 2}$$

$$- \frac{u}{2}(\gamma_{k-1}^2 - \frac{2}{3}\gamma_{k-1}^3)\langle D^2\phi(x_{k-2}); v_{k-2}, \nabla f(x_{k-2})\rangle$$

$$+ \frac{2\sqrt{3u}}{3}\gamma_{k-1}^{\frac{3}{2}}\langle D^2\phi(x_{k-2}); v_{k-2}, U_{k-1}^{(1)}\rangle$$

$$- (2\gamma_{k-1} - 2\gamma_{k-1}^2 + \frac{4}{3}\gamma_{k-1}^3)v_{k-2} \cdot \nabla\phi(x_{k-2})$$

$$- u(\gamma_{k-1} - \gamma_{k-1}^2 + \frac{2}{3}\gamma_{k-1}^3)\nabla f(x_{k-2}) \cdot \nabla\phi(x_{k-2})$$

$$+ 2\sqrt{u}\gamma_{k-1}^{\frac{1}{2}}\nabla\phi(x_{k-2}) \cdot U_{k-1}^{(2)}$$

$$+ \frac{1}{2}(\gamma_{k-1}^2 - 2\gamma_{k-1}^3)D^3\phi(x_{k-2})v_{k-2}^{\otimes 3}$$

$$+ \frac{2u}{3}\gamma_{k-1}^3\langle D^3\phi(x_{k-2}); v_{k-2}, U_{k-1}^{(1)\,\otimes 2}\rangle$$

$$+ \sqrt{u}\gamma_{k-1}^{\frac{5}{2}}\langle D^3\phi(x_{k-2}); v_{k-2}^{\otimes 2}, U_{k-1}^{(1)}\rangle$$

$$- \frac{u}{2}\gamma_{k-1}^3\langle D^3\phi(x_{k-2}); v_{k-2}^{\otimes 2}, \nabla f(x_{k-2})\rangle$$

$$- 2(\gamma_{k-2}^2 - 2\gamma_{k-1}^3)D^2\phi(x_{k-2})v_{k-2}^{\otimes 2}$$

$$- u(\gamma_{k-1}^2 - 2\gamma_{k-1}^3)\langle D^2\phi(x_{k-2}); v_{k-2}, \nabla f(x_{k-2})\rangle$$

$$+ 2\sqrt{u}\gamma_{k-1}^{\frac{3}{2}}\langle D^2\phi(x_{k-2}); v_{k-2}, U_{k-1}^{(2)}\rangle$$

$$+ u\gamma_{k-1}^3\langle D^2\phi(x_{k-2}); v_{k-2}, \nabla f(x_{k-2})\rangle$$

$$+ \frac{u^2}{2}\gamma_{k-1}^3 D^2\phi(x_{k-2})\nabla f(x_{k-2})^{\otimes 2}$$

$$- u^{\frac{3}{2}}\gamma_{k-1}^{\frac{5}{2}}\langle D^2\phi(x_{k-2}); \nabla f(x_{k-2}), U_{k-1}^{(2)}\rangle$$

$$- 2\sqrt{u}\gamma_{k-1}^{\frac{5}{2}}\langle D^2\phi(x_{k-2}); v_{k-2}, U_{k-1}^{(1)}\rangle$$

$$- u^{\frac{3}{2}}\gamma_{k-1}^{\frac{5}{2}}\langle D^2\phi(x_{k-2}); \nabla f(x_{k-2}), U_{k-1}^{(1)}\rangle$$

$$+ \langle D^2\phi(x_{k-2}); \sqrt{u}\sigma_{k-1}^{(1)}U_{k-1}^{(1)}, 2\sqrt{u}\sigma_{k-1}^{(2)}U_{k-1}^{(2)}\rangle + o(\gamma_{k-1}^3)$$

Apply theorem 9 with $p = 4, s = 1$ to the terms of order $o(V^2(x_{k-2}, v_{k-2}))$ in the decomposition. We obtain

$$\lim_n \frac{\sum_{k=2}^n \gamma_k \mathcal{L}\phi(x_{k-1})}{\Gamma_n^{(4)}} = \lim_n \frac{1}{\Gamma_n^{(4)}} \left[ \sum_{k=2}^n \gamma_k \mathcal{L}\phi(x_{k-2}) + \sum_{k=2}^n \gamma_k(\gamma_{k-1} - 3\gamma_{k-1}^2) D^2\phi(x_{k-2}) v_{k-2}^{\otimes 2} \right.$$

$$- \sum_{k=2}^n \frac{3u}{2} \gamma_k \gamma_{k-1}^2 \langle D^2\phi(x_{k-2}); \nabla f(x_{k-2}), v_{k-2} \rangle$$

$$- \sum_{k=2}^n \gamma_k(2\gamma_{k-1} - 2\gamma_{k-1}^2) \nabla\phi(x_{k-2}) \cdot v_{k-2}$$

$$- \sum_{k=2}^n u\gamma_k(\gamma_{k-1} - \gamma_{k-1}^2) \nabla f(x_{k-2}) \cdot \nabla\phi(x_{k-2})$$

$$+ \sum_{k=2}^n \frac{1}{2} \gamma_k \gamma_{k-1}^2 D^3\phi(x_{k-2}) v_{k-2}^{\otimes 3}$$

$$+ \sum_{k=2}^n 2\sqrt{u} \gamma_k \gamma_{k-1}^{\frac{1}{2}} \nabla\phi(x_{k-2}) \cdot U_{k-1}^{(2)}$$

$$\left. + \sum_{k=2}^n \gamma_k \langle D^2\phi(x_{k-2}); \sqrt{u}\sigma_{k-1}^{(1)} U_{k-1}^{(1)}, 2\sqrt{u}\sigma_{k-1}^{(2)} U_{k-1}^{(2)} \rangle \right]$$

$$+ 4u \int_{\mathbb{R}^d} \Delta\phi(x)\pi(dx) - \frac{u}{2} \int_{\mathbb{R}^{2d}} \langle D^3\phi(x); v^{\otimes 2}, \nabla f(x) \rangle \nu(dx, dv)$$

$$+ \frac{u^2}{2} \int_{\mathbb{R}^d} D^2\phi(x) \nabla f(x)^{\otimes 2} \pi(dx)$$

Since $\gamma_{k-1} - \gamma_k = o(\gamma_k^4)$, we can substitute all the $\gamma_k$ on the right hand side with $\gamma_{k-1}$ and it won't change the limits. For the last term inside the square bracket, notice that $Var(\sqrt{u}\sigma_{k-1}^{(1)} U_{k-1}^{(1)}, 2\sqrt{u}\sigma_{k-1}^{(2)} U_{k-1}^{(2)}) = \frac{u}{2}(1 + e^{-4\gamma_{k-1}} - 2e^{-2\gamma_{k-1}})I_d \sim u(2\gamma_{k-1}^2 - 4\gamma_{k-1}^3)I_d$. Therefore

$$\lim_n \sum_{k=2}^n \gamma_k \langle D^2\phi(x_{k-2}); \sqrt{u}\sigma_{k-1}^{(1)} U_{k-1}^{(1)}, 2\sqrt{u}\sigma_{k-1}^{(2)} U_{k-1}^{(2)} \rangle = \lim_n \frac{1}{\Gamma_n^{(4)}} \sum_{k=2}^n 2u\gamma_{k-1}^3 \Delta\phi(x_{k-2})$$

$$- 4u \int_{\mathbb{R}^d} \Delta\phi(x)\pi(dx)$$

We can rewrite the equation as

$$\lim_n \frac{\sum_{k=2}^n \gamma_k \mathcal{L}\phi(x_{k-1})}{\Gamma_n^{(4)}} = \lim_n \frac{1}{\Gamma_n^{(4)}} \sum_{k=2}^n \gamma_k \mathcal{L}\phi(x_{k-2}) + \lim_n \frac{1}{\Gamma_n^{(4)}} \sum_{k=2}^n 2\sqrt{u}\gamma_{k-1}^{\frac{3}{2}} \nabla\phi(x_{k-2}) \cdot U_{k-1}^{(2)}$$

$$+ \lim_n \frac{1}{\Gamma_n^{(4)}} \sum_{k=2}^n \gamma_{k-1}^2 [D^2\phi(x_{k-2}) v_{k-2}^{\otimes 2} - 2\nabla\phi(x_{k-2}) \cdot v_{k-2} - u\nabla f(x_{k-2}) \cdot \nabla\phi(x_{k-2})]$$

$$+ \lim_n \frac{1}{\Gamma_n^{(4)}} \sum_{k=2}^n \gamma_{k-1}^3 [-3D^\phi(x_{k-2}) v_{k-2}^{\otimes 2} - \frac{3u}{2} \langle D^2\phi(x_{k-2}); \nabla f(x_{k-2}), v_{k-2} \rangle$$

$$+ 2\nabla\phi(x_{k-2}) \cdot v_{k-2} + u\nabla f(x_{k-2}) \cdot \nabla\phi(x_{k-2}) + \frac{1}{2} D^3\phi(x_{k-2}) v_{k-2}^{\otimes 3} + 2u\Delta\phi(x_{k-2})]$$

$$- \frac{u}{2} \int_{\mathbb{R}^{2d}} \langle D^3\phi(x); v^{\otimes 2}, \nabla f(x) \rangle \nu(dx, dv) + \frac{u^2}{2} \int_{\mathbb{R}^d} D^2\phi(x) \nabla f(x)^{\otimes 2} \pi(dx)$$

$$= \lim_n \frac{1}{\Gamma_n^{(4)}} \sum_{k=2}^n \gamma_k \mathcal{L}\phi(x_{k-2}) + \lim_n \frac{1}{\Gamma_n^{(4)}} \sum_{k=2}^n 2\sqrt{u}\gamma_{k-1}^{\frac{3}{2}} \nabla\phi(x_{k-2}) \cdot U_{k-1}^{(2)}$$

$$+ \lim_n \frac{1}{\Gamma_n^{(4)}} (-2Z_n^{(2)} - 3Z_n^{(3)})$$

$$- \frac{u}{2} \int_{\mathbb{R}^{2d}} \langle D^3\phi(x); v^{\otimes 2}, \nabla f(x) \rangle \nu(dx, dv) + \frac{u^2}{2} \int_{\mathbb{R}^d} D^2\phi(x)\nabla f(x)^{\otimes 2} \pi(dx)$$

We can instantly get that

$$\lim_n \frac{1}{\Gamma_n^{(4)}} (2Z_2^{(n)} + 3Z_n^{(3)}) = \lim_n \frac{1}{\Gamma_n^{(4)}} \sum_{k=2}^n 2\sqrt{u} \gamma_{k-1}^{\frac{3}{2}} \nabla\phi(x_{k-2}) \cdot U_{k-1}^{(2)}$$
$$+ \frac{u^2}{2} \int_{\mathbb{R}^d} D^2\phi(x)\nabla f(x)^{\otimes 2} \pi(dx)$$
$$- \frac{u}{2} \int_{\mathbb{R}^{2d}} \langle D^3\phi(x); \nabla f(x), v^{\otimes 2} \rangle \nu(dx, dv)$$

Similarly, apply Taylor expansion to $z_{k-1}^{(2)}$ at $x_{k-2}$, we achieve:

$$\nabla f(x_{k-1}) \cdot \nabla\phi(x_{k-1}) = \nabla f(x_{k-2}) \cdot \nabla\phi(x_{k-2}) + \langle D^2 f(x_{k-2}); \nabla\phi(x_{k-2}), x_{k-1} - x_{k-2} \rangle$$
$$+ \langle D^2\phi(x_{k-2}); \nabla f(x_{k-2}), x_{k-1} - x_{k-2} \rangle$$
$$+ \frac{1}{2} D^3(\nabla f \cdot \nabla\phi)(x_{k-2})(x_{k-1} - x_{k-2})^{\otimes 2} + o(\gamma_k^2)$$

$$\frac{1}{2} D^2\phi(x_{k-1})v_{k-1}^{\otimes 2} = \frac{1}{2} D^2\phi(x_{k-2})v_{k-2}^{\otimes 2} + \frac{1}{2} \langle D^3\phi(x_{k-1}); v_{k-2}^{\otimes 2}, x_{k-1} - x_{k-2} \rangle$$
$$+ \langle D^2\phi(x_{k-2}); v_{k-2}, v_{k-1} - v_{k-2} \rangle + \frac{1}{2} D^2\phi(x_{k-2})(v_{k-1} - v_{k-2})^{\otimes 2}$$
$$+ \frac{1}{4} (\frac{1 - e^{-2\gamma_{k-1}}}{2})^2 D^4\phi(x_{k-2})v_{k-2}^{\otimes 4}$$
$$+ \frac{1}{2} \langle D^3\phi(x_{k-2}); v_{k-2}, x_{k-1} - x_{k-2}, v_{k-1} - v_{k-2} \rangle$$
$$+ \frac{1}{6} \langle D^3\phi(x_{k-2}); x_{k-1} - x_{k-2}, (v_{k-1} - v_{k-2})^{\otimes 2} \rangle + o(\gamma_k^2)$$

Simplifying the coefficients lead us to

$$\nabla f(x_{k-1}) \cdot \nabla\phi(x_{k-1}) = \nabla f(x_{k-2}) \cdot \nabla\phi(x_{k-2}) + (\gamma_{k-1} - \gamma_{k-1}^2) \langle D^2 f(x_{k-2}); \nabla\phi(x_{k-2}), v_{k-2} \rangle$$
$$- \frac{u}{2} \gamma_{k-1}^2 \langle D^2 f(x_{k-2}); \nabla\phi(x_{k-2}), \nabla f(x_{k-2}) \rangle$$
$$+ \frac{2\sqrt{3u}}{3} \gamma_{k-1}^{\frac{3}{2}} \langle D^2 f(x_{k-2}); \nabla\phi(x_{k-2}), U_{k-1}^{(1)} \rangle$$
$$+ (\gamma_{k-1} - \gamma_{k-1}^2) \langle D^2\phi(x_{k-2}); \nabla f(x_{k-2}), v_{k-2} \rangle$$
$$- \frac{u}{2} \gamma_{k-1}^2 D^2\phi(x_{k-2})\nabla f(x_{k-2})^{\otimes 2} + \sqrt{u} \gamma_{k-1}^{\frac{3}{2}} \langle D^2\phi(x_{k-2}); \nabla f(x_{k-2}), U_{k-1}^{(1)} \rangle$$
$$+ \frac{1}{2} \gamma_{k-1}^2 (D^3 f \nabla\phi + 2D^2\phi D^2 f + D^3\phi\nabla f)(x_{k-2})v_{k-2}^{\otimes 2} + o(\gamma_{k-1}^2)$$

$$\frac{1}{2} D^2\phi(x_{k-1})v_{k-1}^{\otimes 2} = \frac{1}{2} D^2\phi(x_{k-2})v_{k-2}^{\otimes 2} + \frac{1}{2}(\gamma_{k-1} - \gamma_{k-1}^2)D^3\phi(x_{k-2})v_{k-2}^{\otimes 3}$$
$$- \frac{u}{4} \gamma_{k-1}^2 \langle D^3\phi(x_{k-2}); v_{k-2}^{\otimes 2}, \nabla f(x_{k-2}) \rangle + \frac{\sqrt{u}}{2} \gamma_{k-1}^{\frac{3}{2}} \langle D^3\phi(x_{k-2}); v_{k-2}^{\otimes 2}, U_{k-1}^{(1)} \rangle$$
$$- 2(\gamma_{k-1} - \gamma_{k-1}^2)D^2\phi(x_{k-2})v_{k-2}^{\otimes 2} - u(\gamma_{k-1} - \gamma_{k-1}^2) \langle D^2\phi(x_{k-2}); \nabla f(x_{k-2}), v_{k-2} \rangle$$
$$+ 2\sqrt{u} \gamma_{k-1}^{\frac{1}{2}} \langle D^2\phi(x_{k-2}); v_{k-2}, U_{k-1}^{(2)} \rangle + 2\gamma_{k-1}^2 D^2\phi(x_{k-2})v_{k-2}^{\otimes 2}$$
$$+ \frac{u^2}{2} \gamma_{k-1}^2 D^2\phi(x_{k-2})\nabla f(x_{k-2})^{\otimes 2} + 2u(\gamma_{k-1} - 2\gamma_{k-1}^2)D^2\phi(x_{k-2})U_{k-1}^{(2)}{}^{\otimes 2}$$
$$+ 2u\gamma_{k-1}^2 \langle D^2\phi(x_{k-2}); \nabla f(x_{k-2}), v_{k-2} \rangle - 4\sqrt{u} \gamma_{k-1}^{\frac{3}{2}} \langle D^2\phi(x_{k-2}); v_{k-2}, U_{k-1}^{(2)} \rangle$$

$$- 2u^{\frac{3}{2}}\gamma_{k-1}^{\frac{3}{2}}\langle D^2\phi(x_{k-2}); \nabla f(x_{k-2}), U_{k-1}^{(2)}\rangle + \frac{1}{4}\gamma_{k-1}^2 D^4\phi(x_{k-2})v_{k-2}^{\otimes 4}$$

$$- \gamma_{k-1}^2 D^3\phi(x_{k-2})v_{k-2}^{\otimes 3} - \frac{u}{2}\gamma_{k-1}^2\langle D^3\phi(x_{k-2}); v_{k-2}^{\otimes 2}, \nabla f(x_{k-2})\rangle$$

$$+ \frac{1}{2}\langle D^3\phi(x_{k-2}); v_{k-2}, \sqrt{u}\sigma_{k-1}^{(1)}U_{k-1}^{(1)}, 2\sqrt{u}\sigma_{k-1}^{(2)}U_{k-1}^{(12)}\rangle$$

$$+ \frac{2u}{3}\gamma_{k-1}^2\langle D^3\phi(x_{k-1}); v_{k-2}; U_{k-1}^{(2)}{}^{\otimes 2}\rangle + o(\gamma_k^2)$$

Take the limits and we obtain:

$$\lim_n \frac{\sum_{k=2}^n \gamma_k^2 \nabla f(x_{k-1}) \cdot \nabla\phi(x_{k-1})}{\Gamma_n^{(4)}} = \lim_n \frac{1}{\Gamma_n^{(4)}}\left[\sum_{k=2}^n \gamma_{k-1}^2 \nabla f(x_{k-2}) \cdot \nabla\phi(x_{k-2})\right.$$

$$+ \sum_{k=2}^n \gamma_{k-1}^3 \langle D^2 f(x_{k-2}); \nabla\phi(x_{k-2}), v_{k-2}\rangle$$

$$\left.+ \sum_{k=2}^n \gamma_{k-1}^3 \langle D^2\phi(x_{k-2}); \nabla f(x_{k-2}), v_{k-2}\rangle\right]$$

$$- \frac{u}{2}\int_{\mathbb{R}^d} \langle D^2 f(x); \nabla\phi(x), \nabla f(x)\rangle\pi(dx)$$

$$- \frac{u}{2}\int_{\mathbb{R}^d} D^2\phi(x)\nabla f(x)^{\otimes 2}\pi(dx)$$

$$+ \frac{1}{2}\int_{\mathbb{R}^d} \left(D^3 f(x)\nabla\phi(x) + 2D^2 f(x)D^2\phi(x) + D^3\phi(x)\nabla f(x)\right) v^{\otimes 2}\nu(dx, dv)$$

$$\lim_n \frac{1}{2}\frac{\sum_{k=2}^n \gamma_k^2 D^2\phi(x_{k-1})v_{k-1}^{\otimes 2}}{\Gamma_n^{(4)}} = \lim_n \left\{\frac{1}{\Gamma_n^{(4)}}\sum_{k=2}^n \frac{1}{2}\gamma_{k-1}^2 D^2\phi(x_{k-2})v_{k-2}^{\otimes 2}\right.$$

$$+ \frac{1}{\Gamma_n^{(4)}}\sum_{k=2}^n \gamma_{k-1}^3 \left[\frac{1}{2}D^3\phi(x_{k-2})v_{k-2}^{\otimes 3} - 2D^2\phi(x_{k-2})v_{k-2}^{\otimes 2}\right.$$

$$\left.\left.- u\langle D^2\phi(x_{k-2}); \nabla f(x_{k-2}), v_{k-2}\rangle + 2uD^2\phi(x_{k-2})U_{k-1}^{(2)}{}^{\otimes 2}\right]\right\}$$

$$- \frac{3u}{4}\int_{\mathbb{R}^{2d}} \langle D^3\phi(x); \nabla f(x), v^{\otimes 2}\rangle\nu(dx, dv) + \frac{1}{4}\int_{\mathbb{R}^{2d}} D^4\phi(x)v^{\otimes 4}\pi(dx)$$

$$+ \frac{u^2}{2}\int_{\mathbb{R}^d} D^2\phi(x)\nabla f(x)^{\otimes 2}\pi(dx)$$

**Claim:**

a) $\lim_n \frac{1}{\Gamma_n^{(4)}}\sum_{k=1}^n \gamma_k^2 \nabla\phi(x_{k-1}) \cdot v_{k-1} = 0.$

b) $\lim_n \frac{1}{\Gamma_n^{(4)}}\sum_{k=1}^n \gamma_k^3 (\frac{u}{2}\nabla\phi(x_{k-1}) \cdot \nabla f(x_{k-1}) - \frac{1}{2}D^2\phi(x_{k-1})v_{k-1}^{\otimes 2}) = 0.$

c) $\lim_n \frac{1}{\Gamma_n^{(4)}}\sum_{k=1}^n \gamma_k^3 \nabla\phi(x_{k-1}) \cdot v_{k-1} = 0.$

We'll prove the **Claim** at the end of our proof. We can use the **Claim** and our expansion of $Z_n^{(2)}$ to find the following relation:

$$\lim_n \frac{1}{\Gamma_n^{(4)}} Z_n^{(2)} = \lim_n \frac{1}{\Gamma_n^{(4)}}\sum_{k=2}^n \gamma_k^2 \nabla\phi(x_{k-1}) \cdot v_{k-1}$$

$$+ \lim_n \frac{1}{\Gamma_n^{(4)}}\sum_{k=2}^n \gamma_k^2 \left[\frac{u}{2}\nabla f(x_{k-1}) \cdot \nabla\phi(x_{k-1}) - \frac{1}{2}D^2\phi(x_{k-1})v_{k-1}^{\otimes 2}\right]$$

$$= \lim_n \frac{1}{\Gamma_n^{(4)}} \sum_{k=2}^{n} \gamma_{k-1}^2 \left[ \frac{u}{2} \nabla f(x_{k-2}) \cdot \nabla\phi(x_{k-2}) - \frac{1}{2} D^2\phi(x_{k-2})v_{k-2}^{\otimes 2} \right]$$

$$+ \lim_n \frac{1}{\Gamma_n^{(4)}} \sum_{k=2}^{n} \gamma_{k-1}^3 \left[ \frac{u}{2} \langle D^2 f(x_{k-2}); \nabla\phi(x_{k-2}), v_{k-2} \rangle + \frac{3u}{2} \langle D^2\phi(x_{k-2}); \nabla f(x_{k-2}), v_{k-2} \rangle \right.$$

$$\left. - \frac{1}{2} D^3\phi(x_{k-2})v_{k-2}^{\otimes 3} + 2D^2\phi(x_{k-2})v_{k-2}^{\otimes 2} - 2u\Delta\phi(x_{k-2}) \right]$$

$$- \frac{u^2}{4} \int_{\mathbb{R}^d} \langle D^2 f(x); \nabla\phi(x), \nabla f(x) \rangle \pi(dx) - \frac{u^2}{4} \int_{\mathbb{R}^d} D^2\phi(x)\nabla f(x)^{\otimes 2} \pi(dx)$$

$$+ \frac{u}{4} \int_{\mathbb{R}^d} \left( D^3 f(x)\nabla\phi(x) + 2D^2 f(x)D^2\phi(x) + D^3\phi(x)\nabla f(x) \right) v^{\otimes 2}\nu(dx, dv)$$

$$+ \frac{3u}{4} \int_{\mathbb{R}^{2d}} \langle D^3\phi(x); \nabla f(x), v^{\otimes 2} \rangle \nu(dx, dv) - \frac{1}{4} \int_{\mathbb{R}^{2d}} D^4\phi(x)v^{\otimes 4}\pi(dx)$$

$$- \frac{u^2}{2} \int_{\mathbb{R}^d} D^2\phi(x)\nabla f(x)^{\otimes 2}\pi(dx)$$

$$= \lim_n \frac{1}{\Gamma_n^{(4)}} [Z_n^{(2)} + 3Z_n^{(3)}] - \frac{u^2}{4} \int_{\mathbb{R}^d} \langle D^2 f(x); \nabla\phi(x), \nabla f(x) \rangle \pi(dx)$$

$$- \frac{3u^2}{4} \int_{\mathbb{R}^d} D^2\phi(x)\nabla f(x)^{\otimes 2}\pi(dx) - \frac{1}{4} \int_{\mathbb{R}^{2d}} D^4\phi(x)v^{\otimes 4}\pi(dx)$$

$$+ \frac{u}{4} \int_{\mathbb{R}^d} \left( D^3 f(x)\nabla\phi(x) + 2D^2 f(x)D^2\phi(x) + 4D^3\phi(x)\nabla f(x) \right) v^{\otimes 2}\nu(dx, dv)$$

The last identity follows from **Claim**-a),b) and the fact that $\lim_n \frac{1}{\Gamma_n^{(4)}} \sum_{k=1}^{n} \gamma_k^3 \langle D^2 f(x_{k-1}); \nabla\phi(x_{k-1}), v_{k-1} \rangle = 0$. To prove $\lim_n \frac{1}{\Gamma_n^{(4)}} \sum_{k=1}^{n} \gamma_k^3 \langle D^2 f(x_{k-1}); \nabla\phi(x_{k-1}), v_{k-1} \rangle = 0$, we can assume $\psi$ is a new test function satisfying $\nabla\psi(x) = D^2 f(x)\nabla\phi(x)$. Then the statement follows from **Claim**-c). This could be done because $\psi$ satisfies the all assumptions on $\phi$ stated in the theorem. Therefore we obtain

$$\lim_n \frac{1}{\Gamma_n^{(4)}} Z_n^{(3)} = \frac{u^2}{12} \int_{\mathbb{R}^d} \langle D^2 f(x); \nabla\phi(x), \nabla f(x) \rangle \pi(dx) + \frac{u^2}{4} \int_{\mathbb{R}^d} D^2\phi(x)\nabla f(x)^{\otimes 2}\pi(dx)$$

$$- \frac{u}{12} \int_{\mathbb{R}^d} \left( D^3 f(x)\nabla\phi(x) + 2D^2 f(x)D^2\phi(x) + 4D^3\phi(x)\nabla f(x) \right) v^{\otimes 2}\nu(dx, dv)$$

$$+ \frac{1}{12} \int_{\mathbb{R}^{2d}} D^4\phi(x)v^{\otimes 4}\pi(dx)$$

Combine with our previous results on $2Z_n^{(2)} + 3Z_n^{(3)}$ and we obtain

$$\lim_n \frac{1}{\Gamma_n^{(4)}} [Z_n^{(2)} + Z_n^{(3)}] = \lim_n \frac{1}{\Gamma_n^{(4)}} \sum_{k=1}^{n} \sqrt{u}\gamma_k^{\frac{3}{2}} \nabla\phi(x_{k-1}) \cdot U_k^{(2)} + \frac{u^2}{8} \int_{\mathbb{R}^d} D^2\phi(x)\nabla f(x)^{\otimes 2}\pi(dx)$$

$$- \frac{u}{12} \int_{\mathbb{R}^{2d}} \langle D^3\phi(x); \nabla f(x), v^{\otimes 2} \rangle \nu(dx, dv) + \frac{u}{24} \int_{\mathbb{R}^{2d}} \langle D^3 f(x); \nabla\phi(x), v^{\otimes 2} \rangle \nu(dx, dv)$$

$$+ \frac{u}{12} \int_{\mathbb{R}^{2d}} (D^2 f D^2\phi)(x)v^{\otimes 2}\nu(dx, dv) - \frac{u^2}{24} \int_{\mathbb{R}^d} \langle D^2 f(x); \nabla\phi(x), \nabla f(x) \rangle \pi(dx)$$

$$- \frac{1}{24} \int_{\mathbb{R}^{2d}} D^4\phi(x)v^{\otimes 4}\pi(dx)$$

Then we plug this result in our original decomposition:

$$\lim_n \frac{1}{\Gamma_n^{(4)}} \sum_{k=1}^{n} \gamma_k \mathcal{L}\phi(x_{k-1}) = \lim_n \frac{1}{\Gamma_n^{(4)}} \sum_{k=1}^{n} \gamma_k^{\frac{3}{2}} \frac{2\sqrt{3}}{3} \nabla\phi(x_{k-1}) \cdot (\sqrt{u}U_k^{(1)})$$

$$+ \lim_n \frac{1}{\Gamma_n^{(4)}} \sum_{k=1}^n \sqrt{u}\gamma_k^{\frac{3}{2}} \nabla\phi(x_{k-1}) \cdot U_k^{(2)} + \frac{u^2}{8} \int_{\mathbb{R}^d} D^2\phi(x)\nabla f(x)^{\otimes 2}\pi(dx)$$

$$- \frac{u}{12} \int_{\mathbb{R}^{2d}} \langle D^3\phi(x); \nabla f(x), v^{\otimes 2}\rangle \nu(dx, dv) + \frac{u}{24} \int_{\mathbb{R}^{2d}} \langle D^3 f(x); \nabla\phi(x), v^{\otimes 2}\rangle \nu(dx, dv)$$

$$+ \frac{u}{12} \int_{\mathbb{R}^{2d}} (D^2 f D^2\phi)(x) v^{\otimes 2}\nu(dx, dv) - \frac{u^2}{24} \int_{\mathbb{R}^d} \langle D^2 f(x); \nabla\phi(x), \nabla f(x)\rangle \pi(dx)$$

$$- \frac{1}{24} \int_{\mathbb{R}^{2d}} D^4\phi(x) v^{\otimes 4}\pi(dx) + \frac{u}{4} \int_{\mathbb{R}^{2d}} \langle D^3\phi(x); \nabla f(x), v^{\otimes 2}\rangle \nu(dx, dv)$$

$$- \frac{u^2}{8} \int_{\mathbb{R}^d} D^2\phi(x)\nabla f(x)^{\otimes 2}\pi(dx) - \frac{1}{24} \int_{\mathbb{R}^{2d}} D^4\phi(x) v^{\otimes 4}\nu(dx, dv)$$

$$= \lim_n \frac{1}{\Gamma_n^{(4)}} \sum_{k=1}^n \gamma_k^{\frac{3}{2}} \nabla\phi(x_{k-1}) \cdot \left( \frac{2\sqrt{3}}{3} \sqrt{u} U_k^{(1)} + \frac{1}{2} 2\sqrt{u} U_k^{(2)} \right)$$

$$+ \frac{u}{6} \int_{\mathbb{R}^{2d}} \langle D^3\phi(x); \nabla f(x), v^{\otimes 2}\rangle \nu(dx, dv) + \frac{u}{24} \int_{\mathbb{R}^{2d}} \langle D^3 f(x); \nabla\phi(x), v^{\otimes 2}\rangle \nu(dx, dv)$$

$$+ \frac{u}{12} \int_{\mathbb{R}^{2d}} (D^2\phi D^2 f)(x) v^{\otimes 2}\nu(dx, dv) - \frac{1}{12} \int_{\mathbb{R}^{2d}} D^4\phi(x) v^{\otimes 4}\nu(dx, dv)$$

$$- \frac{u^2}{24} \int_{\mathbb{R}^d} \langle D^2 f(x); \nabla\phi(x), \nabla f(x)\rangle \pi(dx)$$

It remains to determine the normal limit. Since $(U_k^{(1)}, U_k^{(2)})$ is Gaussian in $\mathbb{R}^{2d}$ with mean zero and covariance matrix $\frac{1+e^{-4\gamma_k}-2e^{-2\gamma_k}}{4\sigma_k^{(1)}\sigma_k^{(2)}} I_d$, we can find the distribution of $U_k := (\frac{2\sqrt{3}}{3}\sqrt{u}U_k^{(1)} + \frac{1}{2}2\sqrt{u}U_k^{(2)})$. $\{U_k\}$ are independent $2d$-Gaussian Random vectors with $U_k \sim \mathcal{N}(0, \Sigma_k)$, where

$$\Sigma_k = \mathbb{E}\left[ \left( \frac{2\sqrt{3}}{3}\sqrt{u}U_k^{(1)} + \frac{1}{2}2\sqrt{u}U_k^{(2)} \right)^T \left( \frac{2\sqrt{3}}{3}\sqrt{u}U_k^{(1)} + \frac{1}{2}2\sqrt{u}U_k^{(2)} \right) \right]$$

$$= \frac{4u}{3}I_d + \frac{4u\sqrt{3}}{3} \frac{1 + e^{-4\gamma_k} - 2e^{-2\gamma_k}}{4\sigma_k^{(1)}\sigma_k^{(2)}}I_d + uI_d$$

$$\sim \frac{10}{3}uI_d + O(\gamma_k)I_d$$

Apply our weak convergence result and CLT for arrays of square-integrable martingale increments, we have that when $0 < \hat{\gamma} < +\infty$:

$$\frac{1}{\Gamma_n^{(4)}} \sum_{k=1}^n \gamma_k^{\frac{3}{2}} \nabla\phi(x_{k-1}) \cdot U_k \implies \mathcal{N}(0, \sigma^2)$$

where

$$\sigma^2 = \lim_n \frac{1}{\Gamma_n^{(4)^2}} \sum_{k=1}^n \gamma_k^3 |\nabla\phi(x_{k-1})|^2 \left( \frac{10}{3}u + O(\gamma_k) \right) = \frac{10}{3}u\hat{\gamma}^{-2} \int_{\mathbb{R}^d} |\nabla\phi(x)|^2\pi(dx)$$

In conclusion, when $\hat{\gamma} \in (0, +\infty)$:

$$\frac{\Gamma_n}{\Gamma_n^{(4)}}\nu_n^\gamma(\mathcal{L}\phi) \implies \mathcal{N}\left( \rho, \frac{10}{3}u\hat{\gamma}^{-2} \int_{\mathbb{R}^d} |\nabla\phi(x)|^2\pi(dx) \right)$$

where

$$\rho = \frac{u}{6} \int_{\mathbb{R}^{2d}} \langle D^3\phi(x); \nabla f(x), v^{\otimes 2}\rangle \nu(dx, dv) + \frac{u}{24} \int_{\mathbb{R}^{2d}} \langle D^3 f(x); \nabla\phi(x), v^{\otimes 2}\rangle \nu(dx, dv)$$

$$+ \frac{u}{12} \int_{\mathbb{R}^{2d}} (D^2\phi D^2 f)(x) v^{\otimes 2}\nu(dx, dv) - \frac{1}{12} \int_{\mathbb{R}^{2d}} D^4\phi(x) v^{\otimes 4}\nu(dx, dv)$$

$$-\frac{u^2}{24}\int_{\mathbb{R}^d}\langle D^2 f(x); \nabla\phi(x), \nabla f(x)\rangle\pi(dx)$$

When $\hat{\gamma}=0$,

$$\frac{\Gamma_n}{\sqrt{\Gamma_n^{(3)}}}\nu_n^\gamma(\mathcal{L}\phi)\implies \mathcal{N}(0, \frac{10}{3}u\int_{\mathbb{R}^d}|\nabla\phi(x)|^2\pi(dx))$$

When $\hat{\gamma}=+\infty$,

$$\frac{1}{\Gamma_n^{(4)}}\sum_{k=1}^n\gamma_k^{\frac{3}{2}}\nabla\phi(x_{k-1})\cdot(\frac{2\sqrt{3}}{3}\sqrt{u}U_k^{(1)}+\frac{1}{2}2\sqrt{u}U_k^{(2)})\to 0 \quad \text{in probability}$$

Therefore when $\hat{\gamma}=+\infty$,

$$\frac{\Gamma_n}{\Gamma_n^{(4)}}\nu_n^\gamma(\mathcal{L}\phi)\to\rho \qquad \text{in probability}$$

**Proof of the claim:** First we'll show that $\frac{1}{\Gamma_n^{(3)}}\sum_{k=1}^n\gamma_k^2\mathcal{L}\phi(x_{k-1})\to 0$. We can use our decomposition of $\mathcal{L}\phi(x_{k-1})$ and obtain:

$$\sum_{k=1}^n\gamma_k^2\mathcal{L}\phi(x_{k-1})=\sum_{k=1}^n\Big\{\gamma_k\left(\phi(x_k)-\phi(x_{k-1})\right)+\gamma_k^3 v_{k-1}\cdot\nabla\phi(x_{k-1})$$
$$+\frac{u}{2}\gamma_k^3\nabla f(x_{k-1})\cdot\nabla\phi(x_{k-1})-\frac{1}{2}\gamma_k^3 D^2\phi(x_{k-1})v_{k-1}^{\otimes 2}\Big\}$$

Since $\gamma_{k-1}-\gamma_k\sim o(\gamma_k^4)$ and $\{\phi(x_n)\}$ is tight, $\frac{1}{\Gamma_n^{(3)}}\sum_{k=1}^n\gamma_k\left(\phi(x_k)-\phi(x_{k-1})\right)\to 0$. Then we can apply theorem 9 with $p=6, s=1$ and obtain

$$\frac{1}{\Gamma_n^{(3)}}\sum_{k=1}^n\gamma_k^2\mathcal{L}\phi(x_{k-1})\to\int_{\mathbb{R}^{2d}}v\cdot\nabla\phi(x)\nu(dx,dv)+\frac{u}{2}\int_{\mathbb{R}^d}\nabla\phi(x)\cdot\nabla f(x)\pi(dx)$$
$$-\frac{1}{2}\int_{\mathbb{R}^{2d}}D^2\phi(x)v^{\otimes 2}\nu(dx,dv)$$
$$=0$$

The last identity follows from integration by parts and Fubini theorem. In the same way, we can also prove $\frac{1}{\Gamma_n^{(4)}}\sum_{k=1}^n\gamma_k^3\mathcal{L}\phi(x_{k-1})\to 0$.

Next, we'll show $\lim_n\frac{1}{\Gamma_n^{(3)}}\sum_{k=1}^n\gamma_k^2(\frac{u}{2}\nabla\phi(x_{k-1})\cdot\nabla f(x_{k-1})-\frac{1}{2}D^2\phi(x_{k-1})v_{k-1}^{\otimes 2})=0$, we'll use the same trick as we did in the proof of theorem 5. We Taylor expand $\mathcal{L}\phi(x_{k-1})$ at $(x_{k-2}, v_{k-2})$:

$$\gamma_k^2\mathcal{L}\phi(x_{k-1})=\gamma_k^2\mathcal{L}\phi(x_{k-2})+\gamma_k^2(\gamma_{k-1}-\gamma_{k-1}^2)D^2\phi(x_{k-2})v_{k-2}^{\otimes 2}$$
$$-\frac{u}{2}\gamma_k^2\gamma_{k-1}^2\langle D^2\phi(x_{k-2}); v_{k-2}, \nabla f(x_{k-2})\rangle$$
$$-\gamma_k^2(2\gamma_{k-1}-2\gamma_{k-1}^2)v_{k-2}\cdot\nabla\phi(x_{k-2})$$
$$-u\gamma_k^2(\gamma_{k-1}-\gamma_{k-1}^2)\nabla f(x_{k-2})\cdot\nabla\phi(x_{k-2})$$
$$+\frac{1}{2}\gamma_k^2\gamma_{k-1}^2 D^3\phi(x_{k-2})v_{k-2}^{\otimes 3}-2\gamma_k^2\gamma_{k-2}^2 D^2\phi(x_{k-2})v_{k-2}^{\otimes 2}$$
$$-u\gamma_k^2\gamma_{k-1}^2\langle D^2\phi(x_{k-2}); v_{k-2}, \nabla f(x_{k-2})\rangle$$
$$+\gamma_k^2\langle D^2\phi(x_{k-2}); \sqrt{u}\sigma_{k-1}^{(1)}U_{k-1}^{(1)}, 2\sqrt{u}\sigma_{k-1}^{(2)}U_{k-1}^{(2)}\rangle+o(\gamma_{k-1}^3)$$

Since $\gamma_{k-1}-\gamma_k=o(\gamma_k^4)$, we can change $\gamma_k$ on the left hand side to $\gamma_{k-1}$ when we take limits with scale $\Gamma_n^{(4)}$. Apply theorem 9 with $p=8, s=1$ to terms with order $o(\gamma_k^3)$-coefficients.

$$\lim_n\frac{1}{\Gamma_n^{(4)}}\sum_{k=2}^n\gamma_k^2\mathcal{L}\phi(x_{k-1})=\lim_n\frac{1}{\Gamma_n^{(4)}}\sum_{k=2}^n\gamma_{k-1}^2\mathcal{L}\phi(x_{k-2})-2\lim_n\frac{1}{\Gamma_n^{(4)}}\sum_{k=2}^n\gamma_{k-1}^3\mathcal{L}\phi(x_{k-2})$$

$$-2\lim_n \frac{1}{\Gamma_n^{(4)}}\sum_{k=2}^n \gamma_{k-1}^3\left(\frac{u}{2}\nabla\phi(x_{k-2})\cdot\nabla f(x_{k-2})-\frac{1}{2}D^2\phi(x_{k-2})v_{k-2}^{\otimes 2}\right)$$

$$-3\int_{\mathbb{R}^{2d}}D^2\phi(x)v^{\otimes 2}\nu(dx,dv)+u\int_{\mathbb{R}^d}\nabla\phi(x)\cdot\nabla f(x)\pi(dx)$$

$$+\lim_n\frac{1}{\Gamma_n^{(4)}}\sum_{k=2}^n\gamma_{k-1}^2\langle D^2\phi(x_{k-2});\sqrt{u}\sigma_{k-1}^{(1)}U_{k-1}^{(1)},2\sqrt{u}\sigma_{k-1}^{(2)}U_{k-1}^{(2)}\rangle$$

Since we proved $\frac{1}{\Gamma_n^{(4)}}\sum_{k=1}^n\gamma_k^3\mathcal{L}\phi(x_{k-1})\to 0$ and from Theorem 5, we've shown that

$$\lim_n\frac{1}{\Gamma_n^{(4)}}\sum_{k=2}^n\gamma_{k-1}^2\langle D^2\phi(x_{k-2});\sqrt{u}\sigma_{k-1}^{(1)}U_{k-1}^{(1)},2\sqrt{u}\sigma_{k-1}^{(2)}U_{k-1}^{(2)}\rangle=2u\int_{\mathbb{R}^d}\Delta\phi(x)\pi(dx)$$

We obtain

$$\lim_n\frac{1}{\Gamma_n^{(4)}}\sum_{k=2}^n\gamma_{k-1}^2\left(\frac{u}{2}\nabla\phi(x_{k-2})\cdot\nabla f(x_{k-2})-\frac{1}{2}D^2\phi(x_{k-2})v_{k-2}^{\otimes 2}\right)$$

$$=\frac{1}{2}\left[\lim_n\frac{1}{\Gamma_n^{(4)}}\sum_{k=2}^n\gamma_k^2\mathcal{L}\phi(x_{k-1})-\lim_n\frac{1}{\Gamma_n^{(4)}}\sum_{k=2}^n\gamma_{k-1}^2\mathcal{L}\phi(x_{k-2})\right]$$

$$=0$$

Therefore, $\lim_n\frac{1}{\Gamma_n^{(4)}}\sum_{k=1}^n\gamma_k^3\left(\frac{u}{2}\nabla\phi(x_{k-1})\cdot\nabla f(x_{k-1})-\frac{1}{2}D^2\phi(x_{k-1})v_{k-1}^{\otimes 2}\right)=0$.

To prove the **Claim**, we need to use the decomposition again:

$$\sum_{k=1}^n\gamma_k^2\mathcal{L}\phi(x_{k-1})=\sum_{k=1}^n\left\{\gamma_k[\phi(x_k)-\phi(x_{k-1})]+\left(\gamma_k^3-\frac{2}{3}\gamma_k^4\right)v_{k-1}\cdot\nabla\phi(x_{k-1})\right.$$

$$+\frac{u}{2}\left(\gamma_k^3-\frac{2}{3}\gamma_k^4\right)\nabla f(x_{k-1})\cdot\nabla\phi(x_{k-1})$$

$$-\frac{1}{2}\left(\gamma_k^3-2\gamma_k^4\right)D^2\phi(x_{k-1})v_{k-1}^{\otimes 2}$$

$$-\frac{2u}{3}\gamma_k^4 D^2\phi(x_{k-1})U_k^{(1)\otimes 2}$$

$$+\frac{u}{2}\gamma_k^4\langle D^2\phi(x_{k-1});v_{k-1},\nabla f(x_{k-1})\rangle$$

$$\left.-\frac{1}{6}\gamma_k^4 D^3\phi(x_{k-1})v_{k-1}^{\otimes 3}+o(\gamma_k^4)\right\}$$

Since $\{\phi(x_n)\}$ is tight and $\gamma_{k-1}-\gamma_k=o(\gamma_k^4)$, we have $\frac{1}{\Gamma_n^{(4)}}\sum_{k=1}^n\gamma_k(\phi(x_k)-\phi(x_{k-1}))\to 0$. For the terms with coefficients of order $\gamma_k^3$, we can apply theorem 9 with $p=8,s=1$. Then we obtain:

$$\lim_n\frac{1}{\Gamma_n^{(4)}}\sum_{k=1}^n\gamma_k^2\mathcal{L}\phi(x_{k-1})=\lim_n\frac{1}{\Gamma_n^{(4)}}\sum_{k=1}^n\gamma_k^3\left(\mathcal{L}\phi(x_{k-1})+\frac{u}{2}\nabla\phi(x_{k-1})\cdot\nabla f(x_{k-1})-\frac{1}{2}D^2\phi(x_{k-1})v_{k-1}^{\otimes 2}\right)$$

$$-\frac{u}{3}\int_{\mathbb{R}^d}\nabla\phi(x)\cdot\nabla f(x)\pi(dx)+\int_{\mathbb{R}^{2d}}D^2\phi(x)v^{\otimes 2}\nu(dx,dv)$$

$$-\frac{2u}{3}\int_{\mathbb{R}^d}\int_{\mathbb{R}^d}D^2\phi(x)z^{\otimes 2}\mu(dz)\pi(dx)$$

$$=\lim_n\frac{1}{\Gamma_n^{(4)}}\sum_{k=1}^n\gamma_k^3\left(\mathcal{L}\phi(x_{k-1})+\frac{u}{2}\nabla\phi(x_{k-1})\cdot\nabla f(x_{k-1})-\frac{1}{2}D^2\phi(x_{k-1})v_{k-1}^{\otimes 2}\right)$$

$$=0$$

The second identity follows from integration by parts and Fubini theorem. The last identity follows from the two statements we just proved. $\qquad\square$