[Reviews · NeurIPS 2020]

Review 1

Summary and Contributions: The paper deals with the computation of an unknown expectation w.r.t. a strongly log concave distribution \pi using MCMC methods. The method studied are 1. the recently proposed Randomized midpoint (RULMC), which a discretization of the underdamped Langevin diffusion 2. a new MCMC algorithm called RLMC that can be seen a the randomized midpoint discretization technique applied to the overdamped Langevin diffusion. RULMC was studied recently and is somehow optimal for the task of **sampling** w.r.t to \pi using a Langevin diffusion. Numerical integration is a task a bit different and the authors study RULMC and RLMC for the purpose of numerical integration by establishing a Central Limit Theorem (CLT) for both algorithms. In my opinion, the question is relevant but no surprising result is shown. The authors make the following theoretical contributions (there is no numerical experiments): Prop 2.1: Complexity of RLMC (to achieve \varpesilon accuracy) in 2-Wasserstein distance with constant step size Th 1: ergodicity of the Markov kernel given by RLMC Prop 2.2: 2-Wassertein distance between the invariant measure of RLMC and \pi Prop 2.3: Complexity of RLMC (to achieve \varpesilon accuracy) in 2-Wasserstein distance with decreasing step size Th 2: CLT for RLMC Similar results for RULMC Th 3. Similar to Th 1 Prop 3.1. Similar to Prop 2.2 Prop 3.2. Similar to Prop 2.3 Th 4. Similar to Th 2. Th 5. Generalized version of Th 4. The results are discussed in details below

Strengths: The questions addressed by the paper are natural and already answered for various simpler forms of Langevin algorithm. As RULMC is optimal in some sense, proving a CLT for RULMC is important for the understanding of the algorithm Prop 2.1: The complexity result Prop 2.1 is the analogue of a result obtained in SL19 for RULMC (more precisely, the result showing that RULMC is optimal). The complexity obtained in Prop 2.1 is not optimal but shows some improvement over the vanilla overdamped Langevin algorithm. Similar complexity was obtained for other versions of Langevin algorithm using more advanced techniques, in this sense Prop 2.1 is interesting. Th 1: The ergodicity of RLMC is important and is in my opinion the first result one should try to prove when looking at a new markov chain Prop 2.2. : Such bias result is a standard question in the Langevin literature and perhaps as important as the complexity results. In fact, complexity results like Prop 2.1 and a bias results are usually closely related, although here they are obtained using different techniques. Prop 2.3 is similar to Prop 2.1 but using a decreasing step size. The authors discuss the minor differences in remark 2. Th 2 is the main result regarding RLMC. It provides a CLT for the estimator of the unknown expectation to be computed, allowing to understand the fluctuations of this estimator. Prop 3.1. shows an improvement over Prop 2.2 when RULMC is used. The bias of RULMC is proven to be O(h^{3/2}). I don't know if this strong result is known. Th 4 which applies to RULMC shows some improvement over RLMC. Th 5 generalizes Th 5 which is appreciated.

Weaknesses: On the other hand, there is no surprising result as far as I understand. Moreover, there is no numerical experiments. As the authors mentioned, their results can be useful for computing confidence intervals. This point could for example be discussed in an experimental section. Prop 2.1 is a side result as the authors said, and is obtained by imitating the proof of SL19. Moreover, the author only consider the case where the algorithm is initialized from a minimizer of f. Why? Same question for Prop 2.3 and 3.2. Th 1: Such purely asymptotic result may not be relevant for the neurips community. Prop 2.2. : The order of the bias is worse than the bias of the vanilla overdamped Langevin algorithm (O(h)). Since the result is not obtained similarly as the complexity result Prop 2.1 I am wondering if Prop 2.2 is tight. Th 2: I cannot evaluate this theorem due to a lack of clarity. First I don't understand the difference between biaised CLT and unbiased CLT. Why is the mean of the limit distribution not equal to the expectation to be computed? What is an "optimal" step size. Moreover, a better description of the set of \varphi considered would be appreciated, e.g. sufficient conditions to have \varphi = \mathcal A \phi for some \phi. Does the limit of \hat{\gamma}_n always exist? It is not clear to me. More importantly, the CLT proven does not show any improvement w.r.t. to CLT obtained for simpler versions of Langevin algorithm. Therefore Th 2 is indeed a negative result, which limits the impact of RLMC. Th 3. Same as Th. 1 Prop 3.1 seems to be an easy corollary of SL19, am I correct? Moreover, the constant are large.

Correctness: The appendix is quite long but the proofs look correct. There is no numerical experiment.

Clarity: The paper is in general well written, with a structure allowing to compare the two algorithms (RLMC and RULMC). My main concern regarding clarity is the extensive use of unclear pronouns 'it', 'this', 'that'. Sometimes, the reader does not know to what they refer to, making reading more difficult. Moreover, the paragraph "our contributions" can be simplified to be clearer and the CLT theorems could be better explained to highlight the contributions. Specifically, the comment after each CLT is hard to understand.

Relation to Prior Work: The paper answers natural question regarding RULMC (and RLMC) and studies pros and cons of both approaches. I think that such questions were answered for various other forms of Langevin algorithm (I am not sure though). These results could have been discussed more in details for the sake of comparison (for instance the CLTs). I don't see clear comparison between the obtained results and existing ones (except a quick comparison to Euler discretization of Langevin equation). That's why I think that there is no surprising result in the paper.

Reproducibility: Yes

Additional Feedback: typos l. 93 and l. 195 ----------------------------------------------------------------------------------------- Thanks for the response file. I don't understand (1). Th4: Do the authors mean that the CLT rates of RULMC are better than the CLT rates of Euler LMC? Is it explained in the paper? I assume (but I am not sure) that the authors meant that a discussion similar to Remark 3 is possible for Th4, and that this discussion implies some improvement. Prop 2.1: Yes I had understood, this is an interesting point Initialization: I know, but it means that the analysis should start by assuming that the initialization is an epsilon-minimizer, not a true one. However, I know that many papers make this assumption and I am not penalizing this paper because of this assumption. Th 1 and 3, Ergodicity: OK Prop 2.2, 3.1: OK. I means that RMP induces more bias than Euler although it provides better W2 rates. Bias CLT. Thanks for the clarification! No surprising result: What satisfies me is that you provided CLTs which can be seen as lower bounds and upper bounds at the same time. So if there is no improvement, this is the nature of the problem (and not an upperbound being not tight). Few weeks after reviewing this paper, I realize that my main concern is that the CLT are not well explained. I suggest the authors to put more emphasis in this, which is the main contribution. Meanwhile, I appreciated some of the answers. I also acknowledge that the authors are asking an interesting question regarding CLTs, which is not enough studied for sampling algorithms, even though W2 rates are often provided.


Review 2

Summary and Contributions: In this paper, some probabilistic properties of the randomized midpoint discretization method are studied. The authors evaluated the bias of the stationally distribution to the target distribution when the constant step-size is used in the algorithm. Then, for the decreasing step-size choice, the asymptotic normality of the numerical integration using the randomized midpoint is established. The above results are obtained for both over-damped and under-damped Langevin dynamics.

Strengths: The results in this paper seem mathematically sound. According to the description of the relation to the previous works, the theoretical findings in this paper are sufficiently novel.

Weaknesses: Showing some numerical experiments would be of great help for the reader to understand the validity of the theoretical findings. Even if there is a bias for the randomized midpoint method with a constant step-size, the method can be practical by taking into account the trade-off between the bias and variance. Showing some insight on that point based on theoretical findings and/or numerical experiments would be nice. On the CLT result, complicated variance term appears. Can the authors show some simple examples?

Correctness: All proofs are shown in the long appendix. I could not check details of all proofs.

Clarity: The description of the paper is clear.

Relation to Prior Work: Relation to prior work is clearly discussed.

Reproducibility: Yes

Additional Feedback: Thanks for the response file. I keep my current score. This work provide mathematically rigorous results. I believe that additional examples and numerical results will strengthen the results.


Review 3

Summary and Contributions: The authors study the convergence properties of the "randomized midpoint method," which is a novel way to discretize an SDE introduced in Shen and Lee 2019. The authors here study the overdamped and underdamped Langevin diffusions and show these new discretizations are ergodic for a constant step size h. Aadditionally, they quantify the size of this asymptotic bias in the Wasserstein metric as a function of h, and finally they prove a CLT for test functions that have a solution to the Stein equation.

Strengths: The main strength of the paper is the proof of a CLT for this new type of discretization (for some test functions which are known to be mean zero a priori). In the overdamped case, the rates obtained are similar to those from Langevin Monte Carlo (LMC), but in the underdamped case, the rates are better than the overdamped case. The other contributions are useful too, but the ergodic theorems are minor, and the underdamped bias contributions is already mentioned in Shen and Lee, so only the overdamped bias proposition is novel.

Weaknesses: The main novel results are the W_2 bias bounds for the overdamped Langevin (this is b/c Shen & Lee basically already had done the underdamped case) and the CLT results for both diffusions. The former result is significant (as it is competitive with the other works cited) but it less novel because it is an application of previous methods to the overdamped case. It would also be great to see some examples comparing the results from the underdamped and overdamped diffusions. Having some empirical work that corroborated the CLT theorem would be a natural improvement for the paper.

Correctness: The propositions and theorems mentioned in the paper look correct.

Clarity: Yes, the paper is well-written (there are few typos).

Relation to Prior Work: The authors do a decent job of comparing their work to others in the field. However, the Shen and Lee paper actually do a better job of comparing all the recent methods (see their Table 1). One place I think the comparisons are lacking is with Theorem 4/5. What are the state-of-the-art rates for the underdamped CLT? I didn't see it mentioned in the paper.

Reproducibility: Yes

Additional Feedback: L131: Should just be "Before we state..." L202: Theorem 2: It might be helpful to state why the mean of the CLT is 0 in condition (i). It also might be worth reiterating that even though this only applies to functions that are a priori mean zero under pi, these CLTs can still apply to test functions which are not mean zero under pi as long as their demeaned version has a solution to the Stein equation. L235: Why are U_1, U_2, U_3 in R^{3d}? L236-237: Why are these covariances only equal to scalars? Shouldn't they be matrices? L305: What is the significance of O(n^{5/8})? Why is the "optimal"? ==== Post-rebuttal ==== Upon review of all the comments and author rebuttal, I am sticking with my current score. The work does seem promising, but the impact is still a bit limited, which is the main reason I'm not a bigger supporter.

[Author Response · NeurIPS 2020]

We thank the reviewers for the feedback. We first highlight our main contributions and then respond to specific points.

**R1, R2, R4: Contributions:** (1) We emphasize that to the best of our knowledge, Theorem 4 in our paper is the **first**
**result** to show that for a specific class of test functions, one obtains **improved asymptotic CLT rates** by discretizing
**Underdamped Langevin Diffusion (ULD)** using Euler or Randomized Mid-Point (RMP) methods. Prior works
demonstrated improved $W_2$ rates for sampling with Euler or RMP discretizations of ULD compared to Overdamped
Langevin Diffusion (OLD). However, such results are not insightful as far as the CLT rate for numerical integration is
concerned, which is the focus of our paper. (2) Surprisingly, while **RMP discretization** obtains improved $W_2$ rates for
sampling over Euler discretization, it **has the same CLT rates as Euler discretization** in both OLD and ULD settings.
This points to the second major contribution of our submission: One should take the $W_2$ rate improvements with a
pinch of salt if the main goal is to compute integrals, arguably the main application of samplers in machine learning.

**R1: Regarding Proposition 2.1** - As discussed in Lines 142-145, we would like to clarify that, to obtain the same $W_2$
rates based on OLD, other discretizations require higher-order smoothness assumption whereas Proposition 2.1 only
requires gradient-smoothness. **Initialization:** For the RMP discretization, if the initialization is far from the minimizer,
we have to decrease the step size much quicker as large step size in the region where gradient is large (far from
minimizer) can result in very inaccurate discretization of the process. In this sense, the rate will deteriorate. Also in the
regime we consider, obtaining a point that is $\epsilon$ close to such a minimizer only costs $\mathcal{O}(\log(1/\epsilon))$ iterations. Compared
to the sampling rate, which is $\mathcal{O}(1/\epsilon)$, this is significantly cheaper. **Thms 1 and 3, and Ergodicity:** Ergodicity results
in Thms 1 and 3 are important properties of RLMC and RULMC, and they form the basis of the CLT results. In other
words, to obtain confidence intervals (CIs) which is through establishing a CLT, stated (asymptotic) ergodicity results
are sufficient. We emphasize that it is possible to obtain geometric ergodicity results (with rates); however, we preferred
not to add this, as they are not needed for establishing CLT results, which are our main focus. **Prop 2.2:** Indeed this
is an important observation and the bias is higher than vanilla Euler discretization of OLD. This is also reflected in
the CLT bias term $\rho$ in Lines 208-209. **Prop 3.1:** We emphasize that the conclusion in Prop 3.1 does not follow from
[SL19]. Indeed, one possibility to use the result in [SL19] to obtain bias in $W_2$ metric is to let the iterations go to
infinity in their main theorem. However, their result (and proof) assume an upper bound on the number of iterations
(depending on the choices of $\epsilon$). This leads to a worse than our stated result. **Biased/Unbiased CLT & Limit of**
$\hat{\gamma}_n$ **always exist?** These two aspects are related and form the crux of our main result. Note that in Theorem 2, lines 205,
206 and 207 respectively, we consider three cases: when the limit $\hat{\gamma}_\infty := \lim_{n\to\infty} \hat{\gamma}_n$ is (i) equal to 0, (ii) between 0
and $\infty$, and (iii) equal to $+\infty$ (and hence doesn't exist). Next, note that in the definition of $\pi_n^\gamma(\varphi)$, we have already
centered with the true expectation (which is zero in our setup by definition). Only when the limit $\hat{\gamma}_\infty$ equals zero, we
obtain an unbiased CLT, meaning the normal distribution is exactly centered at the true expectation. If the limit $\hat{\gamma}_\infty$ is in
the interval $(0, \infty)$, the CLT is biased meaning it is no longer centered around the truth – as a consequence, we need to
do appropriate bias-correction when obtaining practical confidence intervals. When the limit $\hat{\gamma}_\infty$ is $+\infty$, we converge
to a degenerate random variable (i.e., a constant). We hope this clarifies our main results. The same interpretation
applies also to Theorem 4. **Comparison & No surprising result is shown:** Please see Lines 2-10 above for our main
contributions. Apart from the result in [LP02] (which we have compared against), we are not aware of any CLT results,
in particular for ULD. We would greatly appreciate pointers to specific papers if the reviewer thinks otherwise – we
would be happy to cite them and compare our results with those in the suggested papers.

**R2: Bias variance Trade-off:** With respect to sampling (i.e., $W_2$ rates), roughly speaking, the optimal rate obtained
in [SL19] is exactly based on picking a constant step-size by trading off bias and variance. However, as we show, to
obtain a CLT for numerical integration, especially the one centered on the true value of the integral, one needs to have a
specific decreasing step-size choice. **Complicated terms in CLT:** We will add specific examples in our revision, to
provide more insights. However, the main take-away from our general results is the asymptotic rate improvement of
CLT with RMP/Euler discretization of ULD. **Simulations:** We definitely agree with the reviewer that adding numerical
experiments would be enlightening. We will add simulations to the camera-ready version if the paper is accepted.

**R4: Comparison:** Note that [LS19] provides a table comparing results only for the $W_2$ rates for which there are several
related works to compare against. Our main contribution in this work is the CLT rate improvement with the RMP
discretization of ULD. We are not aware of prior works in this direction, to the best of our knowledge. Nevertheless,
we will take R4's advice and add a table comparing CLT rates for RMP and Euler discretizations of both ULD and
OLD established in our paper and [LP02]. If the reviewer is aware of any other state-of-the-art CLT results for some
discretization of ULD or OLD, we greatly appreciate it and would add it to the table in our revision. **L235-**
**237:** We mean the tuple $(U_1, U_2, U_3)$ is in $R^{3d}$ i.e., $U_i \in \mathbb{R}^d$, for $i = 1, 2, 3$. Also, we missed an identity matrix in the
covariance definition. Thanks for catching this. We will fix these typos in our revision. **L131, L202:** These suggestions
are well-taken and we would incorporate them in our revision. **L305:** $\mathcal{O}(n^{5/8})$ is the best achievable rate by picking for
$\alpha$ among polynomially decreasing step-size choices of the form $\gamma_k = k^{-\alpha}$. Hence, we call it *optimal* following the
terminology in [LP02]. We will clarify it in our revision as *optimal among polynomially decreasing step-size choices.*

[Meta-Review · NeurIPS 2020]

The paper deals with the computation of an unknown expectation w.r.t. a strongly log concave distribution \pi using MCMC methods. The referees found the results to be generally solid, if not surprising, and of limited impact. From the discussion: "I still believe that the authors can clarify their paper, especially discussing the CLTs obtained (which is the main contribution). Moreover, they do not have surprising results like rates improvements, except in a specific case, as far as I understand. However, they are tackling an interesting question and the fact that they provided CLT, which can be seen as upper bounds and lower bounds at the same time, satisfies me. I am also satisfied by some of their answers."